# GRADIENT DESCENT ON NEURAL NETWORKS TYPICALLY OCCURS AT THE EDGE OF STABILITY

**Jeremy Cohen  Simran Kaur  Yuanzhi Li  J. Zico Kolter[1]  and  Ameet Talwalkar[2]**
Carnegie Mellon University and: [1]Bosch AI  [2]Determined AI
Correspondence to: `jeremycohen@cmu.edu`

## ABSTRACT

We empirically demonstrate that full-batch gradient descent on neural network training objectives typically operates in a regime we call the Edge of Stability. In this regime, the maximum eigenvalue of the training loss Hessian hovers just above the value $2/(\text{step size})$, and the training loss behaves non-monotonically over short timescales, yet consistently decreases over long timescales. Since this behavior is inconsistent with several widespread presumptions in the field of optimization, our findings raise questions as to whether these presumptions are relevant to neural network training. We hope that our findings will inspire future efforts aimed at rigorously understanding optimization at the Edge of Stability.

## 1 INTRODUCTION

Neural networks are almost never trained using (full-batch) gradient descent, even though gradient descent is the conceptual basis for popular optimization algorithms such as SGD. In this paper, we train neural networks using gradient descent, and find two surprises. First, while little is known about the dynamics of neural network training in general, we find that in the special case of gradient descent, there is a simple characterization that holds across a broad range of network architectures and tasks. Second, this characterization is strongly at odds with prevailing beliefs in optimization.

In more detail, as we train neural networks using gradient descent with step size $\eta$, we measure the evolution of the *sharpness* — the maximum eigenvalue of the training loss Hessian. Empirically, the behavior of the sharpness is consistent across architectures and tasks: so long as the sharpness is less than the value $2/\eta$, it tends to continually rise (§3.1). We call this phenomenon *progressive sharpening*. The significance of the value $2/\eta$ is that gradient descent on quadratic objectives is unstable if the sharpness exceeds this threshold (§2). Indeed, in neural network training, if the sharpness ever crosses $2/\eta$, gradient descent quickly becomes destabilized — that is, the iterates start to oscillate with ever-increasing magnitude along the direction of greatest curvature. Yet once

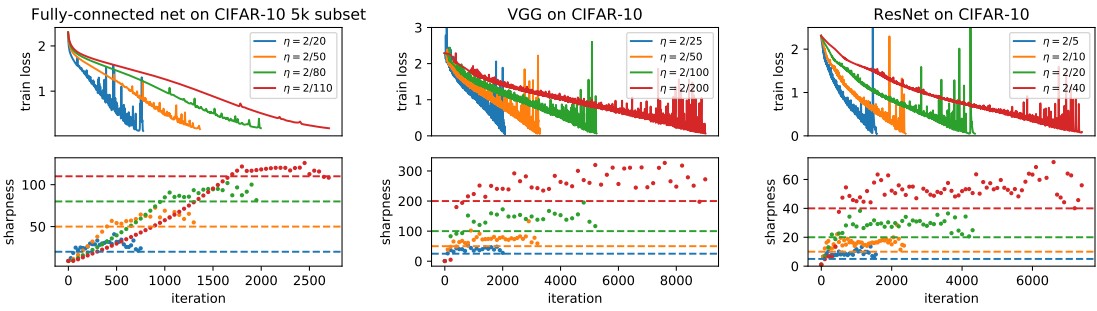

Figure 1: **Gradient descent typically occurs at the Edge of Stability.** On three separate architectures, we run gradient descent at a range of step sizes $\eta$, and plot both the train loss (top row) and the sharpness (bottom row). For each step size $\eta$, observe that the sharpness rises to $2/\eta$ (marked by the horizontal dashed line of the appropriate color) and then hovers right at, or just above, this value.

this happens, gradient descent does not diverge entirely or stall. Instead, it enters a new regime we call the *Edge of Stability*[1] (§3.2), in which (1) the sharpness hovers right at, or just above, the value $2/\eta$; and (2) the train loss behaves non-monotonically, yet consistently decreases over long timescales. In this regime, gradient descent is constantly "trying" to increase the sharpness, but is constantly restrained from doing so. The net effect is that gradient descent continues to successfully optimize the training objective, but in such a way as to avoid further increasing the sharpness.[2]

In principle, it is possible to run gradient descent at step sizes $\eta$ so small that the sharpness never rises to $2/\eta$. However, these step sizes are suboptimal from the point of view of training speed, sometimes dramatically so. In particular, for standard architectures on the standard dataset CIFAR-10, such step sizes are so small as to be completely unreasonable — at all reasonable step sizes, gradient descent eventually enters the Edge of Stability (see §4). Thus, at least for standard networks on CIFAR-10, the Edge of Stability regime should be viewed as the "rule," not the "exception."

As we describe in §5, the Edge of Stability regime is inconsistent with several pieces of conventional wisdom in optimization theory: convergence analyses based on $L$-smoothness or monotone descent, quadratic Taylor approximations as a model for local progress, and certain heuristics for step size selection. We hope that our empirical findings will both nudge the optimization community away from widespread presumptions that appear to be untrue in the case of neural network training, and also point the way forward by identifying precise empirical phenomena suitable for further study.

Certain aspects of the Edge of Stability have been observed in previous empirical studies of full-batch gradient descent (Xing et al., 2018; Wu et al., 2018); our paper provides a unified explanation for these observations. Furthermore, Jastrzębski et al. (2020) proposed a simplified model for the evolution of the sharpness during *stochastic* gradient descent which matches our empirical observations in the special case of full-batch SGD (i.e. gradient descent). However, outside the full-batch special case, there is no evidence that their model matches experiments with any degree of quantitative precision, although their model does successfully predict the *directional* trend that large step sizes and/or small batch sizes steer SGD into regions of low sharpness. We discuss SGD at greater length in §6. To summarize, while the sharpness does not obey simple dynamics during SGD (as it does during GD), there are indications that the "Edge of Stability" intuition might generalize somehow to SGD, just in a way that does not center around the sharpness.

## 2 BACKGROUND: STABILITY OF GRADIENT DESCENT ON QUADRATICS

In this section, we review the stability properties of gradient descent on quadratic functions. Later, we will see that the stability of gradient descent on neural training objectives is partly well-modeled by the stability of gradient descent on the quadratic Taylor approximation.

On a quadratic objective function $f(\mathbf{x}) = \frac{1}{2}\mathbf{x}^T\mathbf{A}\mathbf{x} + \mathbf{b}^T\mathbf{x} + c$, gradient descent with step size $\eta$ will diverge if[3] any eigenvalue of $\mathbf{A}$ exceeds the threshold $2/\eta$. To see why, consider first the one-dimensional quadratic $f(x) = \frac{1}{2}ax^2 + bx + c$, with $a > 0$. This function has optimum $x^* = -b/a$. Consider running gradient descent with step size $\eta$ starting from $x_0$. The update rule is $x_{t+1} = x_t - \eta(ax_t + b)$, which means that the error $x_t - x^*$ evolves as $(x_{t+1} - x^*) = (1 - \eta a)(x_t - x^*)$. Therefore, the error at step $t$ is $(x_t - x^*) = (1 - \eta a)^t(x_0 - x^*)$, and so the iterate at step $t$ is $x_t = (1 - \eta a)^t(x_0 - x^*) + x^*$. If $a > 2/\eta$, then $(1 - \eta a) < -1$, so the sequence $\{x_t\}$ will oscillate around $x^*$ with ever-increasing magnitude, and diverge.

Now consider the general $d$-dimensional case. Let $(a_i, \mathbf{q}_i)$ be the $i$-th largest eigenvalue/eigenvector of $\mathbf{A}$. As shown in Appendix B, when the gradient descent iterates $\{\mathbf{x}_t\}$ are expressed in the special coordinate system whose axes are the eigenvectors of $\mathbf{A}$, each coordinate evolves separately. In particular, the coordinate for each eigenvector $\mathbf{q}_i$, namely $\langle \mathbf{q}_i, \mathbf{x}_t \rangle$, evolves according to the dynamics of gradient descent on a one-dimensional quadratic objective with second derivative $a_i$.

---

[1]This nomenclature was inspired by the title of Giladi et al. (2020).

[2]In the literature, the term "sharpness" has been used to refer to a variety of quantities, often connected to generalization (e.g. Keskar et al. (2016)). In this paper, "sharpness" strictly means the maximum eigenvalue of the training loss Hessian. We do not claim that this quantity has any connection to generalization.

[3]For *convex* quadratics, this is "if and only if." However, if $\mathbf{A}$ has a negative eigenvalue, then gradient descent with any (positive) step size will diverge along the corresponding eigenvector.

Therefore, if $a_i > 2/\eta$, then the sequence $\{\langle \mathbf{q}_i, \mathbf{x}_t \rangle\}$ will oscillate with ever-increasing magnitude; in this case, we say that the iterates $\{\mathbf{x}_t\}$ diverge *along the direction* $\mathbf{q}_i$. To illustrate, Figure 2 shows a quadratic function with eigenvalues $a_1 = 20$ and $a_2 = 1$. In Figure 2(a), we run gradient descent with step size $\eta = 0.09$; since $0 < a_2 < a_1 < 2/\eta$, gradient descent converges along both $\mathbf{q}_1$ and $\mathbf{q}_2$. In Figure 2(b), we use step size $\eta = 0.11$; since $0 < a_2 < 2/\eta < a_1$, gradient descent converges along $\mathbf{q}_2$ yet diverges along $\mathbf{q}_1$, so diverges overall.

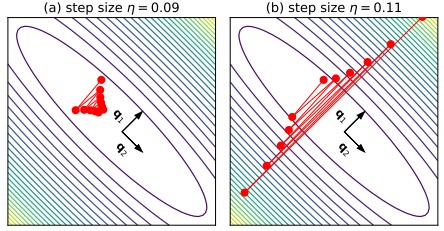

Figure 2: Gradient descent on a quadratic with eigenvalues $a_1 = 20$ and $a_2 = 1$.

Polyak momentum (Polyak, 1964) and Nesterov momentum (Nesterov, 1983; Sutskever et al., 2013) are notable variants of gradient descent which often improve the convergence speed. On quadratic functions, these two algorithms also diverge if the sharpness exceeds a certain threshold, which we call the "maximum stable sharpness," or MSS. In particular, we prove in Appendix B that gradient descent with step size $\eta$ and momentum parameter $\beta$ diverges if the sharpness exceeds:

$$\text{MSS}_{\text{Polyak}}(\eta, \beta) = \frac{1}{\eta}\left(2 + 2\beta\right), \quad \text{MSS}_{\text{Nesterov}}(\eta, \beta) = \frac{1}{\eta}\left(\frac{2 + 2\beta}{1 + 2\beta}\right). \tag{1}$$

The Polyak result previously appeared in Goh (2017); the Nesterov one seems to be new. Note that this discussion only applies to full-batch gradient descent. As we discuss in §6, several recent papers have proposed stability analyses for SGD (Wu et al., 2018; Jastrzębski et al., 2020).

Neural network training objectives are not globally quadratic. However, the second-order Taylor approximation around any point $\mathbf{x}_0$ in parameter space is a quadratic function whose "$\mathbf{A}$" matrix is the Hessian at $\mathbf{x}_0$. If any eigenvalue of this Hessian exceeds $2/\eta$, gradient descent with step size $\eta$ would diverge if run on this quadratic function — the iterates would oscillate with ever-increasing magnitude along the corresponding eigenvector. Therefore, at any point $\mathbf{x}_0$ in parameter space where the sharpness exceeds $2/\eta$, gradient descent with step size $\eta$ would diverge if run on the quadratic Taylor approximation to the training objective around $\mathbf{x}_0$.

## 3 GRADIENT DESCENT ON NEURAL NETWORKS

In this section, we empirically characterize the behavior of gradient descent on neural network training objectives. Section 4 will show that this characterization holds broadly.

### 3.1 PROGRESSIVE SHARPENING

When training neural networks, it seems to be a general rule that **so long as the sharpness is small enough for gradient descent to be stable** ($< 2/\eta$, for vanilla gradient descent), **gradient descent has an overwhelming tendency to continually increase the sharpness**. We call this phenomenon *progressive sharpening*. By "overwhelming tendency," we mean that gradient descent can occasionally decrease the sharpness (especially at the beginning of training), but these brief decreases always seem be followed by a return to continual increase. Jastrzębski et al. (2020) previously hypothesized (in their Assumption 4) that a similar phenomenon may hold for SGD, but the evidence for, and the precise scope of, this effect are both currently far clearer for gradient descent than for SGD.

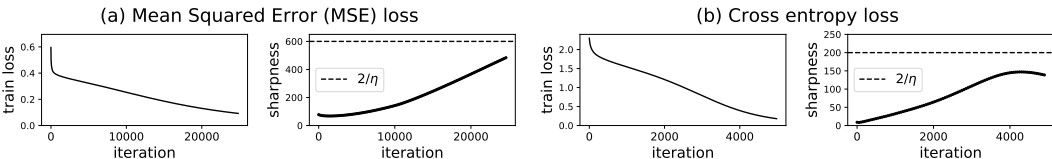

Figure 3: **So long as the sharpness is less than $2/\eta$, it tends to continually increase during gradient descent.** We train a network to completion (99% accuracy) using gradient descent with a very small step size. We consider both MSE loss (**left**) and cross-entropy loss (**right**).

Progressive sharpening is illustrated in Figure 3. Here, we use (full-batch) gradient descent to train a network on a subset of 5,000 examples from CIFAR-10, and we monitor the evolution of the sharpness during training. The network is a fully-connected architecture with two hidden layers of width 200, and tanh activations. In Figure 3(a), we train using the mean squared error loss for classification (Hui & Belkin, 2020), encoding the correct class with 1 and the other classes with 0. We use the small step size of $\eta = 2/600$, and stop when the training accuracy reaches 99%. We plot both the train loss and the sharpness, with a horizontal dashed line marking the stability threshold $2/\eta$. Observe that the sharpness continually rises during training (except for a brief dip at the beginning). This is progressive sharpening. For this experiment, we intentionally chose a step size $\eta$ small enough that the sharpness remained beneath $2/\eta$ for the entire duration of training.

**Cross-entropy.** When training with cross-entropy loss, there is an exception to the rule that the sharpness tends to continually increase: with cross-entropy loss, the sharpness typically drops at the end of training. This behavior can be seen in Figure 3(b), where we train the same network using the cross-entropy loss rather than MSE. This drop occurs because once most data points are classified correctly, gradient descent tries to drive the cross-entropy loss to zero by scaling up the margins, as detailed in Soudry et al. (2018). As we explain in Appendix C, this causes the sharpness to drop.

**The effect of width.** It is known that when networks parameterized in a certain way (the "NTK parameterization") are made infinitely wide, the Hessian moves a vanishingly small amount during training (Jacot et al., 2018; Lee et al., 2019; Li & Liang, 2018), which implies that no progressive sharpening occurs. In Appendix D, we experiment with networks of varying width, under both NTK and standard parameterizations. We find that progressive sharpening occurs to a lesser degree as networks become increasingly wide. Nevertheless, our experiments in §4 demonstrate that progressive sharpening occurs to a dramatic degree for standard architectures on the standard dataset CIFAR-10.

We do not know why progressive sharpening occurs, or whether "sharp" solutions differ in any important way from "not sharp" solutions. These are important questions for future work. Note that Mulayoff & Michaeli (2020) studied the latter question in the context of deep linear networks.

## 3.2 THE EDGE OF STABILITY

In the preceding section, we ran gradient descent using step sizes $\eta$ so small that the sharpness never reached the stability threshold $2/\eta$. In Figure 4(a), we start to train the same network at the larger step size of $\eta = 0.01$, and pause training once the sharpness rises to $2/\eta = 200$. Recall from §2 that in any region where the sharpness exceeds $2/\eta$, gradient descent with step size $\eta$ would be unstable if run on the quadratic Taylor approximation to the training objective — the gradient descent iterates would oscillate with ever-increasing magnitude along the leading Hessian eigenvector. Empirically, we find that gradient descent on the real neural training objective behaves similarly — at first. Namely, let $\mathbf{q}_1$ be the leading Hessian eigenvector at the iteration where the sharpness reaches $2/\eta$. In Figure 4(b), we resume training the network, and we monitor both the train loss and the quantity $\langle \mathbf{q}_1, \mathbf{x}_t \rangle$ for the next 215 iterations. Observe that $\langle \mathbf{q}_1, \mathbf{x}_t \rangle$ oscillates with ever-increasing magnitude, similar to the divergent quadratic example in Figure 2(b). At first, these oscillations are too small to affect the objective appreciably, and so the train loss continues to monotonically decrease. But eventually, these oscillations grow big enough that the train loss spikes.

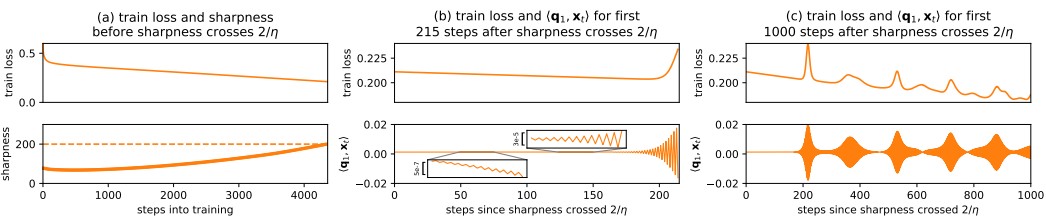

Figure 4: **Once the sharpness crosses $2/\eta$, gradient descent becomes destabilized**. We run gradient descent at $\eta = 0.01$. **(a)** The sharpness eventually reaches $2/\eta$. **(b)** Once the sharpness crosses $2/\eta$, the iterates start to oscillate along $\mathbf{q}_1$ with ever-increasing magnitude. **(c)** Somehow, GD does not diverge entirely; instead, the train loss continues to decrease, albeit non-monotonically.

Once gradient descent becomes destabilized in this manner, classical optimization theory gives no clues as to what will happen next. One might imagine that perhaps gradient descent might diverge entirely, or that gradient descent might stall while failing to make progress, or that gradient descent might jump to a flatter region and remain there. In reality, none of these outcomes occurs. In Figure 4(c), we plot both the train loss and $\langle \mathbf{q}_1, \mathbf{x}_t \rangle$ for 1000 iterations after the sharpness first crossed $2/\eta$. Observe that gradient descent somehow avoids diverging entirely. Instead, after initially spiking around iteration 215, the train loss continues to decrease, albeit non-monotonically.

This numerical example is representative. In general, after the sharpness initially crosses $2/\eta$, gradient descent enters a regime we call the *Edge of Stability*, in which (1) the sharpness hovers right at, or just above, the value $2/\eta$; and (2) the train loss behaves non-monotonically over short timescales, yet decreases consistently over long timescales. Indeed, in Figure 5, we run gradient descent at a range of step sizes using both MSE and cross-entropy loss. The left plane plots the train loss curves, with a vertical dotted line (of the appropriate color) marking the iteration where the sharpness first crosses $2/\eta$. Observe that the train loss decreases monotonically before this dotted line, but behaves non-monotonically afterwards. The middle plane plots the evolution of the sharpness, with a horizontal dashed line (of the appropriate color) at the value $2/\eta$. Observe that once the sharpness reaches $2/\eta$, it ceases to increase further, and instead hovers right at, or just above, the value $2/\eta$ for the remainder of training. (The precise meaning of "just above" varies: in Figure 5, for MSE loss, the sharpness hovers just a minuscule amount above $2/\eta$, while for cross-entropy loss, the gap between the sharpness and $2/\eta$ is small yet non-miniscule.)

At the Edge of Stability, gradient descent is "trying" to increase the sharpness further, but is being restrained from doing so. To demonstrate this, in Figure 7, we train at step size $2/200$ until reaching the Edge of Stability, and then at iteration 6,000 (marked by the vertical black line), we drop the step size to $\eta = 2/300$. Observe that after the learning rate drop, the sharpness immediately starts to increase, and only stops increasing once gradient descent is back at the Edge of Stability. Appendix O repeats this experiment on more architectures. Intuitively, gradient descent with fixed step sizes acts like a constrained optimization algorithm: the use of step size $\eta$ imposes an implicit $2/\eta$ constraint on the sharpness (Nar & Sastry, 2018), and at the Edge of Stability this constraint is "active."

Observe from Figure 5 that there do exist step sizes $\eta$ (in purple) small enough that the sharpness never rises to $2/\eta$. We call such a step size *stable*. However, observe that with cross-entropy loss, it takes 3700 iterations to train at the stable step size in purple, but only 1000 iterations to train at the larger step size in blue. In general, we always observe that stable step sizes are suboptimal in terms

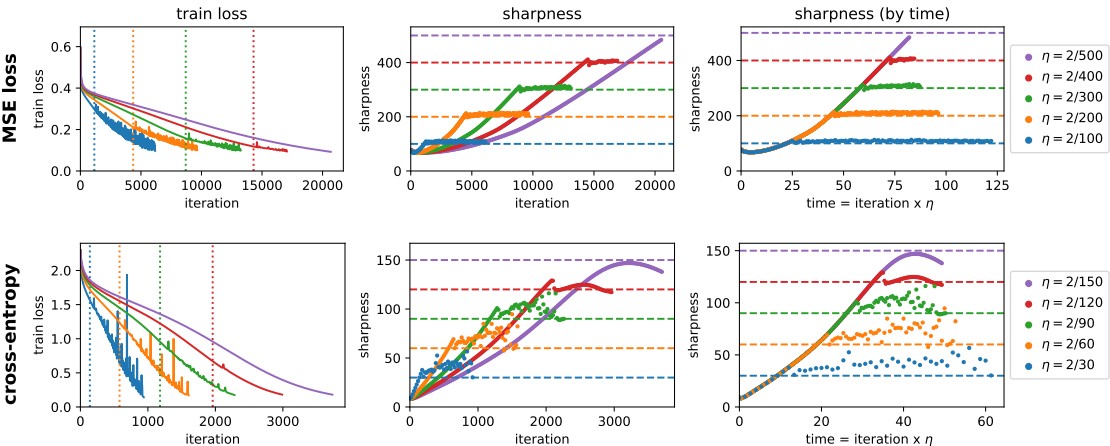

Figure 5: **After the sharpness reaches $2/\eta$, gradient descent enters the Edge of Stability**. A network is trained with gradient descent at a range of step sizes (see legend), using both MSE loss (**top row**) and cross-entropy (**bottom row**). **Left**: the train loss curves, with a vertical dotted line at the iteration where the sharpness first crosses $2/\eta$. **Center**: the sharpness, with a horizontal dashed line at the value $2/\eta$. **Right**: sharpness plotted by time (= iteration $\times \eta$) rather than iteration.

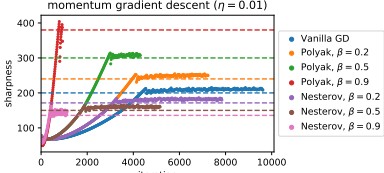
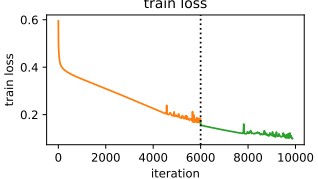
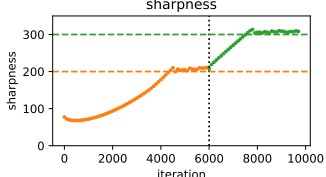

Figure 6: **Momentum.** We run GD with step size $\eta = 0.01$ and Polyak or Nesterov momentum at various $\beta$. For each algorithm, the horizontal dashed line marks the MSS from Equation 1.

Figure 7: **After a learning rate drop, progressive sharpening resumes**. We start training at $\eta = 2/200$ (orange) and then after 6000 iterations (dotted vertical black line), we cut the step size to $\eta = 2/300$ (green). Observe that as soon as the step size is cut, the sharpness starts to rise.

of convergence speed. In fact, in §4 we will see that for standard networks on CIFAR-10, stable step sizes are so suboptimally small that they are completely unreasonable.

The "Edge of Stability" effect generalizes to gradient descent with momentum. In Figure 6, we train using gradient descent with step size $\eta = 0.01$, and varying amounts of either Polyak or Nesterov momentum. Observe that in each case, the sharpness rises until reaching the MSS given by Equation 1, and then plateaus there. Appendix N has more momentum experiments.

In Appendix P, we briefly examine the evolution of the next few Hessian eigenvalues during gradient descent. We find that each of these eigenvalues rises until plateauing near $2/\eta$.

**Prior work**. Aspects of the Edge of Stability have been observed previously in the literature. Wu et al. (2018) noted that the sharpness at the solution reached by full-batch gradient descent was not just *less than* $2/\eta$, as was expected due to stability considerations, but was mysteriously *approximately equal* to $2/\eta$. In retrospect, we can attribute this observation to progressive sharpening. Xing et al. (2018) observed that full-batch gradient descent eventually enters a regime (the Edge of Stability) in which the training loss behaves non-monotonically, and the iterates oscillate along the direction of largest curvature; however, they did not relate this regime to the sharpness. Lewkowycz et al. (2020) found that in neural network training, if the sharpness at initialization is larger than $2/\eta$, then after becoming initially destabilized, gradient descent does not always diverge entirely (as the quadratic Taylor approximation would suggest), but rather sometimes "catapults" into a flatter region that is flat enough to stably accommodate the step size. It seems plausible that whichever properties of neural training objectives permit this so-called "catapult" behavior may also be the same properties that permit successful optimization at the Edge of Stability. Indeed, optimization at the Edge of Stability can conceivably be viewed as a never-ending series of micro-catapults. As we discuss at greater length in §6, several papers (Jastrzębski et al., 2017; 2019) have observed that large step sizes steer *stochastic* gradient descent into less sharp regions of the loss landscape, and Jastrzębski et al. (2020) attributed this effect to the stability properties of SGD. Finally, our precise characterization of the behavior of the sharpness during full-batch gradient descent adds to a growing body of work that empirically investigates the Hessian spectrum of neural networks (Sagun et al., 2017; Ghorbani et al., 2019; Li et al., 2020a; Papyan, 2018; 2019; 2020).

### 3.3 THE GRADIENT FLOW TRAJECTORY

In the right pane of Figure 5, we plot the evolution of the sharpness during gradient descent, with "time" = iteration $\times \eta$, rather than iteration, on the x-axis. This allows us to directly compare the sharpness after, say, 100 iterations at $\eta = 0.01$ to the sharpness after 50 iterations at $\eta = 0.02$; both are time 1. Observe that when plotted by time, the sharpnesses for gradient descent at different step sizes coincide until the time where each reaches $2/\eta$. This is because for this network, gradient descent at $\eta = 0.01$ and gradient descent at $\eta = 0.02$ initially travel the same path (moving at a speed proportional to $\eta$) until each reaches the point on that path where the sharpness hits $2/\eta$. This path is the *gradient flow trajectory*. The gradient flow solution at time $t$ is defined as the limit as $\eta \to 0$ of the gradient descent iterate at iteration $t/\eta$ (if this limit exists). The empirical finding of interest is that for this particular network, gradient descent does not only track the gradient flow trajectory in the limit of infinitesimally small step sizes, but for any step size that is less than 2/sharpness.

We can numerically approximate gradient flow trajectories by using the Runge-Kutta RK4 algorithm (Press et al., 1992) to numerically integrate the gradient flow ODE. Empirically, for many *but not all* networks studied in this paper, we find that gradient descent at any step size $\eta$ closely tracks the Runge-Kutta trajectory until reaching the point on that trajectory where the sharpness hits $2/\eta$. (This sometimes occurs even for networks with ReLU activations or max-pooling, which give rise to training objectives that are not continuously differentiable, which means that the gradient flow trajectory is not necessarily guaranteed to exist.) For such networks, the gradient flow trajectory provides a coherent framework for reasoning about which step sizes will eventually enter the Edge of Stability. Let $\lambda_0$ be the sharpness at initialization, and let $\lambda_{\max}$ be the maximum sharpness along the gradient flow trajectory. If $\eta < 2/\lambda_{\max}$, then gradient descent will stably track the gradient flow trajectory for the entire duration of training, and will never enter the Edge of Stability. On the other hand, if $\eta \in [2/\lambda_{\max}, 2/\lambda_0]$, then gradient descent will stably track the gradient flow trajectory only until reaching the point on that trajectory where the sharpness hits $2/\eta$; shortly afterwards, gradient descent will become destabilized, depart the gradient flow trajectory, and enter the Edge of Stability.

## 4 FURTHER EXPERIMENTS

Section 3 focused for exposition on a single architecture and task. In this section, we show that our characterization of gradient descent holds broadly across a wide range of architectures and tasks. We detail several known caveats and qualifications in Appendix A.

**Architectures.** In Appendix J, we fix the task of training a 5k subset of CIFAR-10, and we systematically vary the network architecture. We consider fully-connected networks, as well as convolutional networks with both max-pooling and average pooling. For all of these architectures, we consider tanh, ReLU, and ELU activations, and for fully-connected networks we moreover consider softplus and hardtanh —- eleven networks in total. We train each network with both cross-entropy and MSE loss. In each case, we successfully reproduce Figure 5.

Since batch normalization (Ioffe & Szegedy, 2015) is known to have unusual optimization properties (Li & Arora, 2019), it is natural to wonder whether our findings still hold with batch normalization. In Appendix K, we confirm that they do, and we reconcile this point with Santurkar et al. (2018).

**Tasks.** In Appendix L, we verify our findings on: (1) a Transformer trained on the WikiText-2 language modeling task; (2) fully-connected tanh networks with one hidden layer, trained on a synthetic one-dimensional toy regression task; and (3) deep linear networks trained on Gaussian data. In each case, the sharpness rises until hovering right at, or just above, the value $2/\eta$.

**Standard networks on CIFAR-10.** In Appendix M, we verify our findings on three standard architectures trained on the full CIFAR-10 dataset: a ResNet with BN, a VGG with BN, and a VGG without BN. For all three architectures, we find that progressive sharpening occurs to a dramatic degree, and, relatedly, that stable step sizes are dramatically suboptimal. For example, when we train the VGG-BN to 99% accuracy using gradient flow / Runge-Kutta, we find that the sharpness rises from 6.3 at initialization to a peak sharpness of 2227.6. Since this is an architecture for which gradient descent closely hews to the gradient flow trajectory, we can conclude that any stable step size for gradient descent would need to be less than $2/2227.6 = 0.000897$. Training finishes at time 14.91, so gradient descent at any stable step size would require at least $14.91/0.000897 = 16,622$ iterations. Yet empirically, this network can be trained to completion at the larger, "Edge of Stability" step size of $\eta = 0.16$ in just 329 iterations. Therefore, training at a stable step size is suboptimal by a factor of at least $16622/329 = 50.5$. The situation is similar for the other two architectures we consider. In short, for standard architectures on the standard dataset CIFAR-10, stable step sizes are not just suboptimally small, they are so suboptimal as to be completely unreasonable. For these networks, gradient descent at any reasonable step size eventually enters the Edge of Stability regime.

**Tracking gradient flow.** Recall that for some (but not all) architectures, gradient descent closely hews to the gradient flow trajectory so long as the sharpness is less than $2/\eta$. Among the architectures considered in Appendix J, we found this to be true for the architectures with continuously differentiable components, as well as some, but not all, with ReLU, hardtanh, and max-pooling. Among the architectures in Appendix L, we found this to be true for the tanh network, but *not* for the deep linear network or the Transformer. Finally, we did find this to be true for the three standard architectures in Appendix M, even though those architectures use ReLU.

## 5 DISCUSSION

We now explain why the behavior of gradient descent at the Edge of Stability contradicts several pieces of conventional wisdom in optimization.

**At reasonable step sizes, gradient descent cannot be analyzed using (even local) $L$-smoothness** Many convergence analyses of gradient descent assume a bound on the sharpness — either globally or, at the very least, along the optimization trajectory. This condition, called $L$-smoothness (Nesterov, 1998), is intended to guarantee that each gradient step will decrease the training objective by a certain amount; the weakest guarantee is that if the local sharpness is less than $2/\eta$, then a gradient step with size $\eta$ is guaranteed to decrease (rather than increase) the training objective. At a bare minimum, any convergence analysis of gradient descent based on $L$-smoothness will require the sharpness along the optimization trajectory to be less than $2/\eta$. Yet to the contrary, at the Edge of Stability, the sharpness hovers just above $2/\eta$. Therefore, at any step size for which gradient descent enters the Edge of Stability (which, on realistic architectures, includes any reasonable step size), gradient descent cannot be analyzed using $L$-smoothness. Li et al. (2020b) previously argued that convergence analyses based on $L$-smoothness do not apply to networks with both batch normalization and weight decay; our paper empirically extends this to neural networks without either.

**$L$-smoothness *may* be inappropriate when analyzing other optimization algorithms too** It is common for optimization papers seemingly motivated by deep learning to analyze algorithms under the "non-convex but $L$-smooth"' setting (Reddi et al., 2016; Agarwal et al., 2017; Zaheer et al., 2018; Zhou et al., 2018; Chen et al., 2019; Li & Orabona, 2019; Ward et al., 2019; You et al., 2020; Vaswani et al., 2019; Sankararaman et al., 2019; Reddi et al., 2021; Défossez et al., 2020; Xie et al., 2020; Liu et al., 2020; Defazio, 2020). Since our experiments focus on gradient descent, it does not *necessarily* follow that $L$-smoothness assumptions are unjustified when analyzing other optimization algorithms. However, gradient descent is arguably the simplest optimization algorithm, so we believe that the fact that (even local) $L$-smoothness fails even there should raise serious questions about the suitability of the $L$-smoothness assumption in neural network optimization more generally. In particular, the burden of proof should be on authors to empirically justify this assumption.

**At reasonable step sizes, gradient descent does not monotonically decrease the training loss** In neural network training, SGD does not monotonically decrease the training objective, in part due to minibatch randomness. However, it is often assumed that *full-batch* gradient descent *would* monotonically decrease the training objective, were it used to train neural networks. For example, Zhang et al. (2020) proposed a "relaxed $L$-smoothness" condition that is less restrictive than standard $L$-smoothness, and proved a convergence guarantee for gradient descent under this condition which asserted that the training objective will decrease monotonically. Likewise, some neural network analyses such as Allen-Zhu et al. (2019) also assert that the training objective will monotonically decrease. Yet, at the Edge of Stability, the training loss behaves non-monotonically over short timescales even as it consistently decreases over long timescales. Therefore, convergence analyses which assert monotone descent cannot possibly apply to gradient descent at reasonable step sizes.

**The Edge of Stability is inherently non-quadratic** It is tempting to try to reason about the behavior of gradient descent on neural network training objectives by analyzing, as a proxy, the behavior of gradient descent on the local quadratic Taylor approximation (LeCun et al., 1998). However, at the Edge of Stability, the behavior of gradient descent on the real neural training objective is irreconcilably different from the behavior of gradient descent on the quadratic Taylor approximation: the former makes consistent (if choppy) progress, whereas the latter would diverge (and this divergence would happen quickly, as we demonstrate in Appendix E). Thus, the behavior of gradient descent at the Edge of Stability is inherently non-quadratic.

**Dogma for step size selection may be unjustified** An influential piece of conventional wisdom concerning step size selection has its roots in the quadratic Taylor approximation model of gradient descent. This conventional wisdom (LeCun et al., 1993; 1998; Schaul et al., 2013) holds that if the sharpness at step $t$ is $\lambda_t$, then the current step size $\eta_t$ must be set no greater than $2/\lambda_t$ (in order to prevent divergence); and furthermore, barring additional information about the objective function, that $\eta_t$ should optimally be set to $1/\lambda_t$ . Our findings complicate this conventional wisdom.

To start, it is nearly impossible to satisfy these prescriptions with a fixed step size: for any fixed (and reasonable) step size $\eta_t = \eta$, progressive sharpening eventually drives gradient descent into regions where the sharpness is just a bit greater than $2/\eta$ — which means that the step size $\eta$ is purportedly impermissible. Furthermore, in Appendix F, we try running gradient descent with the purportedly optimal $\eta_t = 1/\lambda_t$ rule, and find that this algorithm is soundly outperformed by the purportedly impermissible baseline of gradient descent with a fixed $\eta_t = 1/\lambda_0$ step size, where $\lambda_0$ is the sharpness at initialization. The $\eta_t = 1/\lambda_t$ rule continually anneals the step size, and in so doing ensures that the training objective will decrease at each iteration, whereas the fixed $\eta_t = 1/\lambda_0$ step size often increases the training objective. However, this non-monotonicity turns out to be a worthwhile price to pay in return for the ability to take larger steps.

## 6 STOCHASTIC GRADIENT DESCENT

Our precise characterization of the behavior of the sharpness only applies to full-batch gradient descent. In contrast, during SGD, the sharpness does not always settle at any fixed value (Appendix G), let alone one that can be numerically predicted from the hyperparameters. Nevertheless, prior works (Jastrzębski et al., 2017; 2019; 2020) have demonstrated that large step sizes do steer SGD into regions of the landscape with lower sharpness; the $2/\eta$ rule we have identified for full-batch gradient descent is a special case of this observation. Furthermore, small *batch sizes* also steer SGD into regions with lower sharpness (Keskar et al., 2016; Jastrzębski et al., 2017), as we illustrate in Appendix G. Jastrzębski et al. (2020) attributed this phenomenon to the stability properties of SGD.

Even though our findings only strictly hold for gradient descent, they may have relevance to SGD as well. First, since gradient descent is a special case of SGD, any general characterization of the dynamics of SGD must reduce to the Edge of Stability in the full-batch special case. Second, there are indications that the Edge of Stability may have some analogue for SGD. One way to interpret our main findings is that gradient descent "acclimates" to the step size in such a way that each update sometimes increases and sometimes decrease the train loss, yet an update with a smaller step size would consistently decrease the training loss. Along similar lines, in Appendix H we demonstrate that SGD "acclimates" to the step size and batch size in such a way that each SGD update sometimes increases and sometimes decreases the training loss in expectation, yet an SGD update with a smaller step size or larger batch size would consistently decrease the training loss in expectation.

In extending these findings to SGD, the question arises of how to model "stability" of SGD. This is a highly active area of research. Wu et al. (2018) proposed modeling stability *in expectation*, and gave a sufficient (but not necessary) criterion for the stability of SGD in expectation. Building on this framework, Jastrzębski et al. (2020) argued that if the Hessian is aligned with the second moment matrix of per-example gradients, then SGD is stable so long as a certain expression (involving the sharpness) is below a certain threshold. In the special full-batch case, their criterion reduces to the sharpness being beneath $2/\eta$ — a constraint which we have shown is "tight" throughout training. However, in the general SGD case, there is no evidence that their stability constraint is tight throughout training. Giladi et al. (2020) showed that the generalization gap in asynchronous SGD can be mostly ameliorated by setting the step size so as to ensure that stability properties in expectation remain identical to those of a well-tuned implementation of synchronous SGD. Finally, a number of papers have attempted to mathematically model the propensity of SGD to "escape from sharp minima" (Hu et al., 2017; Zhu et al., 2019; Xie et al., 2021).

## 7 CONCLUSION

We have empirically demonstrated that the behavior of gradient decent on neural training objectives is both surprisingly consistent across architectures and tasks, and surprisingly different from that envisioned in the conventional wisdom. Our findings raise a number of questions. Why does progressive sharpening occur? At the Edge of Stability, by what mechanism does gradient descent avoid diverging entirely? Since the conventional wisdom for step size selection is wrong, how should the gradient descent step size be set during deep learning? Does the "Edge of Stability" effect generalize in some way to optimization algorithms beyond gradient descent, such as SGD? We hope to inspire future efforts aimed at addressing these questions.

## 8 ACKNOWLEDGEMENTS

This work was supported in part by DARPA FA875017C0141, the National Science Foundation grants IIS1705121 and IIS1838017, an Amazon Web Services Award, a JP Morgan A.I. Research Faculty Award, a Carnegie Bosch Institute Research Award, a Facebook Faculty Research Award, and a Block Center Grant. Any opinions, findings and conclusions or recommendations expressed in this material are those of the author(s) and do not necessarily reflect the views of DARPA, the National Science Foundation, or any other funding agency.

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

## A  CAVEATS

In this appendix, we list several caveats to our generic characterization of the dynamics of full-batch gradient descent on neural network training objectives.

**1. With cross-entropy loss, the sharpness often drops at the end of training**   As mentioned in the main text, when neural networks are trained on classification tasks using the cross-entropy loss, the sharpness frequently drops near the end of training, once the classification accuracy begins to approach 1. We explain this effect in Appendix C.

**2.  For shallow or wide networks, or on simple datasets, sharpness doesn't rise *that* much**
When the network is shallow (Figure 16 - 17) or wide (Figure 12 - 14), or when the dataset is "easy" or small (Figure 18), the sharpness may rise only a small amount over the gradient flow trajectory. For these optimization problems, "stable step sizes" (those for which gradient descent never enters the Edge of Stability) may be quite reasonable, and the range of "Edge of Stability" step sizes may be quite small.

**3.  Sharpness sometimes drops at the beginning of training**   We sometimes observe that the sharpness drops at the very beginning of training, as the network leaves its initialization. This was more common when training with MSE loss than with cross-entropy loss. For most networks, this drop was very slight. However, the combination of *both* batch normalization and MSE loss sometimes caused situations where the sharpness was considerably large at initialization, and dropped precipitously as soon as training began. Figure 8 illustrates one such network.

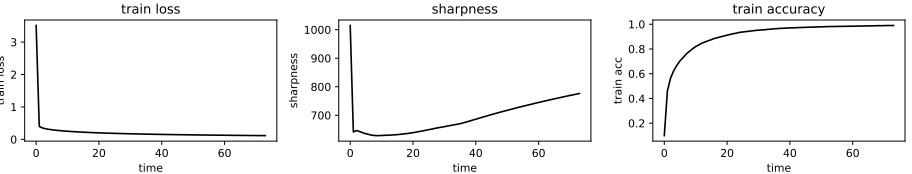

Figure 8: **With batch norm and MSE loss, the sharpness sometimes drops precipitously**: We use gradient flow to train a CNN with ReLU activations and batch norm using the MSE loss. Observe that the sharpness drops substantially at the beginning of training, and never recovers to its initial value. This seems to occur due to the combination of both batch normalization and MSE loss.

**4.  With batch normalization, need to look at sharpness *between* iterates**   As detailed in Appendix K, when we trained batch-normalized networks using very small step sizes $\eta$, we sometimes observed that the sharpness *at the gradient descent iterates themselves* plateaued below $2/\eta$, even though the sharpness *in between the iterates* plateaued just above $2/\eta$, as expected.

**5.  With non-differentiable components, instability sometimes begins when the sharpness is a bit *less than* $2/\eta$**   When training networks with ReLU or hardtanh activations, we sometimes observed that the sharpness started to plateau (and the training loss started to behave non-monotonically) a bit *before* the instant when the sharpness crossed $2/\eta$. For example, see Figures 41, 43 or Figures 45, 47. One potential explanation is that for such networks, the training objective is not continuously differentiable, so the second-order Taylor approximation around a given iterate may be a poor local model for the training objective at even an tiny distance away in weight space, if that weight change causes some activations to switch "ReLU case" from $\leq 0$ to $> 0$, or visa versa.

# B  STABILITY OF GRADIENT DESCENT ON QUADRATIC FUNCTIONS

This appendix describes the stability properties of gradient descent (and its momentum variants) when optimizing the quadratic objective function

$$f(\mathbf{x}) = \frac{1}{2}\mathbf{x}^T\mathbf{A}\mathbf{x} + \mathbf{b}^T\mathbf{x} + c \qquad (2)$$

starting from the initialization $\mathbf{x}_0$.

To review, vanilla gradient descent is defined by the iteration:

$$\mathbf{x}_{t+1} = \mathbf{x}_t - \eta\nabla f(\mathbf{x}_t).$$

Meanwhile, gradient descent with Polyak (also called "heavy ball") momentum (Polyak, 1964; Sutskever et al., 2013; Goodfellow et al., 2016) is defined by the iteration:

$$\mathbf{v}_{t+1} = \beta\mathbf{v}_t - \eta\nabla f(\mathbf{x}_t)$$
$$\mathbf{x}_{t+1} = \mathbf{x}_t + \mathbf{v}_{t+1}$$

where $\mathbf{v}_t$ is a "velocity" vector and $0 \leq \beta < 1$ is the momentum coefficient. For $\beta = 0$ the algorithm reduces to vanilla GD.

Finally, Nesterov momentum Sutskever et al. (2013); Goodfellow et al. (2016) is an adaptation of Nesterov's accelerated gradient (Nesterov, 1983) for deep learning defined by the iteration:

$$\mathbf{v}_{t+1} = \beta\mathbf{v}_t - \eta\nabla f(\mathbf{x}_t + \beta\mathbf{v}_t)$$
$$\mathbf{x}_{t+1} = \mathbf{x}_t + \mathbf{v}_{t+1}$$

where $\mathbf{v}_t$ is a "velocity" vector and $0 \leq \beta < 1$ is the momentum coefficient. For $\beta = 0$ the algorithm reduces to vanilla GD.

All three of these algorithms share a special property: on quadratic functions, they act independently along each Hessian eigenvector. That is, if we express the iterates in the Hessian eigenvector basis, then in this basis the coordinates evolve independent from one another under gradient descent.

**Proposition 1.** *Consider running vanilla gradient descent on the quadratic objective (2) starting from* $\mathbf{x}_0$. *Let* $(\mathbf{q}, a)$ *be an eigenvector/eigenvalue pair of* $\mathbf{A}$. *If* $a > 2/\eta$, *then the sequence* $\{\mathbf{q}^T\mathbf{x}_t\}$ *will diverge.*

*Proof.* The update rule for gradient descent on this quadratic function is:

$$\mathbf{x}_{t+1} = \mathbf{x}_t - \eta(\mathbf{A}\mathbf{x}_t + \mathbf{b})$$
$$= (\mathbf{I} - \eta\mathbf{A})\mathbf{x}_t - \eta\mathbf{b}.$$

Therefore, the quantity $\mathbf{q}^T\mathbf{x}_t$ evolves under gradient descent as:

$$\mathbf{q}^T\mathbf{x}_{t+1} = \mathbf{q}^T(\mathbf{I} - \eta\mathbf{A})\mathbf{x}_t - \eta\mathbf{b}$$
$$= (1 - \eta a)\mathbf{q}^T\mathbf{x}_t - \eta\,\mathbf{q}^T\mathbf{b}. \qquad (\mathbf{q}^T\mathbf{A} = a\mathbf{q})$$

Define $\tilde{x}_t = \mathbf{q}^T\mathbf{x}_t + \frac{1}{a}\mathbf{q}^T\mathbf{b}$, and note that $\{\mathbf{q}^T\mathbf{x}_t\}$ diverges if and only if $\{\tilde{x}_t\}$ diverges.

The quantity $\tilde{x}_t$ evolves under gradient descent according to the simple rule:

$$\tilde{x}_{t+1} = (1 - \eta a)\tilde{x}_t$$

Since $\eta > 0$, if $a > 2/\eta$ then $(1 - \eta a) < -1$, so the sequence $\{\tilde{x}_t\}$ will diverge. $\qquad\square$

We now prove analogous results for Nesterov and Polyak momentum.

**Theorem 1.** *Consider running Nesterov momentum on the quadratic objective (2) starting from any initialization. Let $(\mathbf{q}, a)$ be an eigenvector/eigenvalue pair of $\mathbf{A}$. If $a > \frac{1}{\eta}\left(\frac{2+2\beta}{1+2\beta}\right)$, then the sequence $\{\mathbf{q}^T\mathbf{x}_t\}$ will diverge.*

*Proof.* The update rules for Nesterov momentum on this quadratic function are:

$$\mathbf{v}_{t+1} = \beta(\mathbf{I} - \eta\mathbf{A})\mathbf{v}_t - \eta\mathbf{b} - \eta\mathbf{A}\mathbf{x}_t$$
$$\mathbf{x}_{t+1} = \mathbf{x}_t + \mathbf{v}_{t+1}.$$

Using the fact that $\mathbf{v}_t = \mathbf{x}_t - \mathbf{x}_{t-1}$, we can rewrite this as a recursion in $\mathbf{x}_t$ alone:

$$\mathbf{x}_{t+1} = \mathbf{x}_t + \beta(\mathbf{I} - \eta\mathbf{A})(\mathbf{x}_t - \mathbf{x}_{t-1}) - \eta\mathbf{b} - \eta\mathbf{A}\mathbf{x}_t$$
$$= (1 + \beta)(\mathbf{I} - \eta\mathbf{A})\mathbf{x}_t - \beta(\mathbf{I} - \eta\mathbf{A})\mathbf{x}_{t-1} - \eta\mathbf{b}$$

Define $\tilde{x}_t = \mathbf{q}^T\mathbf{x}_t + \frac{1}{a}\mathbf{q}^T\mathbf{b}$, and note that $\mathbf{q}^T\mathbf{x}_t$ diverges iff $\tilde{x}_t$ diverges. It can be seen that $\tilde{x}_t$ evolves as:

$$\tilde{x}_{t+1} = (1 + \beta)(1 - \eta a)\tilde{x}_t - \beta(1 - \eta a)\tilde{x}_{t-1}$$

This is a linear homogenous second-order difference equation. By Theorem 2.37 in Elaydi (2005), since $\eta > 0$ and $\beta < 1$, if $a > \frac{1}{\eta}\left(\frac{2+2\beta}{1+2\beta}\right)$ then this recurrence diverges.

$\square$

The following result previously appeared in Goh (2017).

**Theorem 2.** *Consider running Polyak momentum on the quadratic objective (2) starting from any initialization. Let $(\mathbf{q}, a)$ be an eigenvector/eigenvalue pair of $\mathbf{A}$. If $a > \frac{1}{\eta}(2 + 2\beta)$, then the sequence $\{\mathbf{q}^T\mathbf{x}_t\}$ will diverge.*

*Proof.* Using the fact that $\mathbf{v}_t = \mathbf{x}_t - \mathbf{x}_{t-1}$, we can re-write the Polyak momentum recursion as a recursion in $\mathbf{x}$ alone:

$$\mathbf{x}_{t+1} = \mathbf{x}_t + \beta\mathbf{v}_t - \eta\nabla f(\mathbf{x}_t)$$
$$= \mathbf{x}_t + (\beta(\mathbf{x}_t - \mathbf{x}_{t-1}) - \eta\nabla f(\mathbf{x}_t))$$
$$= (1 + \beta)\mathbf{x}_t - \beta\mathbf{x}_{t-1} - \eta\nabla f(\mathbf{x}_t).$$

For the quadratic objective (2), this update rule amounts to:

$$\mathbf{x}_{t+1} = (1 + \beta)\mathbf{x}_t - \beta\mathbf{x}_{t-1} - \eta(\mathbf{A}\mathbf{x}_t + \mathbf{b})$$
$$= (1 + \beta - \eta\mathbf{A})\mathbf{x}_t - \beta\mathbf{x}_{t-1} - \eta\mathbf{b}.$$

Multiplying by $\mathbf{q}^T$, we obtain:

$$\mathbf{q}^T\mathbf{x}_{t+1} = (1 + \beta - \eta a)\mathbf{q}^T\mathbf{x}_t - \beta\mathbf{q}^T\mathbf{x}_{t-1} - \eta\mathbf{q}^T\mathbf{b}.$$

Now, define $\tilde{x}_t = \mathbf{q}^T\mathbf{x}_t + \frac{1}{a}\mathbf{q}^T\mathbf{b}$. Note that $\mathbf{q}^T\mathbf{x}_t$ diverges iff $\tilde{x}_t$ diverges. It can be seen that $\tilde{x}_t$ evolves as:

$$\tilde{x}_{t+1} = (1 + \beta - \eta a)\tilde{x}_t - \beta\tilde{x}_{t-1}.$$

This is a linear homogenous second-order difference equation. By Theorem 2.37 in Elaydi (2005), since $\eta > 0$ and $\beta < 1$, if $a > \frac{1}{\eta}(2 + 2\beta)$ then this recurrence diverges. $\square$

## C  CROSS-ENTROPY LOSS

In this appendix, we explain why the sharpness decreases at the end of training when the cross-entropy loss is used. Before considering the full multiclass case, let us first consider the simpler case of binary classification with the logistic loss.

**Binary classification with logistic loss**  We consider a dataset $\{\mathbf{x}_i, y_i)\}_{i=1}^n \subset \mathbb{R}^d \times \{-1, 1\}$, where the examples are vectors in $\mathbb{R}^d$ and the labels are binary $\{-1, 1\}$. We consider a neural network $h : \mathbb{R}^d \times \mathbb{R}^p \to \mathbb{R}$ which maps an input $\mathbf{x} \in \mathbb{R}^d$ and a parameter vector $\theta \in \mathbb{R}^p$ to a prediction $h(\mathbf{x}; \theta) \in \mathbb{R}$. Let $\ell : \mathbb{R} \times \{-1, 1\} \to \mathbb{R}$ be the logistic loss function:

$$\ell(z; y) = \log(1 + \exp(-zy))$$

$$= -\log p \quad \text{where} \quad p = \frac{1}{1 + \exp(-zy)}.$$

The second derivative of this loss function w.r.t $z$ is:

$$\ell''(z; y) = p(1 - p).$$

The full training objective is:

$$f(\theta) = \frac{1}{n} \sum_{i=1}^n f_i(\theta) \qquad \text{where} \quad f_i(\theta) = \ell(f(\mathbf{x}_i; \theta); y_i).$$

The Hessian of this training objective is the average of per-example Hessians:

$$\nabla^2 f(\theta) = \frac{1}{n} \sum_{i=1}^n \nabla^2 f_i(\theta).$$

For any arbitrary loss function $\ell$, we have the so-called Gauss-Newton decomposition (Martens, 2016; Bottou et al., 2018) of the per-example Hessian $\nabla^2 f_i(\theta)$:

$$\nabla^2 f_i(\theta) = \ell''(z_i; y_i) \, \nabla_\theta h(\mathbf{x}_i; \theta) \, \nabla_\theta h(\mathbf{x}_i; \theta)^T + \ell'(z_i; y_i) \, \nabla_\theta^2 h(\mathbf{x}_i; \theta) \quad \text{where} \quad z_i = h(\mathbf{x}_i; \theta)$$

where $\nabla_\theta h(\mathbf{x}_i; \theta) \in \mathbb{R}^p$ is the gradient of the network output with respect to the weights, and $\ell'$ refers to the derivative of $\ell$ with respect to its *first* argument, the score.

Empirically, the first term in the decomposition (usually called the "Gauss-Newton matrix") tends to dominate the second, which implies the following "Gauss-Newton approximation" to the Hessian:

$$\nabla^2 f(\theta) \approx \frac{1}{n} \sum_{i=1}^n \ell''(z_i; y_i) \, \nabla_\theta h(\mathbf{x}_i; \theta) \, \nabla_\theta h(\mathbf{x}_i; \theta)^T \tag{3}$$

In our experience, progressive sharpening affects $\nabla_\theta h(\mathbf{x}_i; \theta) \, \nabla_\theta h(\mathbf{x}_i; \theta)^T$. That is, $\nabla_\theta h(\mathbf{x}_i; \theta) \, \nabla_\theta h(\mathbf{x}_i; \theta)^T$ tends to grow in scale continually during training. For the square loss $\ell(z; y) = \frac{1}{2}(z - y)^2$, the second derivative $\ell''(z; y) = 1$ is the identity, so the $\nabla^2 f_i(\theta)$ grows continually as well. In contrast, for the logistic loss, many of the $\ell''(z_i; y_i)$ decrease at the very end of training. Why is this? In Figure 9, we plot both the logistic loss $\ell$, and its second derivative $\ell''$, as a function of the quantity $yz$, which is often called the "margin."

Crucially, observe that both $\ell$ and $\ell''$ are decreasing in $yz$. Because the loss $\ell$ is decreasing in $yz$, once an example $i$ is classified correctly (i.e. $y_i z_i > 0$), the training objective can be optimized further by increasing the margin $y_i z_i$. Because $\ell''$ is also decreasing in $yz$, if the margin $y_i z_i$ increases, the term $\ell''(z_i; y_i)$ drops. Near the end of training, once most examples are classified correctly, gradient descent can easily increase the margins of all these examples by simply scaling up the final layer weight matrix. This causes the $\ell''(z_i; y_i)$ to drop. Therefore, even though progressive sharpening still applies to $\nabla_\theta h(\mathbf{x}_i; \theta) \, \nabla_\theta h(\mathbf{x}_i; \theta)^T$, the decrease in the $\ell''(z_i; y_i)$'s pulls down the leading eigenvalue of the Gauss-Newton matrix in equation 3.

This effect is illustrated in Figure 10. Here, we train a network on the binary classification task of CIFAR-10 airplane vs. automobile, using the logistic loss. In Figure 10(e), we plot the margin $y_i z_i$ for 10 examples in the dataset. Notice that at the end of training, these margins all continually rise;

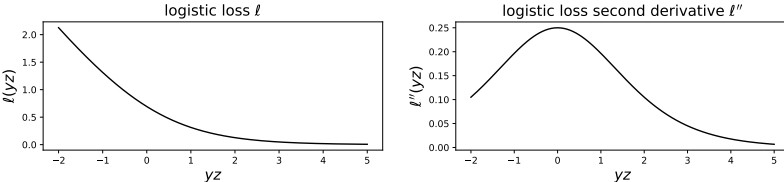

Figure 9: **Left**: logistic loss function $\ell$ as a function of $yz$. **Right**: its second derivative $\ell''$ as a function of $yz$.

this is because gradient descent "games" the objective by increasing the margins of successfully classified examples. When the margin $y_i z_i$ rises, the second derivative $\ell''(z_i; y_i)$ drops. This can be seen from Figure 10(f), where we plot $\ell''(z_i; y_i)$ for these same 10 examples. Now, all the while, the leading eigenvalue of the matrix $\frac{1}{n}\sum_{i=1}^{n}\nabla_\theta h(\mathbf{x}_i; \theta)\nabla_\theta h(\mathbf{x}_i; \theta)^T$ keeps rising, as can be seen from Figure 10(d). However, because the $\ell''s$ are dropping, the leading eigenvalue of the Gauss-Newton matrix $\frac{1}{n}\sum_{i=1}^{n}\ell''(z_i; y_i)\,\nabla_\theta h(\mathbf{x}_i; \theta)\nabla_\theta h(\mathbf{x}_i; \theta)^T$ starts to decrease at the end of training, as can be seen from the green line in Figure 10(c). Finally, since the leading eigenvalue of the Gauss-Newton matrix is an excellent approximation to the leading eigenvalue of the Hessian (i.e. the sharpness), the sharpness also drops at the end of training, as can be seen from the orange line in Figure 10(c).

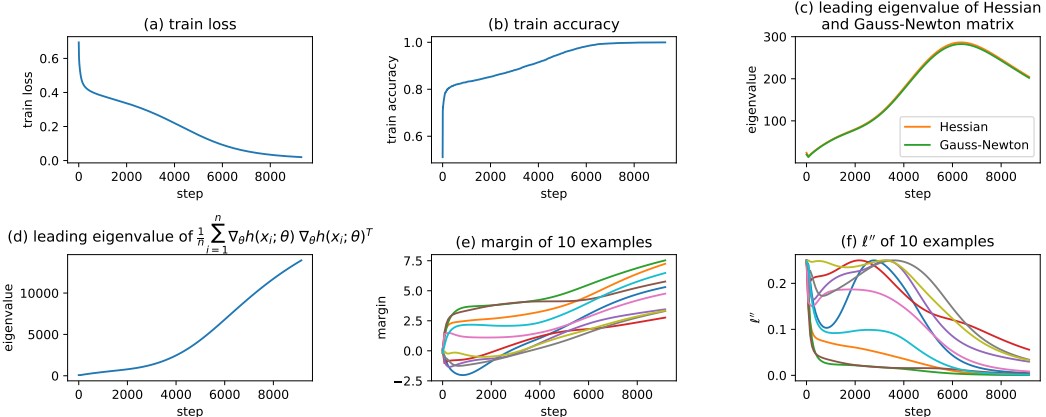

Figure 10: We train a network using the logistic loss on the binary classification problem of CIFAR-10 airplane vs. automobile. **(a)** The train loss. **(b)** The train accuracy. **(c)** The leading eigenvalue of the Hessian and of the Gauss-Newton matrix; observe that the latter is a great approximation to the former. **(d)** The leading eigenvalue of the matrix $\frac{1}{n}\sum_{i=1}^{n}\nabla_\theta h(\mathbf{x}_i; \theta)\nabla_\theta h(\mathbf{x}_i; \theta)^T$, which is the Gauss-Newton matrix without the $\ell''$ terms; observe that this value constantly rises — it does not dip at the end of training. **(e)** The margin $y_i z_i$ of 10 examples; observe that all the margins rise at the end of training. **(f)** the value $\ell''(z_i; y_i)$ for 10 examples; observe that all of these curves decline at the end of training.

**Multiclass classification with cross-entropy loss**   We consider a dataset $\{(\mathbf{x}_i, y_i)\}_{i=1}^{n} \subset \mathbb{R}^d \times \{1, \ldots, k\}$, where the examples are vectors in $\mathbb{R}^d$ and the labels are in $\{1, \ldots, k\}$. We consider a neural network $h : \mathbb{R}^d \times \mathbb{R}^p \to \mathbb{R}^k$ which maps an input $\mathbf{x} \in \mathbb{R}^d$ and a parameter vector $\theta \in \mathbb{R}^p$ to a prediction $h(x; \theta) \in \mathbb{R}^k$. Let $\ell : \mathbb{R}^k \times \{1, \ldots, k\} \to \mathbb{R}$ be the cross-entropy loss function:

$$\ell(\mathbf{z}; y) = -\log \frac{\exp(z_y)}{\sum_{j=1}^{k}\exp(z_j)}$$

$$= -\log p_y \quad \text{where} \quad \mathbf{p} = \frac{\exp(\mathbf{z})}{\sum_j \exp(z_j)}.$$

The Hessian $\nabla^2 \ell(\mathbf{z}; y) \in \mathbb{R}^{k \times k}$ of this loss function w.r.t the class scores $\mathbf{z}$ is:

$$\nabla^2 \ell(\mathbf{z}; y) = \mathrm{diag}(\mathbf{p}) - \mathbf{p}\mathbf{p}^T.$$

Now, for any loss function $\ell : \mathbb{R}^k \times \{1, \ldots, k\} \to \mathbb{R}$, we have the Gauss-Newton decomposition:

$$\nabla^2 f_i(\theta) = \mathbf{J}_i^T \left[ \nabla^2_{\mathbf{z}_i} \ell(\mathbf{z}_i; y_i) \right] \mathbf{J}_i + \sum_{j=1}^{k} [\nabla_{\mathbf{z}_i} \ell(\mathbf{z}_i; y_i)]_j \nabla^2_\theta h_j(\mathbf{x}_i; \theta)$$

where $\mathbf{z}_i = h_i(\mathbf{x}_i; \theta) \in \mathbb{R}^k$ are the logits for example $i$, $\mathbf{J}_i \in \mathbb{R}^{k \times p}$ is the network output-to-weights Jacobian for example $i$, $\nabla^2_{\mathbf{z}_i} \ell(\mathbf{z}_i; y_i) \in \mathbb{R}^{k \times k}$ is the Hessian of $\ell$ w.r.t its input $\mathbf{z}_i$, and $\nabla^2_\theta h_j(\mathbf{x}_i; \theta) \in \mathbb{R}^{p \times p}$ is the Hessian matrix of the $j$-th output of the network $h$ on the $i$-th example.

Dropping the second term yields the Gauss-Newton approximation:

$$\nabla^2 f_i(\theta) \approx \mathbf{J}_i^T \left[ \nabla^2_{\mathbf{z}_i} \ell(\mathbf{z}_i; y_i) \right] \mathbf{J}_i.$$

As in the binary classification case discussed above: at the end of training, for many examples $i$, the $y_i$ entry of $\mathbf{p}_i$ will tend toward 1 and the other entries of $\mathbf{p}_i$ will tend to 0. Once this occurs, the matrix $\mathrm{diag}(\mathbf{p}_i) - \mathbf{p}_i \mathbf{p}_i^T$ will broadly decrease in scale: the diagonal entries of this matrix are of the form $p(1-p)$, which goes to zero as $p \to 0$ or $p \to 1$; and the off-diagonal entries are of the form $-pq$, which also goes to zero if $p \to 0, q \to 1$ or $p \to 0, q \to 0$ or $p \to 1, q \to 0$.

This effect is illustrated in Figure 11. Here, we train a network on CIFAR-10 using the cross-entropy loss. In Figure 11(e), for ten examples $i$ in the dataset (with output scores $\mathbf{z}_i = h(\mathbf{x}_i) \in \mathbb{R}^k$), we plot the margin $\mathbf{z}_i[y_i] - \max_{j \neq y_i} \mathbf{z}_i[j]$, which is the difference between the score of the correct class $y_i$ and the score of the next-highest class. Observe that for all of these examples, this margin rises at the end of training. In Figure 11(f), for those same ten examples, we plot the quantity $\mathbf{p}_i[y_i](1 - \mathbf{p}_i[y_i])$. Observe that for all of these examples, this quantity continually decreases at the end of training. Now, all the while, the leading eigenvalue of the matrix $\frac{1}{n} \sum_{i=1}^{n} \mathbf{J}_i^T \mathbf{J}_i$ keeps rising, as can be seen from Figure 11(d). However, because $\nabla^2 \ell(\mathbf{z}_i; y_i)$ is decreasing, the leading eigenvalue of the Gauss-Newton matrix $\frac{1}{n} \sum_{i=1}^{n} \mathbf{J}_i^T \nabla^2 \ell(\mathbf{z}_i; y_i) \mathbf{J}_i$ starts to decrease at the end of training, as can be seen from the green line in Figure 11(c). Finally, since the leading eigenvalue of the Gauss-Newton matrix is an excellent approximation to the leading eigenvalue of the Hessian (i.e. the sharpness), the sharpness also drops at the end of training, as can be seen from the orange line in Figure 11(c).

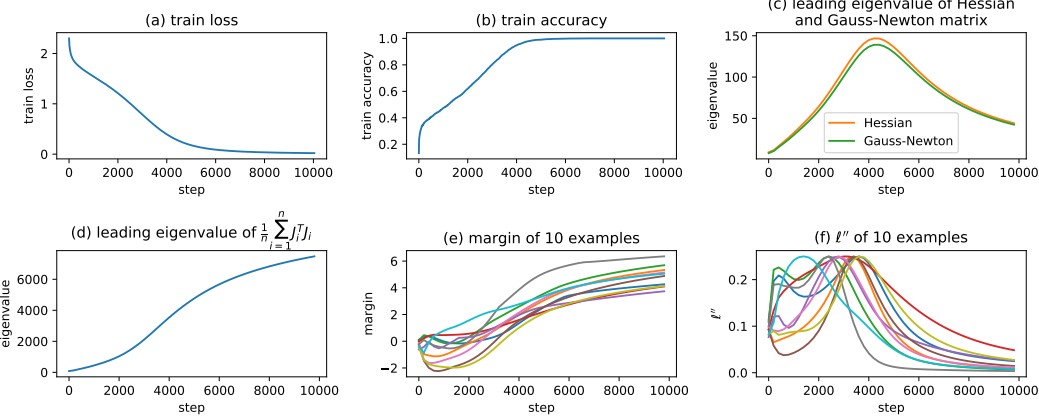

Figure 11: We train a network using the cross-entropy loss on CIFAR-10. **(a)** The train loss. **(b)** The train accuracy. **(c)** The leading eigenvalue of the Hessian and of the Gauss-Newton matrix; observe that the latter is a great approximation to the former. **(d)** The leading eigenvalue of the matrix $\frac{1}{n}\sum_{i=1}^{n}\mathbf{J}_i^T\mathbf{J}_i$, which is the Gauss-Newton matrix except the $\nabla^2\ell$ terms; observe that this value constantly rises — it does not dip at the end of training. **(e)** The margin $\mathbf{z}_i[y_i] - \max_{j\neq y_i}\mathbf{z}_i[j]$ of 10 examples; observe that all these margins rise at the end of training. **(f)** the value $\mathbf{p}_i[y_i](1 - \mathbf{p}_i[y_i])$ for 10 examples; observe that all of these curves decline at the end of training.

## D    EMPIRICAL STUDY OF PROGRESSIVE SHARPENING

In this appendix, we empirically study how problem parameters such as network width, network depth, and dataset size affect the degree to which progressive sharpening occurs. To study progressive sharpening on its own, without the confounding factor of instability, we train neural networks using gradient flow (Runge-Kutta) rather than gradient descent. Informally speaking, gradient flow does "what gradient descent would do if gradient descent didn't have to worry about instability."

We observe that progressive sharpening occurs to a greater degree: (1) for narrower networks than for wider networks (which is consistent with infinite-width NTK theory), (2) for deeper networks than for shallower networks, and (3) for larger datasets than for smaller datasets.

**The effect of width**    When networks parameterized in a certain way (the "NTK parameterization") are made infinitely wide and trained using gradient flow, the Hessian moves a vanishingly small amount during training, implying that no progressive sharpening occurs (Jacot et al., 2018; Lee et al., 2019; Jacot et al., 2020; Li & Liang, 2018). Therefore, a natural hypothesis is that progressive sharpening might attenuate as network width increases. We now run an experiment which supports this hypothesis.

We consider fully-connected architectures with two hidden layers and tanh activations, with widths $\{32, 64, 128, 256, 512, 1024\}$. We train on a size-5,000 subset of CIFAR-10 using the cross-entropy loss. We train using gradient flow (details in §I.5). We consider both NTK parameterization and standard parameterization (Lee et al., 2019).

In Figure 12, for each width, we train NTK-parameterized networks from five different random initializations, and plot the evolution of the sharpness during gradient flow. Observe that the maximum sharpness along the gradient flow trajectory is larger for narrow networks, and smaller for wide networks. (As elsewhere in this paper, note that the sharpness drops at the end of training due to the cross-entropy loss.) In Figure 13, we plot summary statistics from these training runs. Namely, define $\lambda_{\max}$ as the maximum sharpness over the gradient flow trajectory, and define $\lambda_0$ as the initial sharpness. In Figure 13(a), for each width we plot the mean and standard deviation of the maximum sharpness $\lambda_{\max}$ over the five different random initializations. Observe that $\lambda_{\max}$ becomes smaller, on average, as the width is made larger. In Figure 13(b), for each width we plot the mean and standard deviation of the *maximum sharpness gain* $\lambda_{\max}/\lambda_0$ over the five different random initializations. Observe that the maximum sharpness gain $\lambda_{\max}/\lambda_0$ also becomes smaller as the width is made larger. NTK theory suggests that $\lambda_{\max}/\lambda_0$ should deterministically tend to 1 as the width $\to \infty$, and Figure 13(b) is consistent with this prediction.

In Figures 14 and 15, we conduct similar experiments, but with standard parameterization rather than NTK parameterization. Similar to NTK parameterization, we observe in Figure 14 that the sharpness rises more for narrow networks than for wide networks.

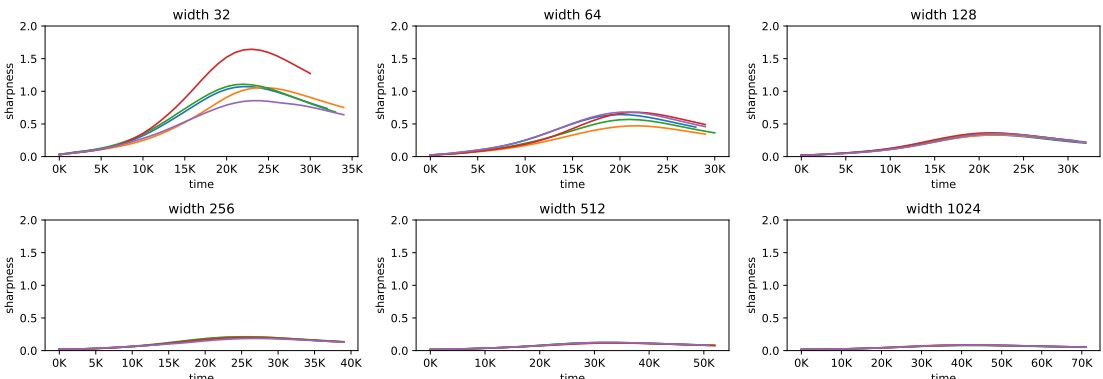

Figure 12: **NTK parameterization: evolution of the sharpness**. We use gradient flow to train NTK-parameterized networks, and we track the evolution of the sharpness during training. For each width, we train from five different random initializations (different colors). Observe that the sharpness rises more when training narrow networks than when training wide networks.

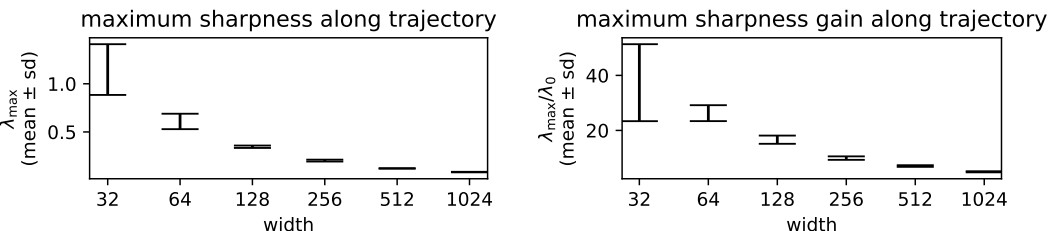

Figure 13: **NTK parameterization: summary statistics. Left**: For each network width, we plot the mean and standard deviation (over the five different random initializations) of the maximum sharpness $\lambda_{\max}$ along the gradient flow trajectory. Observe that $\lambda_{\max}$ decreases in expectation as the width increases. **Right**: For each network width, we plot the mean and standard deviation (over the five different random initializations) of the maximum sharpness gain $\lambda_{\max}/\lambda_0$ along the gradient flow trajectory. Observe that $\lambda_{\max}/\lambda_0$ decreases in expectation as the width increases. Indeed, NTK theory predicts that $\lambda_{\max}/\lambda_0$ should deterministically tend to 1 as width $\to \infty$ and this plot is consistent with that prediction.

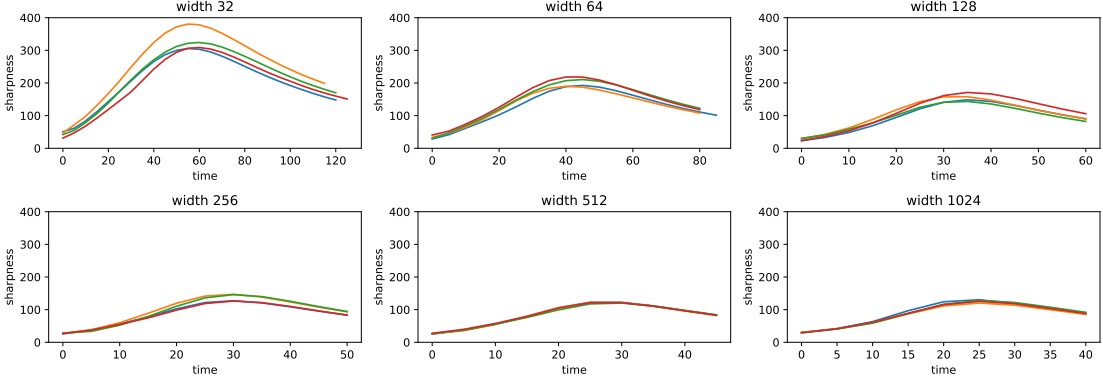

Figure 14: **Standard parameterization: evolution of the sharpness**. We use gradient flow to train standard-parameterized networks, and we track the evolution of the sharpness during training. For each width, we train from five different random initializations (different colors). Observe that the sharpness rises more when training narrow networks than when training wide networks.

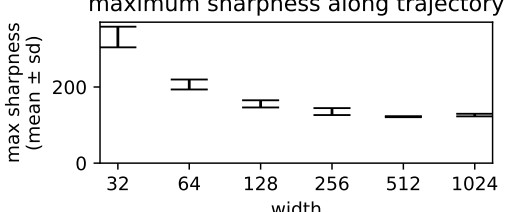 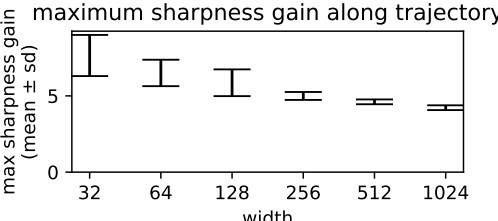

Figure 15: **Standard parameterization: summary statistics. Left**: For each network width, we plot the mean and standard deviation (over the five different random initializations) of the maximum sharpness $\lambda_{\max}$ along the gradient flow trajectory. Observe that $\lambda_{\max}$ tends to decrease in expectation as the width increases, though it is not clear whether this pattern still holds when moving from width 512 to width 1024 — more samples are needed. **Right**: For each network width, we plot the mean and standard deviation (over the five different random initializations) of the maximum sharpness gain $\lambda_{\max}/\lambda_0$ along the gradient flow trajectory. Observe that $\lambda_{\max}/\lambda_0$ decreases in expectation as the width increases. It is not clear whether or not $\lambda_{\max}/\lambda_0$ is deterministically tending to 1, but that does seem possible.

**Effect of depth**    We now explore the effect of network depth on progressive sharpening. We use gradient flow to train fully-connected tanh architectures of width 200 and varying depths — ranging from 1 hidden layer to 4 hidden layers. We train on a 5k subset of CIFAR-10 using both cross-entropy loss (in Figure 16) and square loss (in Figure 17). For each depth, we train from five different random initializations (different colors). Observe that progressive sharpening occurs to a greater degree as network depth increases.

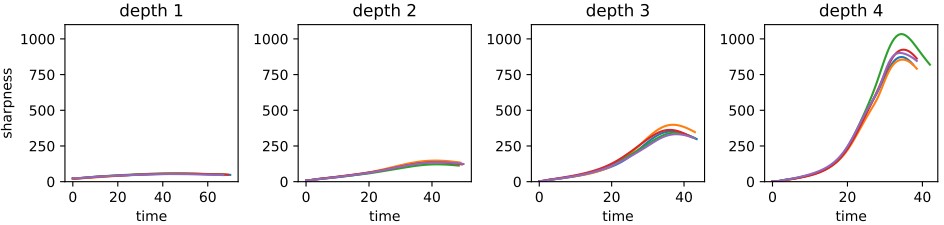

Figure 16: **The effect of depth: cross-entropy**. We use gradient flow to train networks of various depths, ranging from 1 hidden layer to 4 hidden layers, using cross-entropy loss. We train each network from five different random initializations (different colors). Observe that progressive sharpening occurs to a greater degree for deeper networks.

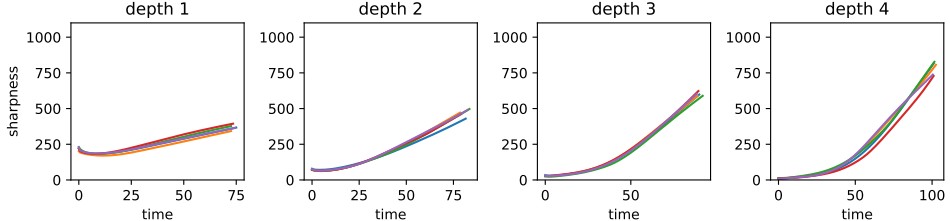

Figure 17: **The effect of depth: mean squared error**. We use gradient flow to train networks of various depths, ranging from 1 hidden layer to 4 hidden layers, using MSE loss. We train each network from five different random initializations (different colors). Observe that progressive sharpening occurs to a greater degree for deeper networks.

**Effect of dataset size**    We now explore the effect of dataset size on progressive sharpening. We use gradient flow to train a network on different-sized subsets of CIFAR-10. The network is a 2-hidden-layer, width-200 fully-connected tanh architecture, and we train using cross-entropy loss. The results are shown in Figure 18. Observe that progressive sharpening occurs to a greater degree as dataset size increases.

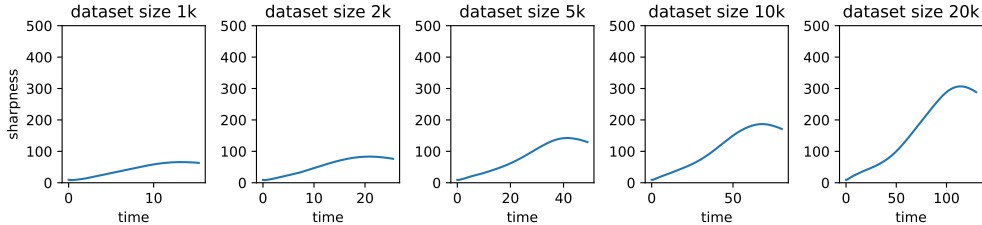

Figure 18: **The effect of dataset size**. We use gradient flow to train a network on varying-sized subsets of CIFAR-10. Observe that progressive sharpening occurs to a greater degree as the dataset size increases.

# E    SPEED OF DIVERGENCE ON QUADRATIC TAYLOR APPROXIMATION

If the sharpness at some iterate is strictly greater than $2/\eta$, then gradient descent with step size $\eta$ is guaranteed to diverge if run on the quadratic Taylor approximation around that iterate. However, the speed of this divergence could conceivably be slow — in particular, the train loss might continue to decrease for many iterations before it starts to increase. In this appendix we empirically demonstrate, to the contrary, that at the Edge of Stability, gradient descent diverges *quickly* if, at some iterate, we start running gradient descent on the quadratic Taylor approximation around that iterate.

We consider the fully-connected tanh network from section 3, trained on a 5,000-sized subsample of CIFAR-10 using both cross-entropy loss and MSE loss. At some timestep $t_0$ during training, we suddenly switch from running gradient descent on the real neural training objective, to running gradient descent on the quadratic Taylor approximation around the iterate at step $t_0$. We do this for three timesteps before gradient descent has entered the Edge of Stability, and three afterwards. Figure 20 shows the results for cross-entropy loss, and Figure 21 shows the (similar) results for MSE loss. Before entering the Edge of Stability (**top row**), gradient descent on the quadratic Taylor approximation behaves similar to gradient descent on the real neural training objective — that is, the orange line almost overlaps the blue line. Yet after entering the Edge of Stability (**bottom row**), gradient descent on the quadratic Taylor approximation quickly diverges, whereas gradient descent on the real neural training objective makes consistent (if choppy) progress.

In short, when gradient descent is *not* at the Edge of Stability, the quadratic Taylor approximation serves as a good model for the local progress of gradient descent. But when gradient descent *is* at the Edge of Stability, the quadratic Taylor approximation is an extremely poor model for the local progress of gradient descent. It is conceivable that there exists some simple modification to the quadratic Taylor model which would fix this issue (e.g. perhaps if one ignores a certain direction, the quadratic Taylor model is accurate). Nevertheless, unless/until such a fix is discovered, it is unclear why quadratic Taylor approximations should yield any insight into the local behavior of gradient descent.

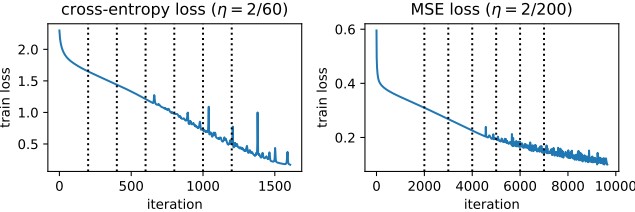

Figure 19: We train a neural network using cross-entropy loss (left) and MSE loss (right). In Figure 20 and 21, we show what happens when, at the iterations marked above by vertical dotted lines, we switch from gradient descent on the real neural training objective to gradient descent on the quadratic Taylor approximation.

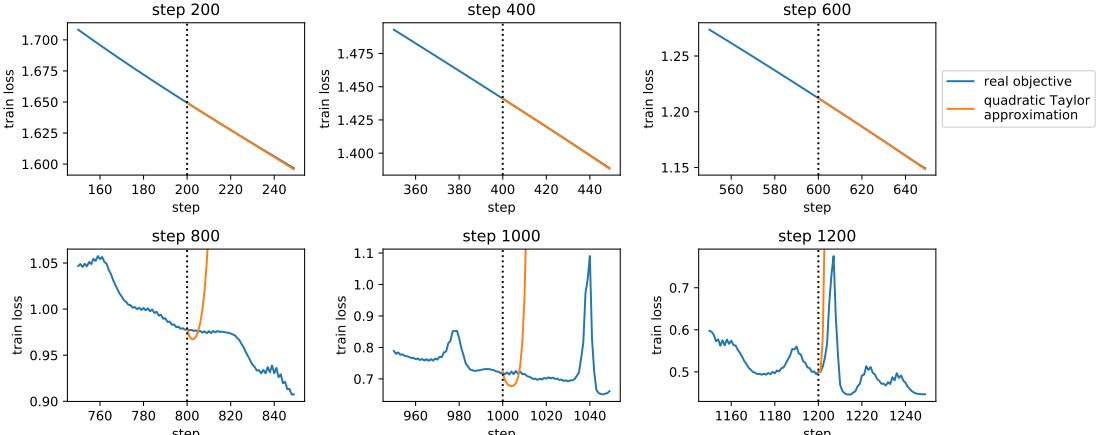

Figure 20: **Cross-entropy loss** ($\eta = 2/60$). At six different iterations during the training of the network from Figure 19 (marked by the vertical dotted black lines), we switch from running gradient descent on the real neural training objective (for which the train loss is plotted in blue) to running gradient descent on the quadratic Taylor approximation around the current iterate (for which the train loss is plotted in orange). **Top row** are timesteps {200, 400, 600} before gradient descent has entered the Edge of Stability; observe that the orange line (Taylor approximation) closely tracks the blue line (real objective). **Bottom row** are timesteps {800, 1000, 1200} during the Edge of Stability; observe that the orange line quickly diverges, whereas the blue line does not.

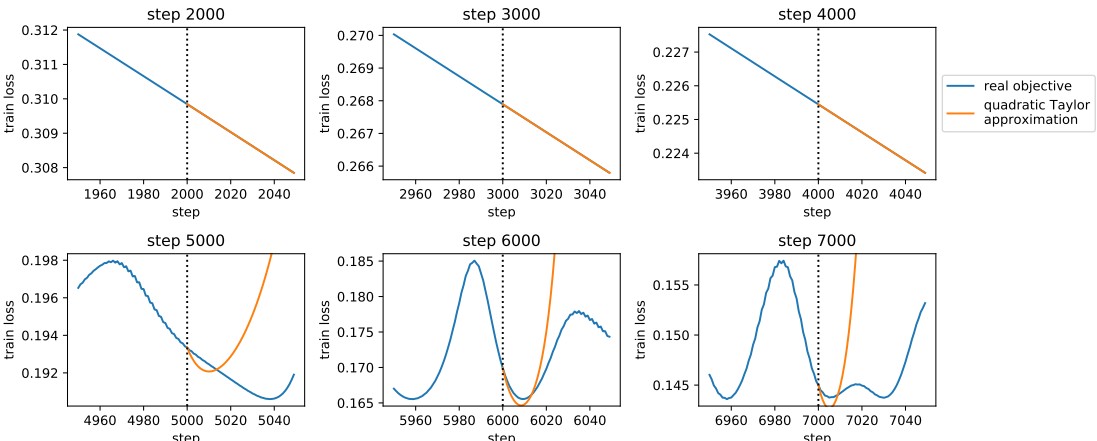

Figure 21: **MSE loss** ($\eta = 2/200$). At six different iterations during the training of the network from Figure 19 (marked by the vertical dotted black lines), we switch from running gradient descent on the real neural training objective (for which the train loss is plotted in blue) to running gradient descent on the quadratic Taylor approximation around the current iterate (for which the train loss is plotted in orange). **Top row** are timesteps (2000, 3000, 4000) before gradient descent has entered the Edge of Stability; observe that the orange line (Taylor approximation) closely tracks the blue line (real objective). **Bottom row** are timesteps (5000, 6000, 7000) during the Edge of Stability; observe that the orange line quickly diverges, whereas the blue line does not.

## F  "OPTIMAL" STEP SIZE SELECTION

One heuristic for setting the step size of gradient descent is to set the step size at iteration $t$ to $\eta_t = 1/\lambda_t$, where $\lambda_t$ is the sharpness at iteration $t$. While this heuristic is computationally impractical due to the time required to compute the sharpness at each iteration, it is often regarded as an ideal, for instance in LeCun et al. (1998) (Eq. 39), LeCun et al. (1993), and Schaul et al. (2013) (Eq 8). The motivation for this heuristic is: if all that is known about the training objective is that the local sharpness is $\lambda$, then a step size of $1/\lambda$ maximizes the guaranteed decrease in the training objective that would result from taking a step.

First, we demonstrate (on a single numerical example) that the dynamic step size $\eta_t = 1/\lambda_t$ is outperformed by the baseline approach of gradient descent with a fixed step size $\eta_t = 1/\lambda_0$, where $\lambda_0$ is the sharpness at initialization. In Figure 22, we train the network from §3 using both the dynamic $\eta = 1/\lambda_t$ step size heuristic as well as the baseline fixed step size of $\eta = 1/\lambda_0$. Observe that the $\eta = 1/\lambda_0$ baseline outperforms the $1/\lambda_t$ heuristic. Intuitively, because of progressive sharpening, the $\eta_t = 1/\lambda_t$ heuristic anneals the step size, and therefore ends up taking steps that are suboptimally small. In contrast, while the $\eta_t = 1/\lambda_0$ baseline quickly becomes unstable, this instability is apparently a worthwhile "price to pay" in return for the benefit of taking larger steps.

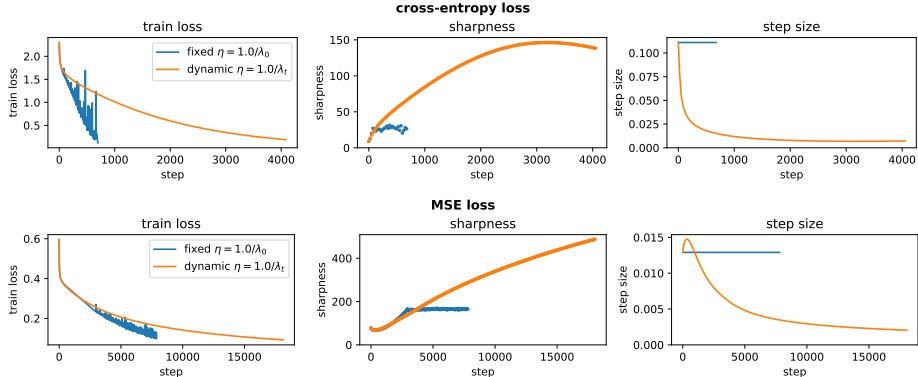

Figure 22: **A dynamic step size $\eta_t = 1/\lambda_t$ underperforms a fixed step size of $\eta_t = 1/\lambda_0$.**

Another natural idea is to dynamically set the step size at iteration $t$ to $\eta_t = 1.9/\lambda_t$. This step size rule takes larger steps than the $\eta = 1/\lambda_t$ rule while still remaining stable. In Figure 23, we compare this $\eta_t = 1.9/\lambda_t$ rule to a baseline approach of gradient descent with a fixed step size $\eta_t = 1.9/\lambda_0$, where $\lambda_0$ is the sharpness at initialization. Observe that the baseline of a fixed $1.9/\lambda_0$ step size outperforms the dynamic $\eta_t = 1.9/\lambda_t$ rule.

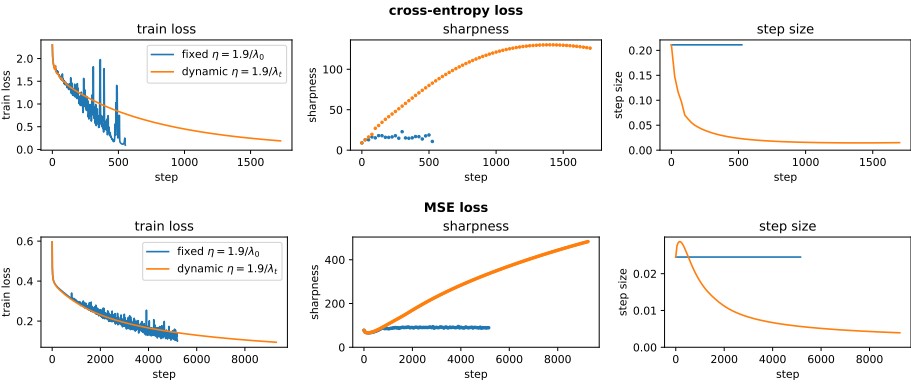

Figure 23: **A dynamic step size $\eta_t = 1.9/\lambda_t$ underperforms a fixed step size of $\eta_t = 1.9/\lambda_0$.**

## G EVOLUTION OF SHARPNESS DURING SGD

In this appendix, we briefly illustrate how the sharpness evolves during *stochastic* gradient descent. In Figure 24, we train the tanh network from §3 using SGD with both cross-entropy loss (top) and mean squared error (bottom). We train using a range of batch sizes (different colors). We observe the following:

1. During large-batch SGD, the sharpness behaves similar to full-batch gradient descent: it rises to $2/\eta$ (marked by the black horizontal dashed line) and then hovers just above that value.

2. Consistent with prior reports, we find that the smaller the batch size, the lower the sharpness (Keskar et al., 2016; Jastrzębski et al., 2017; 2019; 2020).

3. Notice that when training with cross-entropy loss at batch size 8 (the blue line), the sharpness *decreases* throughout most of training. The train accuracy (not pictured) is only 66% when the sharpness starts to decrease, which suggests that the cause of this decrease is unrelated to the effect described in Appendix C, whereby the sharpness decreases at the end of training. Figure 5(a) of Jastrzębski et al. (2020) also depicts a network where the sharpness decreases during SGD training.

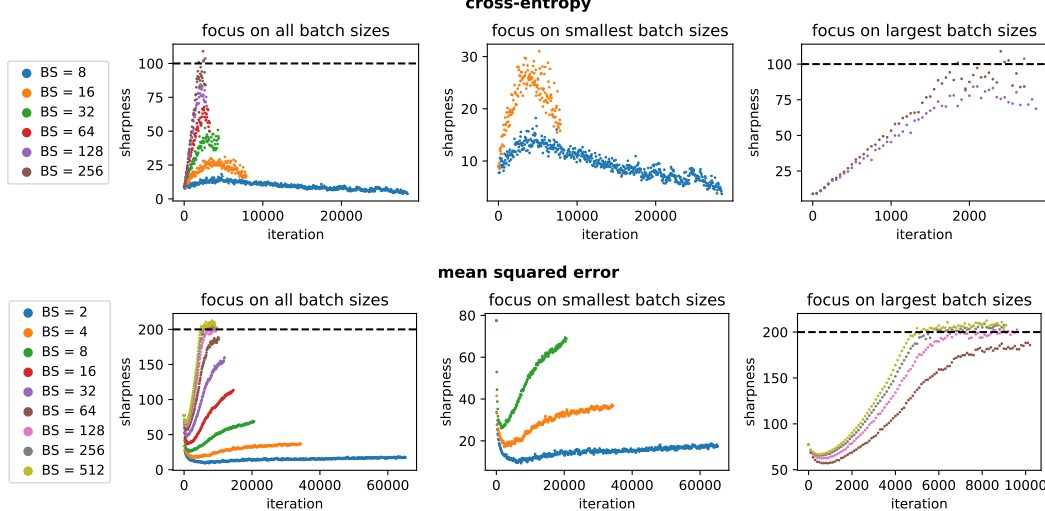

Figure 24: **Evolution of sharpness during SGD**. We train a network using SGD at various batch sizes, and plot the evolution of the sharpness. In the top row, we train with cross-entropy loss and step size $\eta = 0.02$; in the bottom row, we train with mean squared error and step size $\eta = 0.01$. The leftmost column shows all batch sizes; the center column focuses on just the smallest; and the rightmost column focuses on just the largest. The black dashed horizontal line marks $2/\eta$.

## H  SGD ACCLIMATES TO THE HYPERPARAMETERS

In this appendix, we conduct an experiment which suggests that some version of "Edge of Stability" may hold for SGD.

One way to interpret our main findings is that gradient descent "acclimates" to the step size in such a way that each training update sometimes increases, and sometimes decreases, the training loss, yet an update with a smaller step size would always decrease the training loss. We now demonstrate that this interpretation may generalize to SGD. In particular, we will demonstrate that SGD seems to "acclimate" to the step size and batch size in such a way that an actual update sometimes increases and sometimes decreases the loss in expectation, yet a update with a larger step size or smaller batch size would almost always *increase* the loss in expectation, and a step with a smaller step size or a larger batch size would almost always *decrease* the loss in expectation.

In Figure 25, we train the tanh network from §3 with MSE loss, using SGD with step size 0.01 and batch size 32. We periodically compute the training loss (over the full dataset) and plot these on the left pane of Figure 25. Observe that the training loss does not decrease monotonically, but of course this is not surprising — SGD is a random algorithm. However, what may be more surprising is that SGD is not even decreasing the training loss *in expectation*. On the right pane of Figure 25, every 500 steps during training, we use the Monte Carlo method to approximately compute the expected change in training loss that would result from an SGD step (the expectation here is over the randomness involved in selecting the minibatch). Observe that at many points during training, an SGD step would decrease the loss (as desired) in expectation, but at other points, and SGD step would increase the loss in expectation.

In Figure 26(a), while training that network, we compute the expected change in training loss that would result from taking an SGD step with the same step size used during training (i.e. 0.01), but *half the batch size* used during training (i.e. 16). We observe that an SGD step with half the batch size would consistently cause an *increase* in the training loss in expectation. In Figure 26(b) we repeat this experiment, but with *twice* the batch size used during training (i.e. 64). Notice that an SGD step with twice the batch size would consistently cause a *decrease* in the training loss in expectation. In Figure 26(c) and (d) repeat this experiment with the step size; we observe that an SGD step with a larger step size (0.02) would consistently increase the training loss in expectation, while an SGD step with a smaller step size (0.005) would consistently decrease the training loss in expectation.

In Figure 27, as a "control" experiment, we both train *and* measure the expected loss change under the following four hyperparameter settings: (step size 0.01, batch size 16), (step size 0.01, batch size 64), (step size 0.02, batch size 32), and (step size 0.005, batch size 32). In each case, we observe that, after a brief period at the beginning of training, each SGD update sometime increases and sometimes decreases the training loss in expectation.

Therefore, at least for this single network, we can conclude that no matter the hyperparameters, SGD quickly navigates to, and then lingers in, regions of the loss landscape in which an SGD update *with those hyperparameters* sometimes increases, and sometimes decreases the training loss in expectation, yet an SGD update with a smaller step size or larger batch size would consistently decrease the loss in expectation, and an SGD update with a larger step size or smaller batch size would consistently increase the loss in expectation.

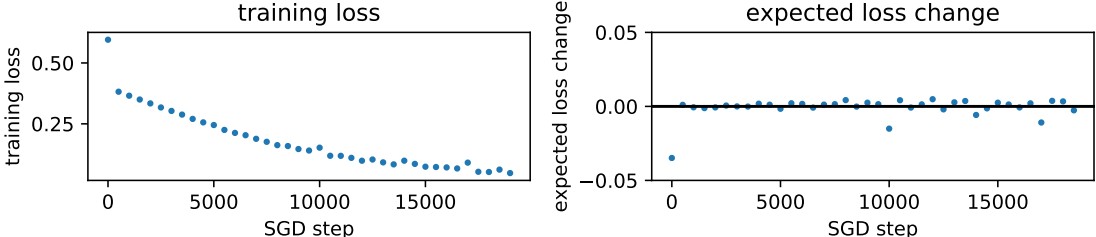

Figure 25: **SGD does not consistently decrease the training loss in expectation.** We train the tanh network from section 3 with MSE loss, using SGD with step size 0.01 and batch size 32. Periodically during training, we compute (left) the full-batch training loss, and (right) the expected change in the full-batch training loss that would result from taking an SGD step (where the expectation is over the randomness in sampling the minibatch). Strikingly, note that after the very beginning of training, the expected loss change is sometimes negative (as desired) but oftentimes positive. See Figure 26.

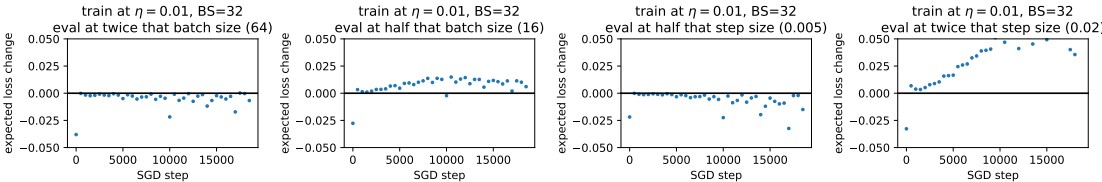

Figure 26: **An SGD step with a smaller learning rate or a larger batch size than the ones used during training** *would* **consistently decrease the loss in expectation.** At regular intervals during the training run depicted in Figure 25 (with $\eta = 0.01$ and batch size 32), we measure the expected change in the full-batch training loss that would result from an SGD step with a different step size or batch size. Observe that taking an SGD step with a smaller step size or a larger batch size would consistently have decreased the loss in expectation, while taking an SGD step with a larger step size or a smaller batch size would have consistently increased the loss in expectation.

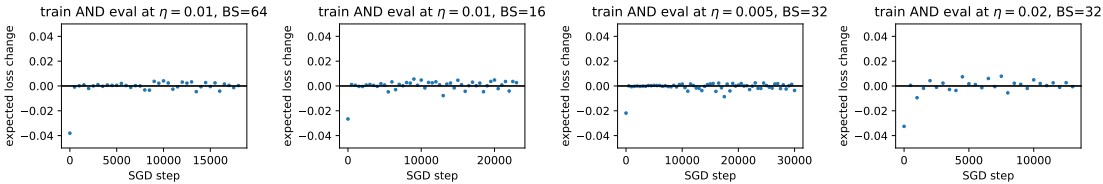

Figure 27: **Control experiment.** Above, in Figure 26(a), as we trained a network with step size 0.01 and batch size 32, we evaluated the expected change in training loss that would result from taking an SGD step with step size 0.01 and batch size 64. Here, in Figure 27(a), as a "control experiment," we train the network with step size 0.01 and batch size 64, and evaluate the expected change in training loss that would result from taking an SGD step with the same step size and batch size. We observe that an SGD step using the same step size and batch size that are used during training would sometimes increase and sometimes decrease the training loss in expectation. The other three panes are analogous.

## I  EXPERIMENTAL DETAILS

### I.1  VARYING ARCHITECTURES ON 5K SUBSET OF CIFAR-10

**Dataset.** The dataset consists of the first 5,000 examples from CIFAR-10. To preprocess the dataset, we subtracted the mean from each channel, and then divided each channel by the standard deviation (where both the mean and stddev were computed over the *full* CIFAR-10 dataset, not the 5k subset).

**Architectures.** We experimented with two architecture families: fully-connected and convolutional. For each of these two families, we experimented with several different activation functions, and for convolutional networks we experimented with both max pooling and average pooling.

The PyTorch code for e.g. the fully-connected ReLU network is as follows:

```
nn.Sequential(
    nn.Flatten(),
    nn.Linear(3072, 200, bias=True),
    nn.ReLU(),
    nn.Linear(200, 200, bias=True)
    nn.ReLU(),
    nn.Linear(200, 10, bias=True)
)
```

Networks with other activation functions would have `nn.ReLU()` replaced by `nn.ELU()`, `nn.Tanh()`, `nn.Softplus()`, or `nn.Hardtanh()`.

The PyTorch code for e.g. the convolutional ReLU network with max-pooling is as follows:

```
nn.Sequential(
    nn.Conv2d(3, 32, bias=True, kernel_size=3, padding=1),
    nn.ReLU(),
    nn.MaxPool2d(2),
    nn.Conv2d(32, 32, bias=True, kernel_size=3, padding=1),
    nn.ReLU(),
    nn.MaxPool2d(2),
    nn.Flatten(),
    nn.Linear(2048, 10, bias=True)
)
```

Networks with other activation functions would have `nn.ReLU()` replaced by `nn.ELU()` or `nn.Tanh()`, and networks with average pooling would have `nn.MaxPool2d(2)` replaced by `nn.AvgPool2d(2)`.

For all of these networks, we use the default PyTorch initialization. That is, both fully-connected layers and convolutional layers have the entries of their weight matrix and bias vector sampled i.i.d from $\text{Uniform}(-\frac{1}{\sqrt{\text{fan in}}}, \frac{1}{\sqrt{\text{fan in}}})$.

**Loss functions.** For a $k$-class classification problem, if the network outputs are $\mathbf{z} \in \mathbb{R}^k$ and the correct class is $i \in [k]$, then the mean squared error (MSE) loss is defined as $\frac{1}{2}\left[(z[i] - 1)^2 + \sum_{j \neq i} z[j]^2\right]$. That is, we encode the correct class with a "1" and the other classes with "0." The cross-entropy loss is defined as $-\log\left(\frac{\exp(z[i])}{\sum_{j \neq i} \exp(z[j])}\right)$.

### I.2  STANDARD ARCHITECTURES ON CIFAR-10

To preprocess the CIFAR-10 dataset, we subtracted the mean from each channel, and then divided each channel by the standard deviation.

Since training with full-batch gradient descent is slow, we opted to experiment on relatively shallow networks. The VGG networks (both with and without BN) are VGG-11's, from the implementation here: `https://github.com/chengyangfu/pytorch-vgg-cifar10/`

`blob/master/vgg.py`, with the dropout layers removed. The ResNet is the (non-fixup) ResNet-32 implemented here: `https://github.com/hongyi-zhang/Fixup`.

For the two networks with batch normalization, running gradient descent with full-dataset batch normalization would not have been feasible under our GPU memory constraints. Therefore, we instead used ghost batch normalization (Hoffer et al., 2017) with 50 ghost batches of size 1,000. This means that we divided the 50,000 examples in CIFAR-10 into 50 fixed groups of size 1,000 each, and defined the overall objective function to be the average of 50 fixed batch-wise objectives. To correctly compute the overall gradient of this training objective, we can just run backprop 50 times (once on each group) and average the resulting gradients.

To compute the sharpness over the full CIFAR-10 dataset would have been computationally expensive. Therefore, in an approximation, we instead computed the sharpness over just the first 5,000 examples in the dataset (or, for the BN networks, over the first 5 batches out of 50).

## I.3 BATCH NORMALIZATION EXPERIMENTS

We used the CNN architecture from §J (described above in §I.1), but with a `BatchNorm2d()` layer inserted after each activation layer.

Since our GPUs did not have enough memory to run batch normalization with the full dataset of size 5,000, we used ghost batch normalization with five ghost batches of size 1,000 (see I.2 for details).

## I.4 TRANSFORMER ON WIKITEXT-2

We used both the Transformer architecture and the preprocessing setup from the official Py-Torch (Paszke et al., 2019) word-level language modeling tutorial: `https://github.com/pytorch/examples/tree/master/word_language_model`. We used the settings `ninp=200, nhead=2, nhid=200, nlayers=2, dropout=0`. We set `bptt = 35`, which means that we divided the corpus into chunks of 35 tokens, and trained the network, using the negative log likelihood loss, to predict each token from the preceding tokens in the same chunk. Since computing the sharpness over the full dataset would not have been computationally practical, we computed the sharpness over a subset comprising 2500 training examples.

## I.5 RUNGE-KUTTA

We used the "RK4" fourth-order Runge-Kutta algorithm (Press et al., 1992) to numerically integrate the gradient flow ODE. The Runge-Kutta algorithm requires a step size. Rather than use a sophisticated algorithm for adaptive step size control, we decided to take advantage of the fact that we were already periodically computing the sharpness: at each step, we set the step size to $\alpha/\lambda$, where $\alpha$ is a tunable parameter and $\lambda$ is the most recent value for the sharpness. We set $\alpha = 1$ or $\alpha = 0.5$.

## I.6 RANDOM PROJECTIONS

In order to ascertain whether gradient descent at step size $\eta$ initially followed the gradient flow trajectory (and, if so, for how long), we monitored the $\ell_2$ distance in weight space between the gradient flow solution at time $t$ and the gradient descent iterate at step $t/\eta$. One way to do would be as follows: (a) when running gradient flow, save the weights after every $\Delta t$ units of time, for some parameter $\Delta t$; (b) when running gradient descent, save the weights at each $(\frac{\Delta t}{\eta})$-th step; (c) plot the difference between these two sequences. (Note that this approach requires $\Delta t$ to be divisible by $\eta$.)

We essentially used this approach, but with one modification: regularly saving the entire network weight vector would have consumed a large amount of disk space, so we instead saved low-dimensional random projections of the network weights. To be clear, let $d$ be the number of network weights, and let $k$ be the number of random projections (a tunable parameter chosen such that $k \ll d$). Then we first generated a matrix $\mathbf{M} \in \mathbb{R}^{k \times d}$ by sampling each entry i.i.d from the standard normal distribution. During training, rather than periodically save the whole weight vector (a $d$-dimensional vector), we premultiplied this vector by the matrix $\mathbf{M}$ to obtain a $k$-dimensional vector, and we periodically saved these vectors instead. Then we plotted the $\ell_2$ distance between the low-dimensional vectors from gradient flow, and the low-dimensional vectors from gradient descent.

## J    EXPERIMENTS: VARY ARCHITECTURES

In this appendix, we fix the task as that of fitting a 5,000-sized subset of CIFAR-10, and we verify our main findings across a broad range of architectures.

**Procedure**    We consider fully-connected networks and convolutional networks, the latter with both max-pooling and average pooling. For all of these, we consider tanh, ReLU, and ELU activations, and for fully-connected networks we moreover consider softplus and hardtanh activations. We train each network with both cross-entropy and MSE loss. See §I.1 for full experimental details.

In each case, we first use the Runge-Kutta method to numerically integrate the gradient flow ODE (see §I.5 for details). For architectures that give rise to continuously differentiable training objectives, the gradient flow ODE is guaranteed to have a unique solution (which we call the gradient flow trajectory), and Runge-Kutta will return a numerical approximation to this solution. On the other hand, for architectures with ReLU, hardtanh, or max-pooling, the training objective is not continuously differentiable, so the gradient flow ODE does not necessarily have a unique solution, and there are no guarantees a priori on what Runge-Kutta will return (more on this below under the "findings" heading). Still, in both cases, since our implementation of Runge-Kutta automatically adjusts the step size based on the local sharpness in order to remain stable, the Runge-Kutta trajectory can be roughly viewed as "what gradient descent would do if instability was not an issue."

We then run gradient descent at a range of step sizes. These step sizes $\eta$ were chosen by hand so that the quantity $2/\eta$ would be spaced uniformly between $\lambda_0$ (the sharpness at initialization) and $\lambda_{\max}$ (the maximum sharpness along the Runge-Kutta trajectory).

**Results**    During Runge-Kutta, we observe that the sharpness tends to continually increase during training (progressive sharpening), with the exception that when cross-entropy loss is used, the sharpness decreases at the very end of training, as explained in Appendix C.

During gradient descent with step size $\eta$, we observe that once the sharpness reaches $2/\eta$, it ceases to increase much further, and instead hovers right at, or just above, the value $2/\eta$. For reasons unknown, it tends to be true that for MSE loss, the sharpness hovers just a tiny bit above the value $2/\eta$, while for cross-entropy loss the gap between the sharpness and the value $2/\eta$ is a bit larger.

For each step size, we monitor the distance between the gradient descent trajectory and the Runge-Kutta trajectory — that is, we monitor the distance between the Runge-Kutta iterate at time $t$ and the gradient descent iterate at step $t/\eta$ (see §I.6 for details). Empirically, for architectures that give rise to continuously differentiable training objectives, we observe that this distance is nearly zero before the sharpness hits $2/\eta$, and it starts to climb immediately afterwards. This means that gradient descent closely tracks the gradient flow trajectory so long as the sharpness remains less than $2/\eta$. Note that this finding was not a foregone conclusion: gradient descent is guaranteed to track the gradient flow trajectory in the limit of infinitesimal step sizes (since gradient descent is the forward Euler discretization of the gradient flow ODE), but for non-infinitesimal step sizes, there is discretization error, which is studied in Barrett & Dherin (2021). Our empirical finding is essentially that this discretization error is small compared to the difference between trajectories caused by instability.

On the other hand, for architectures with non-differentiable components such as ReLU or max-pooling, we sometimes observe that gradient descent tracks the Runge-Kutta trajectory so long as the sharpness remains less than $2/\eta$, but we also sometimes observe that the gradient descent trajectories differ from one another (and from Runge-Kutta) from the beginning of training. In the former case, we can infer that the gradient flow trajectory apparently does exist, and is returned by Runge-Kutta; in the latter case, we can infer that either (a) the gradient flow trajectory does not exist, or (b) that it does exist (and is returned by Runge-Kutta), but the step sizes we used for gradient descent were too large to track it.

## J.1 Fully-connected tanh network

### J.1.1 Square loss

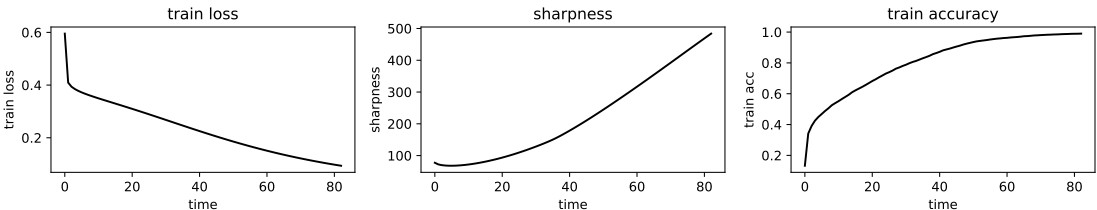

Figure 28: **Gradient flow.** We train the network to 99% accuracy with gradient flow, by using the Runge-Kutta method to discretize the gradient flow ODE (details in §I.5). We plot the train loss (left), sharpness (center), and train accuracy (right) over time. Observe that the sharpness tends to continually increase (except for a slight decrease at initialization).

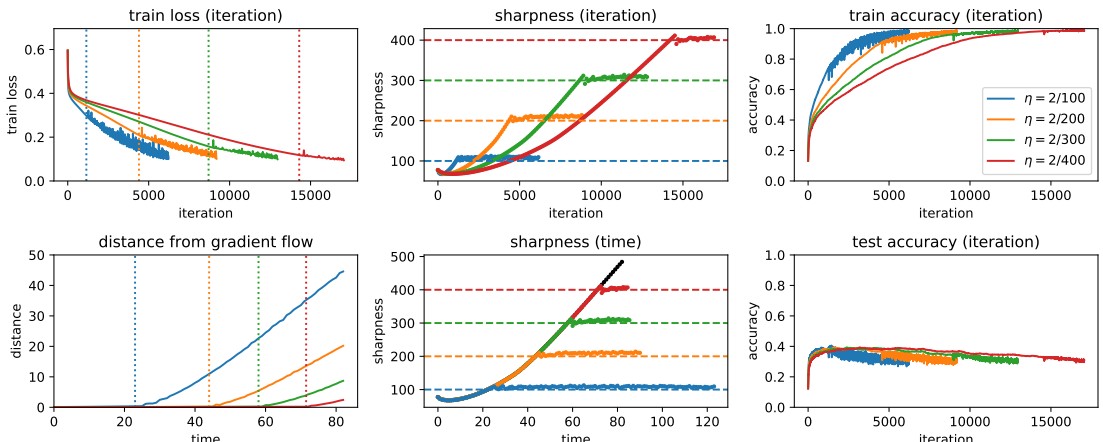

Figure 29: **Gradient descent.** We train the network to 99% accuracy using gradient descent at a range of step sizes $\eta$. **Top left:** we plot the train loss curves, with a vertical line marking the iteration where the sharpness first crosses $2/\eta$. Observe that the train loss monotonically decreases before this point, but behaves non-monotonically afterwards. **Top middle:** we plot the sharpness (measured at regular intervals during training). For each step size, the horizontal dashed line of the appropriate color marks the maximum stable sharpness $2/\eta$. Observe that the sharpness tends to increase during training until reaching the value $2/\eta$, and then hovers just a bit above that value. **Bottom left**: we track the $\ell_2$ distance between (random projections of) the gradient *flow* iterate at time $t$ and the gradient *descent* iterate at iteration $t/\eta$ (details in §I.6). For each step size $\eta$, the vertical dotted line marks the time when the sharpness first crosses $2/\eta$. Observe that the distance is essentially zero until this time, and starts to grow afterwards. From this, we can conclude that gradient descent closely tracks the gradient flow trajectory (moving at a speed $\eta$) until the point on that trajectory where the sharpness reaches $2/\eta$. **Bottom middle**: to further visualize the previous point, we plot the evolution of sharpness during gradient descent, but with time (= iteration × step size) on the x-axis rather than iteration. We plot the gradient *flow* sharpness in black. Observe that the gradient descent sharpness matches the gradient flow sharpness until reaching the value $2/\eta$.

### J.1.2 Cross-entropy loss

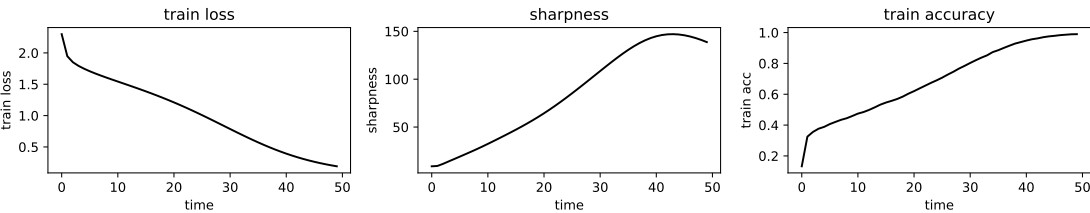

Figure 30: **Gradient flow.** Refer to the Figure 28 caption for more information. Additionally, in this figure the sharpness drops at the end of training due to the cross-entropy loss.

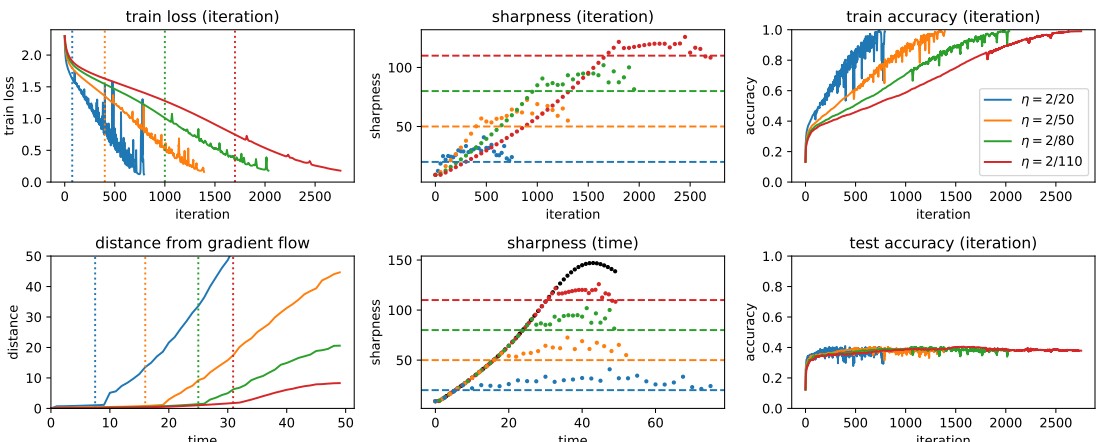

Figure 31: **Gradient descent.** Refer to the Figure 29 caption for more information.

## J.2 FULLY-CONNECTED ELU NETWORK

### J.2.1 SQUARE LOSS

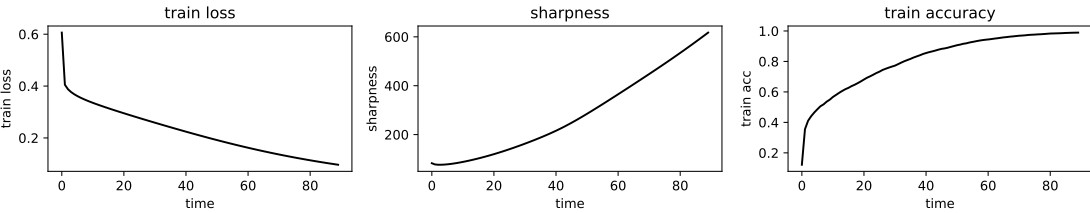

Figure 32: **Gradient flow.** Refer to the Figure 28 caption for more information.

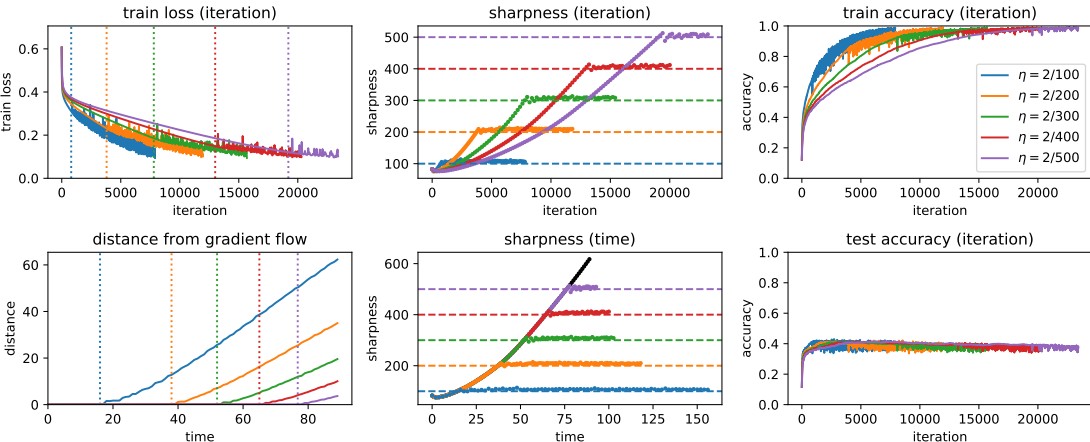

Figure 33: **Gradient descent.** Refer to the Figure 29 caption for more information.

### J.2.2 CROSS-ENTROPY LOSS

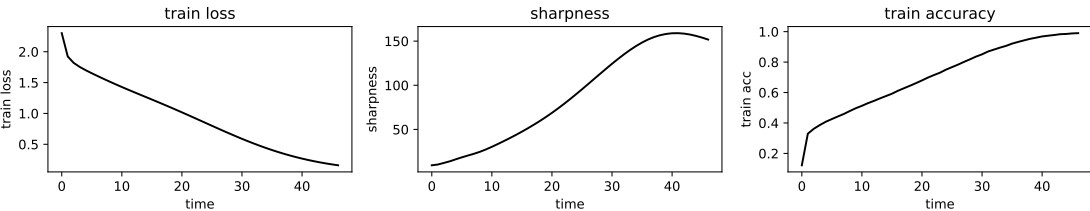

Figure 34: **Gradient flow.** Refer to the Figure 28 caption for more information. Additionally, in this figure the sharpness drops at the end of training due to the cross-entropy loss.

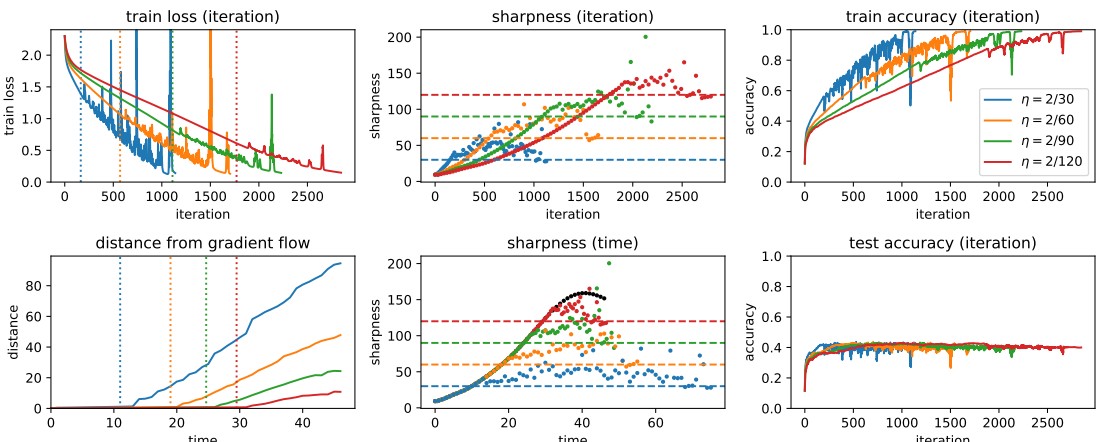

Figure 35: **Gradient descent.** Refer to the Figure 29 caption for more information.

### J.3 FULLY-CONNECTED SOFTPLUS NETWORK

### J.3.1 SQUARE LOSS

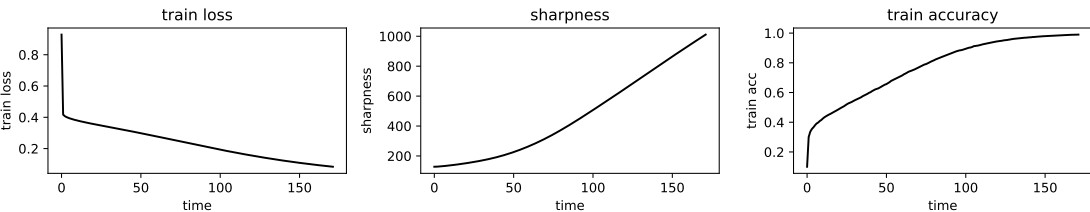

Figure 36: **Gradient flow.** Refer to the Figure 28 caption for more information.

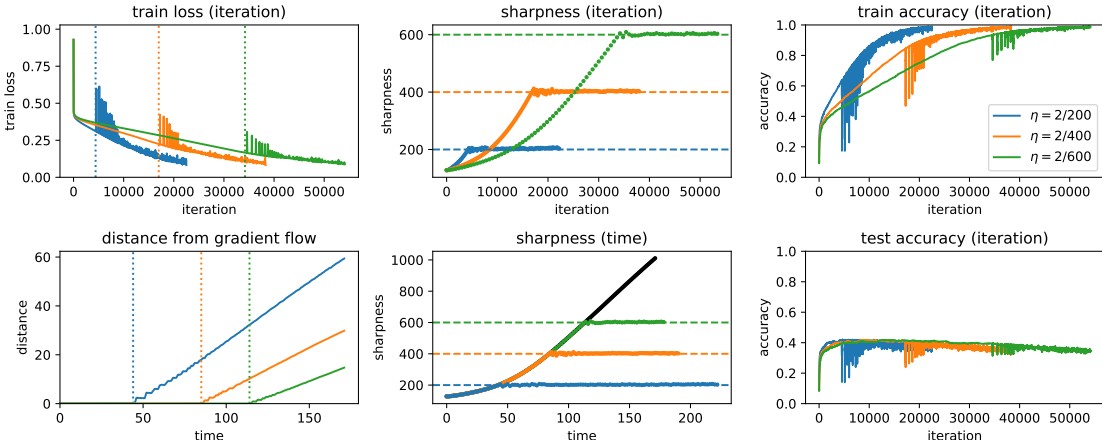

Figure 37: **Gradient descent.** Refer to the Figure 29 caption for more information.

### J.3.2 CROSS-ENTROPY LOSS

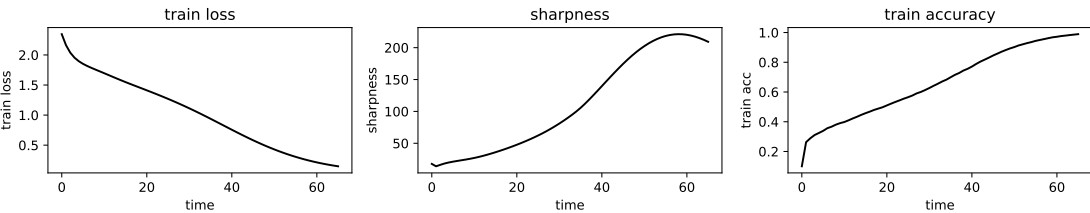

Figure 38: **Gradient flow.** Refer to the Figure 28 caption for more information. Additionally, in this figure the sharpness drops at the end of training due to the cross-entropy loss.

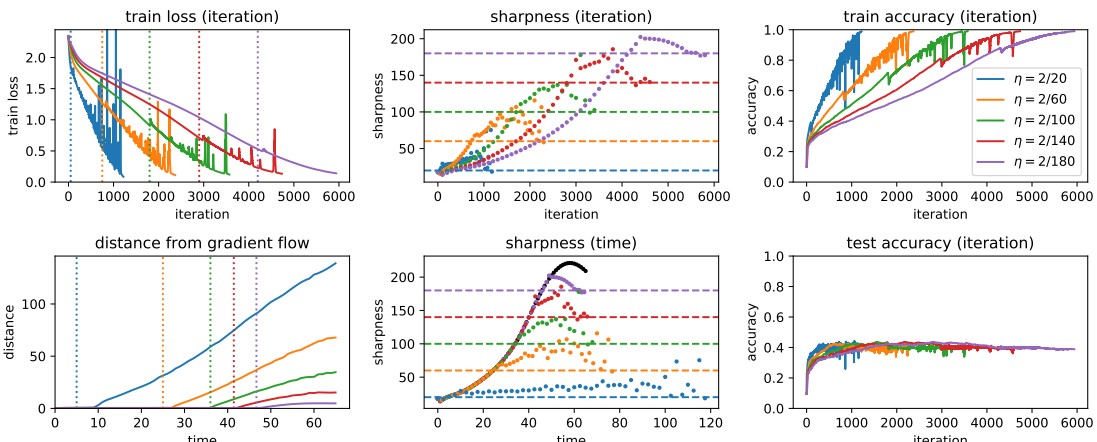

Figure 39: **Gradient descent.** Refer to the Figure 29 caption for more information.

### J.4 FULLY-CONNECTED RELU NETWORK

Note that since the ReLU activation function is not continuously differentiable, the training objective is not continuously differentiable, and so a unique gradient flow trajectory is not guaranteed to exist.

### J.4.1 SQUARE LOSS

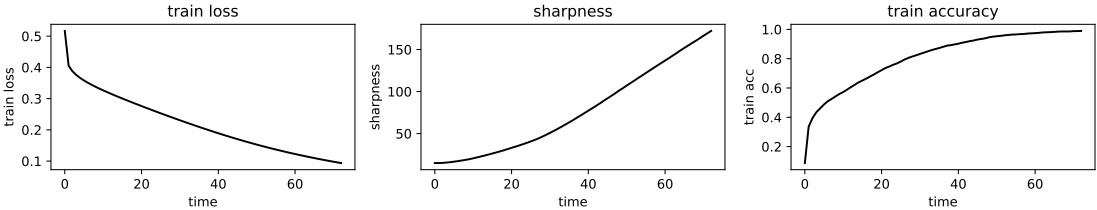

Figure 40: **Runge-Kutta.** We train the network to 99% accuracy using the Runge-Kutta algorithm. (Since a unique gradient flow trajectory is not guaranteed to exist, we hesitate to call this "gradient flow"; Runge-Kutta should essentially be viewed as gradient descent with a very small step size.) Observe that the sharpness tends to continually increase.

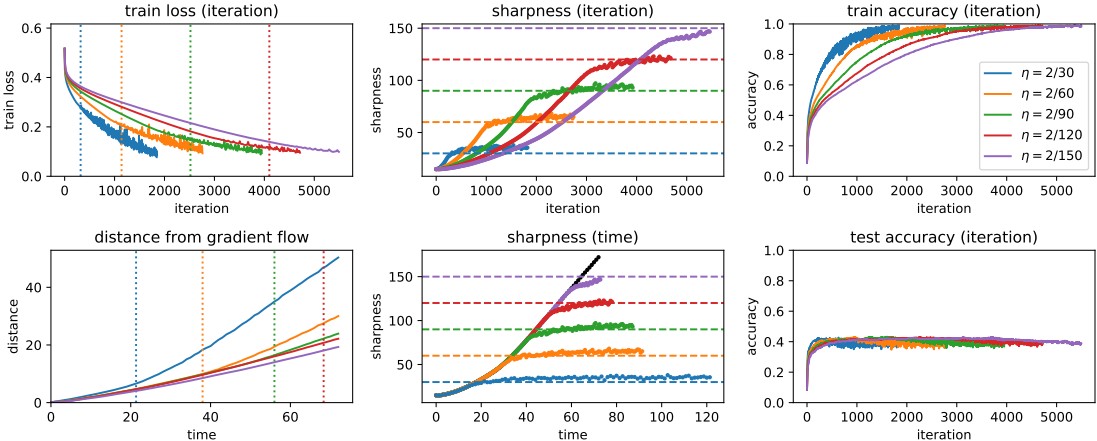

Figure 41: **Gradient descent. All panes except bottom left**: refer to the Figure 29 caption for more information. **Bottom left**: We track the $\ell_2$ distance between (random projections of) the Runge-Kutta iterate at time $t$ and the gradient descent iterate at iteration $t/\eta$. For each step size $\eta$, the vertical dotted loss marks the time when the sharpness first crosses $2/\eta$. In contrast to Figure 29, here the distance between gradient descent and gradient flow starts to noticeably grow from the beginning of training. From this we conclude that for this architecture, gradient descent does *not* track the Runge-Kutta trajectory at first. Furthermore, for this architecture, observe that the training loss starts behaving non-monotonically (and the sharpness begins to plateau) *before* the sharpness hits $2/\eta$. As discussed in Appendix A ("Caveats"), we believe that this is also due to the fact that ReLU is non-differentiable.

### J.4.2 CROSS-ENTROPY LOSS

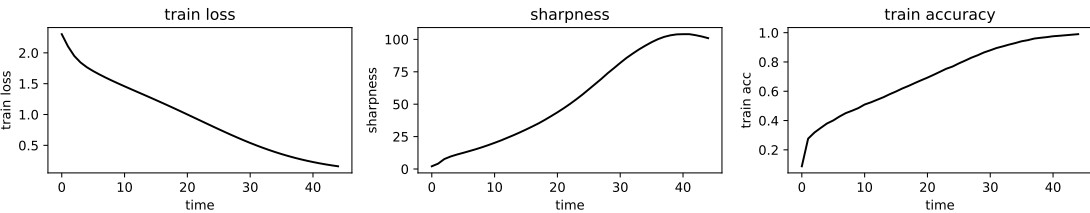

Figure 42: **Runge-Kutta.** Refer to the Figure 40 caption for more information. Additionally, in this figure the sharpness drops at the end of training due to the cross-entropy loss.

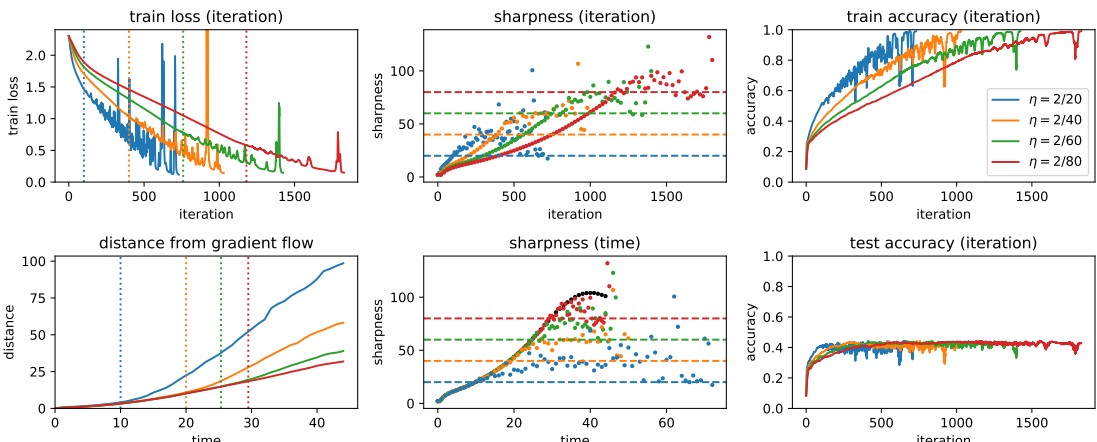

Figure 43: **Gradient descent.** Refer to the Figure 41 caption for more information.

### J.5 FULLY-CONNECTED HARD TANH NETWORK

Note that since the hardtanh function is not continuously differentiable, the training objective is not continuously differentiable, and so a unique gradient flow trajectory is not guaranteed to exist.

#### J.5.1 SQUARE LOSS

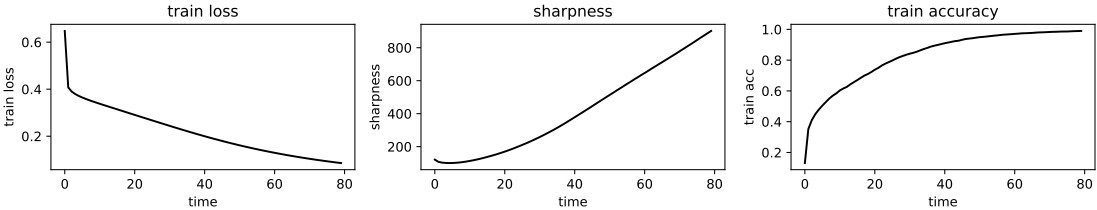

Figure 44: **Runge-Kutta.** Refer to the Figure 40 caption for more information.

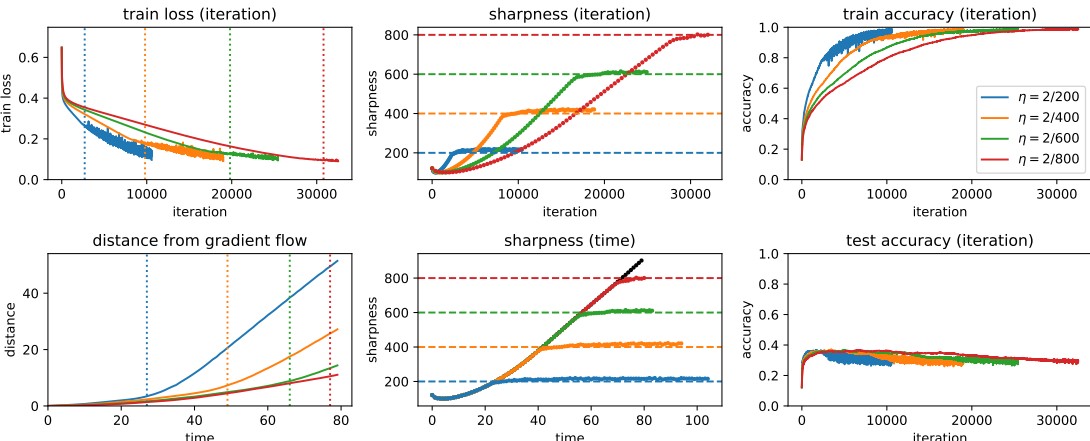

Figure 45: **Gradient descent.** Refer to the Figure 41 caption for more information.

### J.5.2 CROSS-ENTROPY LOSS

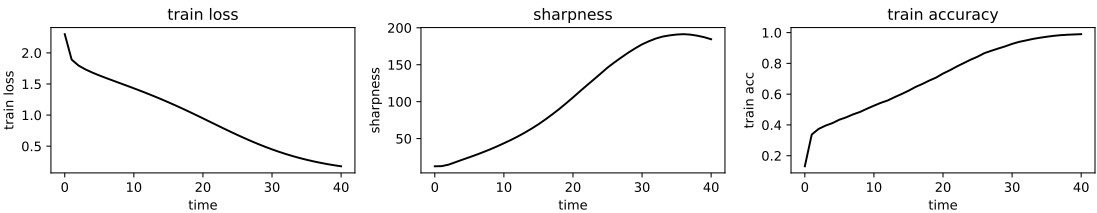

Figure 46: **Runge-Kutta.** Refer to the Figure 40 caption for more information. Additionally, in this figure the sharpness drops at the end of training due to the cross-entropy loss.

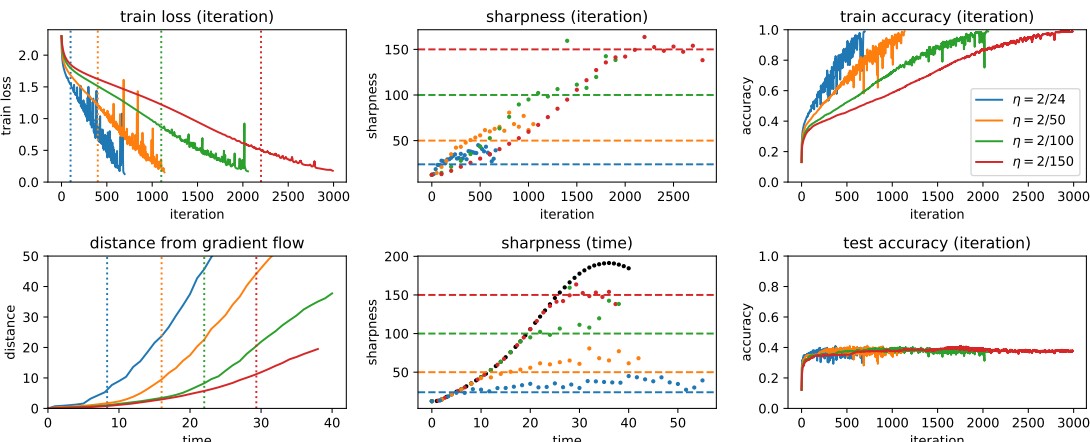

Figure 47: **Gradient descent.** Refer to the Figure 41 caption for more information.

### J.6 CONVOLUTIONAL TANH NETWORK WITH MAX POOLING

Note that since max-pooling is not continuously differentiable, the training objective is not continuously differentiable, and so a unique gradient flow trajectory is not guaranteed to exist.

#### J.6.1 SQUARE LOSS

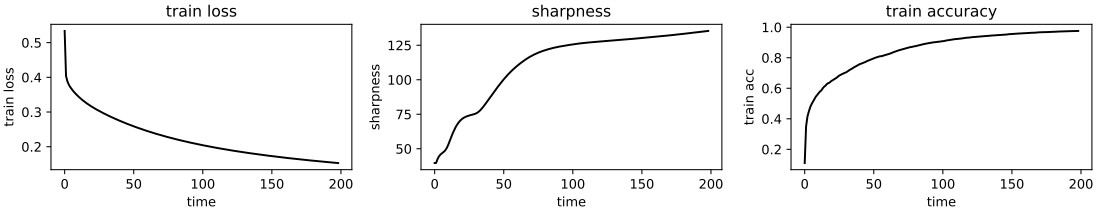

Figure 48: **Runge-Kutta.** We train the network to 99% accuracy using the Runge-Kutta algorithm. (Since a unique gradient flow trajectory is not guaranted to exist, we hesitate to call this "gradient flow.") Observe that the sharpness tends to continually increase.

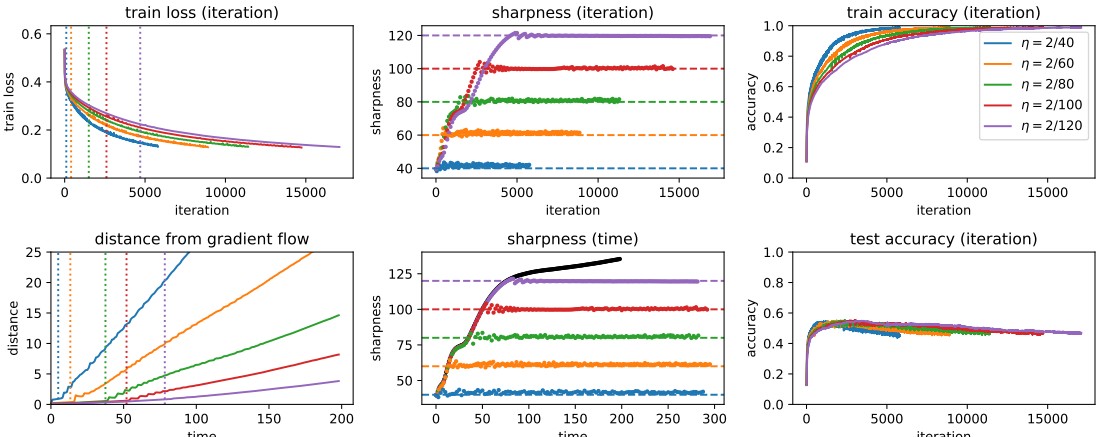

Figure 49: **All panes except bottom left**: refer to the Figure 29 caption for more information. **Bottom left**: We track the $\ell_2$ distance between (random projections of) the Runge-Kutta iterate at time $t$ and the gradient descent iterate at iteration $t/\eta$. For each step size $\eta$, the vertical dotted loss marks the time when the sharpness first crosses $2/\eta$. Observe that the distance is essentially zero until this time, and starts to grow afterwards. From this, we can conclude that even though the training objective is not differentiable, gradient descent does closely track the Runge-Kutta trajectory (moving at a speed $\eta$) until the point on that trajectory where the sharpness reaches $2/\eta$.

### J.6.2 CROSS-ENTROPY LOSS

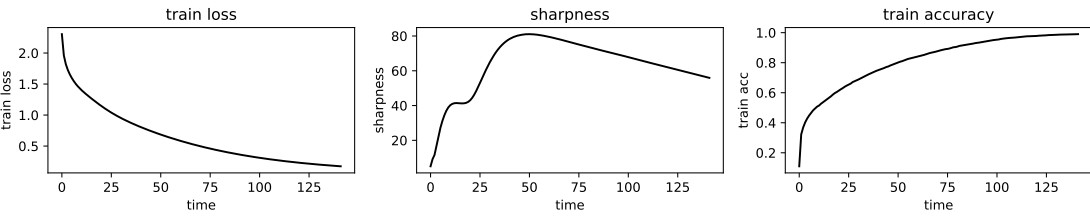

Figure 50: **Runge-Kutta.** Refer to the Figure 48 caption for more information. Additionally, in this figure the sharpness drops at the end of training due to the cross-entropy loss.

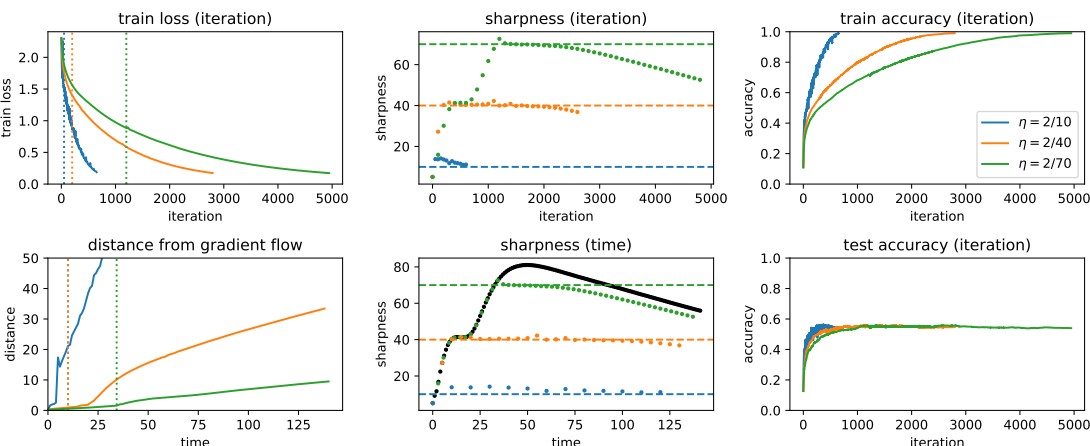

Figure 51: **Gradient descent.** Refer to the Figure 49 caption for more information.

## J.7 CONVOLUTIONAL ELU NETWORK WITH MAX POOLING

Note that since max-pooling is not continuously differentiable, the training objective is not continuously differentiable, and so a unique gradient flow trajectory is not guaranteed to exist.

### J.7.1 SQUARE LOSS

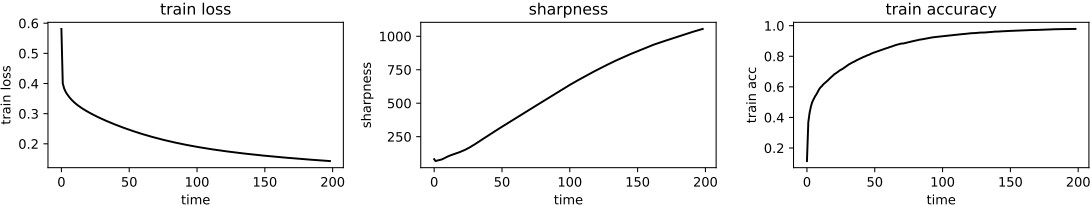

Figure 52: **Runge-Kutta.** Refer to the Figure 48 caption for more information.

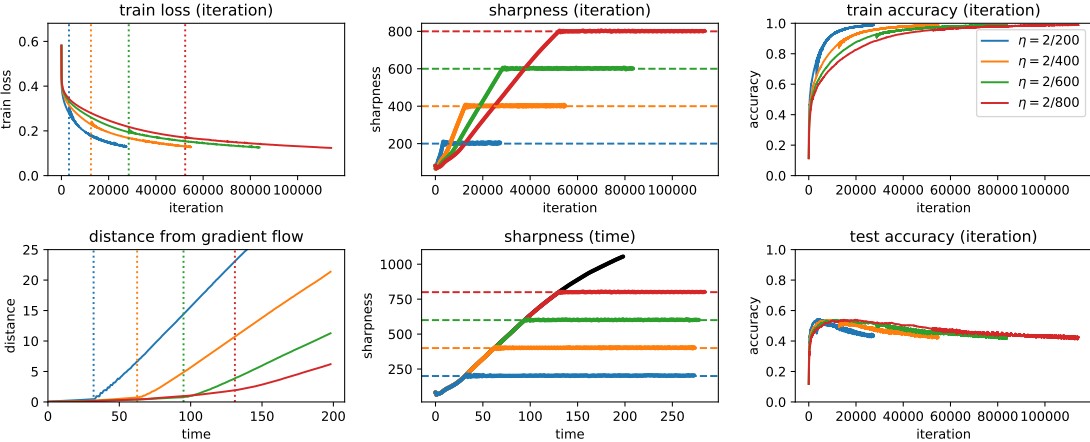

Figure 53: **Gradient descent.** Refer to the Figure 49 caption for more information.

### J.7.2 CROSS-ENTROPY LOSS

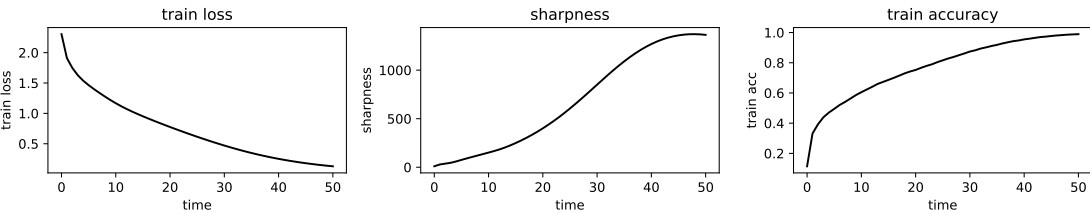

Figure 54: **Runge-Kutta.** Refer to the Figure 48 caption for more information. Additionally, in this figure the sharpness drops at the end of training due to the cross-entropy loss.

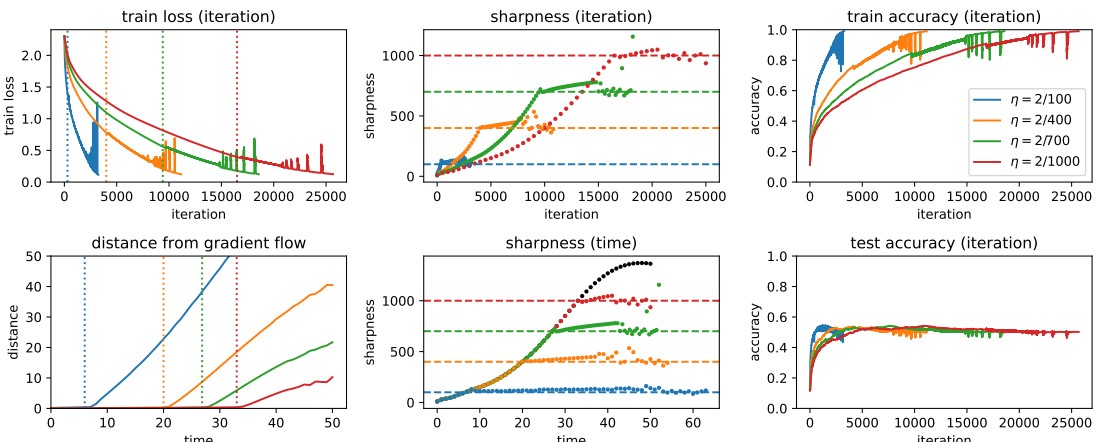

Figure 55: **Gradient descent.** Refer to the Figure 49 caption for more information.

## J.8    CONVOLUTIONAL RELU NETWORK WITH MAX POOLING

Note that since ReLU and max-pooling are not continuously differentiable, the training objective is not continuously differentiable, and so a unique gradient flow trajectory is not guaranteed to exist.

### J.8.1    SQUARE LOSS

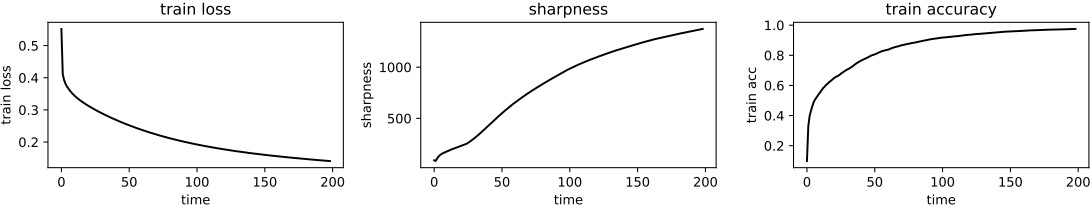

Figure 56: **Runge-Kutta.** Refer to the Figure 48 caption for more information.

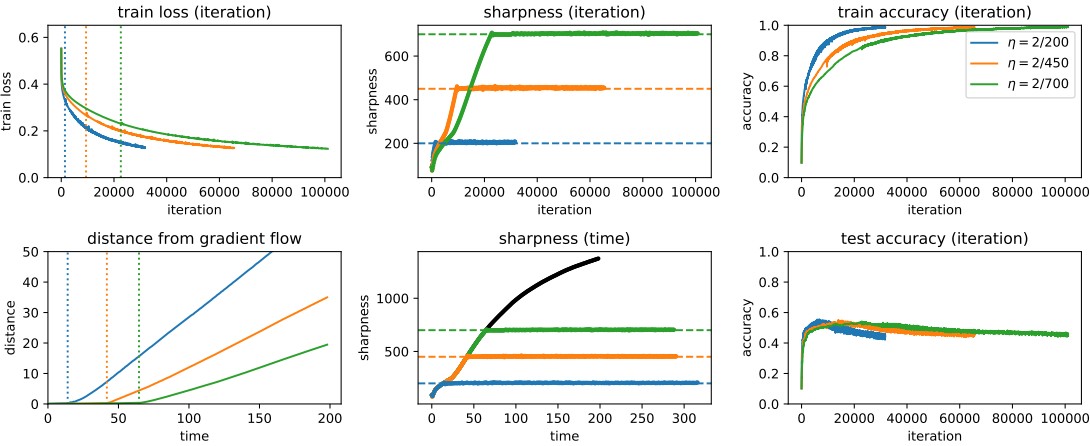

Figure 57: **Gradient descent.** Refer to the Figure 49 caption for more information.

### J.8.2 CROSS-ENTROPY LOSS

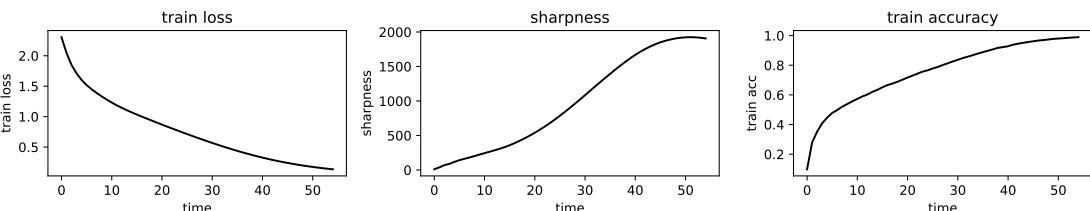

Figure 58: **Runge-Kutta.** Refer to the Figure 48 caption for more information. Additionally, in this figure the sharpness drops at the end of training due to the cross-entropy loss.

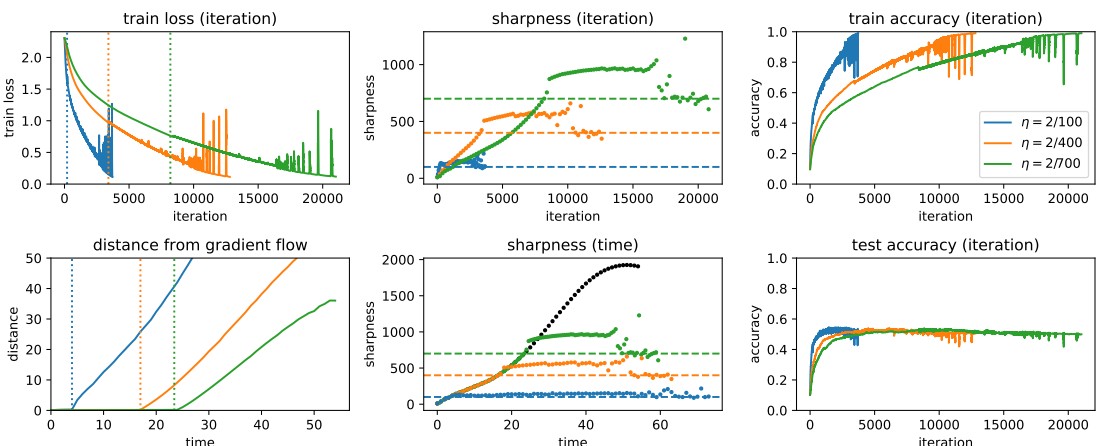

Figure 59: **Gradient descent.** Refer to the Figure 49 caption for more information.

## J.9  CONVOLUTIONAL TANH NETWORK WITH AVERAGE POOLING

### J.9.1  CROSS-ENTROPY LOSS

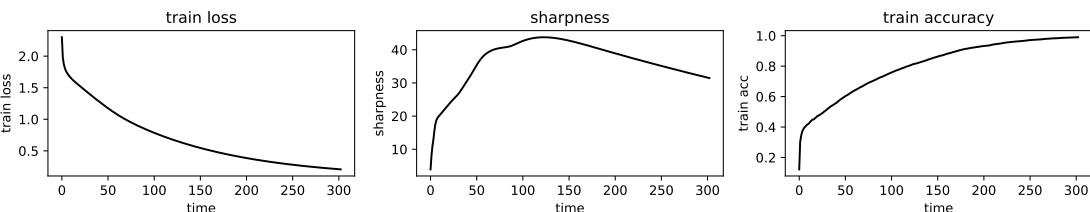

Figure 60: **Gradient flow.** Refer to the Figure 28 caption for more information. Additionally, in this figure the sharpness drops at the end of training due to the cross-entropy loss.

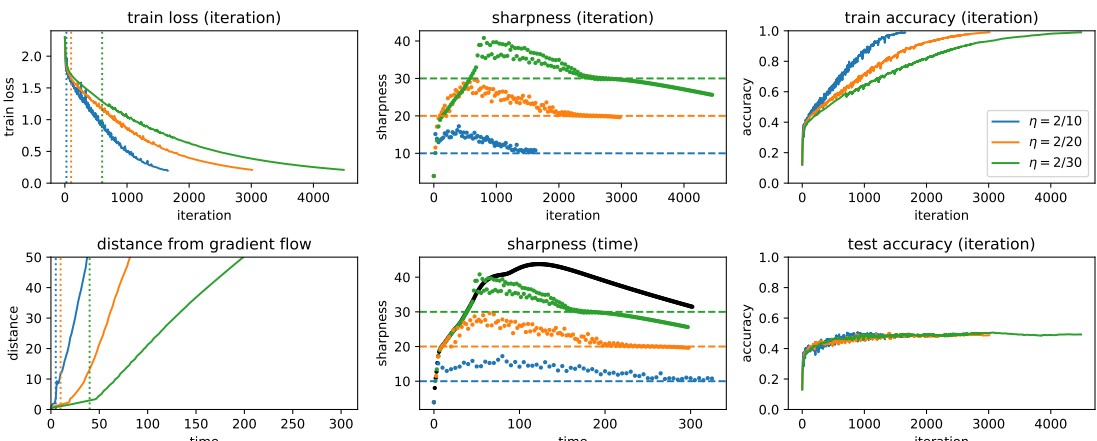

Figure 61: **Gradient descent.** Refer to the Figure 29 caption for more information.

## J.10 CONVOLUTIONAL ELU NETWORK WITH AVERAGE POOLING

### J.10.1 SQUARE LOSS

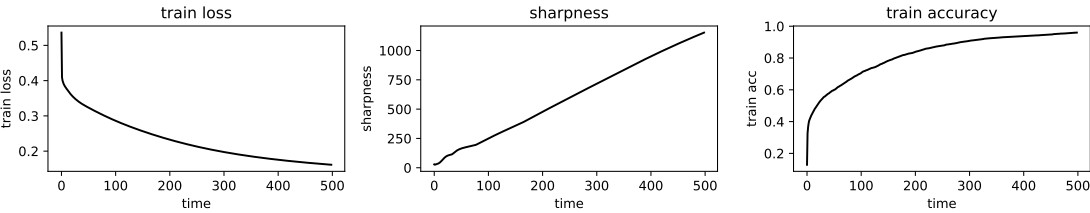

Figure 62: **Gradient flow.** Refer to the Figure 28 caption for more information.

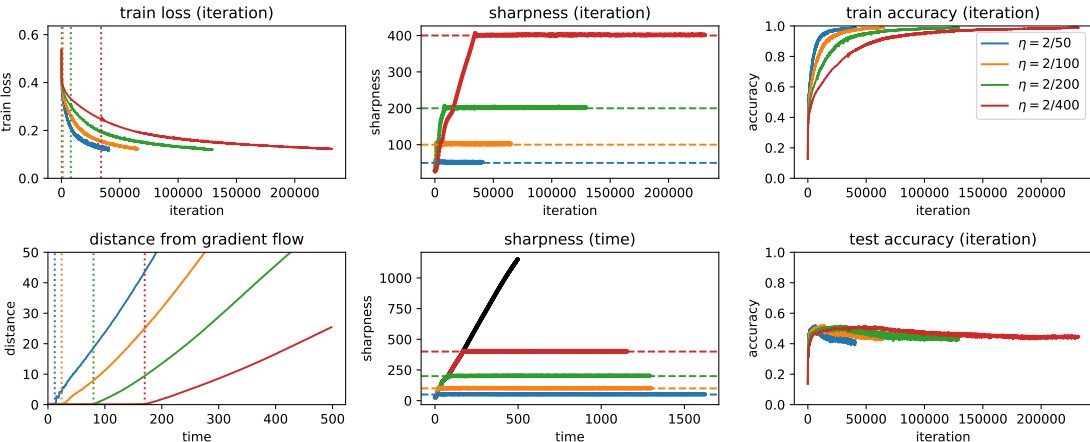

Figure 63: **Gradient descent.** Refer to the Figure 29 caption for more information.

### J.10.2 CROSS-ENTROPY LOSS

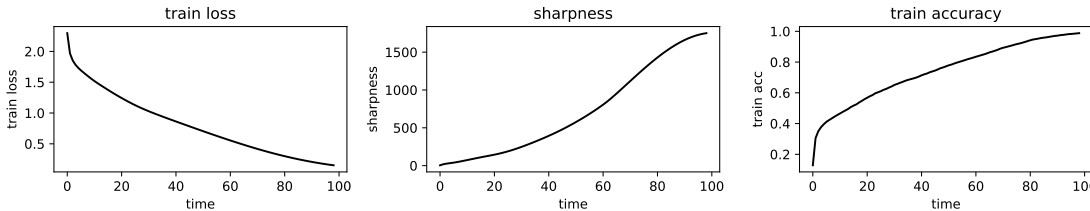

Figure 64: **Gradient flow.** Refer to the Figure 28 caption for more information. Additionally, in this figure the sharpness drops at the end of training due to the cross-entropy loss.

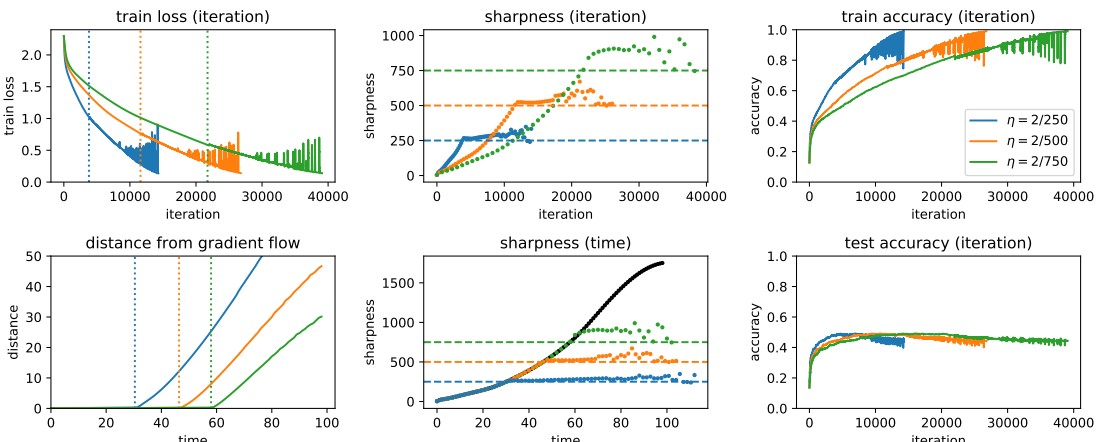

Figure 65: **Gradient descent.** Refer to the Figure 29 caption for more information.

### J.11 Convolutional ReLU network with average pooling

Note that since ReLU is not continuously differentiable, the training objective is not continuously differentiable, and so a unique gradient flow trajectory is not guaranteed to exist.

#### J.11.1 Square loss

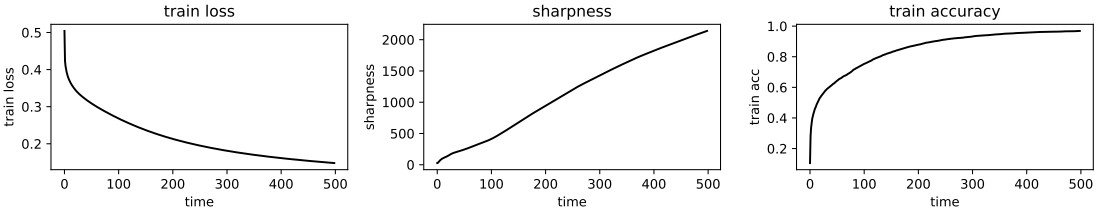

Figure 66: **Runge-Kutta.** Refer to the Figure 48 caption for more information.

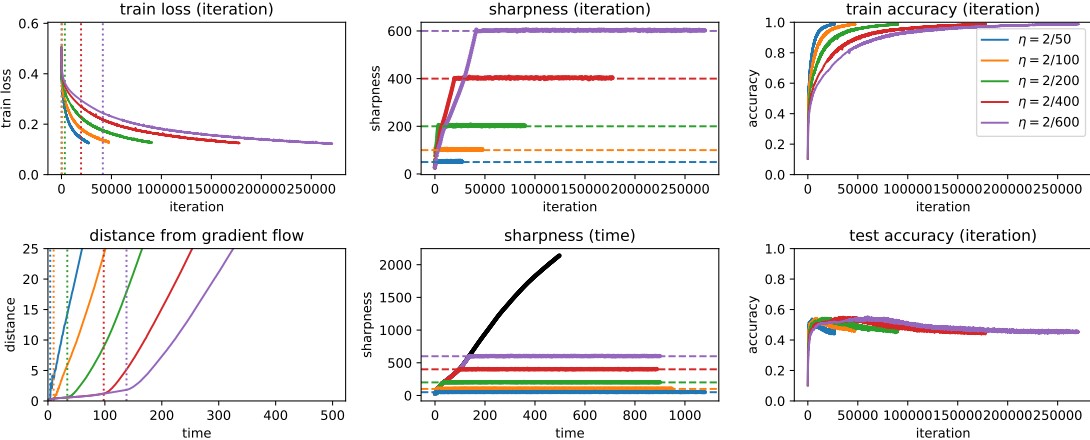

Figure 67: **Gradient descent.** Refer to the Figure 49 caption for more information.

### J.11.2 CROSS-ENTROPY LOSS

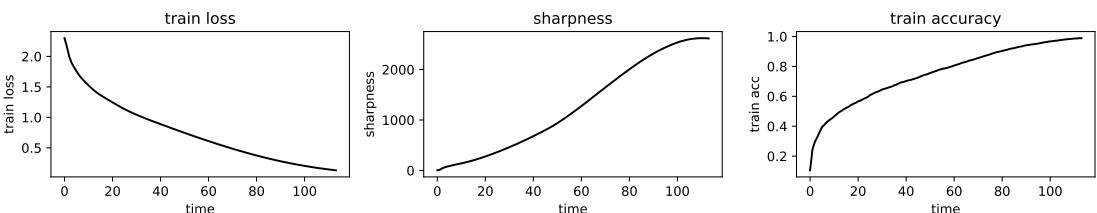

Figure 68: **Runge-Kutta.** Refer to the Figure 48 caption for more information. Additionally, in this figure the sharpness drops at the end of training due to the cross-entropy loss.

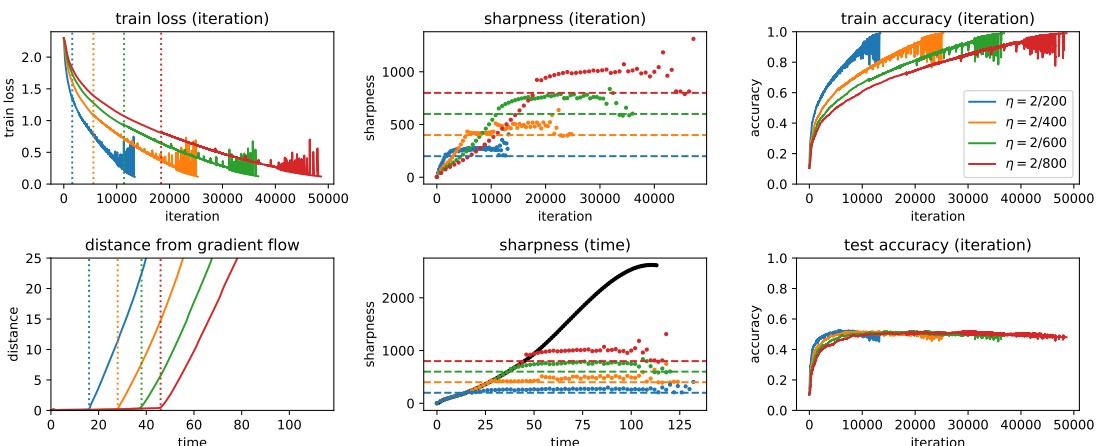

Figure 69: **Gradient descent.** Refer to the Figure 49 caption for more information.

## K   BATCH NORMALIZATION EXPERIMENTS

In this appendix, we demonstrate that our findings hold for networks that are trained with batch normalization (BN) (Ioffe & Szegedy, 2015). We experiment on a size-5,000 subset of CIFAR-10, and we consider convolutional networks with three different activation functions: ELU (Figure 70-71, tanh (Figure 72-73), and ReLU (Figure 74-75). See §I.3 for experimental details.

Empirically, our findings hold for batch-normalized networks. The one catch is that when training batch-normalized networks at very small step sizes, it is apparently inadequate to measure the sharpness directly at the iterates themselves, as we do elsewhere in the paper. Namely, observe that in Figure 71(e), when we run gradient descent at the red step size, the sharpness (measured directly at the iterates) plateaus a bit *beneath* the value $2/\eta$. At first, this might sound puzzling: after all, if the sharpness is less than $2/\eta$ then gradient descent should be stable. The explanation is that the sharpness *in between* successive iterates does in fact cross $2/\eta$. In Figure 71(b), we track the maximum sharpness on the path "in between" successive iterates. (To estimate the maximum sharpness between a pair of successive iterates, we compute the sharpness at a grid of eight points spaced evenly between them, and then take the maximum of these values.) Observe that this quantity does rise to $2/\eta$ and hover there. We do not know why measuring the sharpness between iterates is necessary for batch-normalized networks, whereas for non-BN networks it suffices to measure the sharpness only at the iterates themselves.

In §K.1, we reconcile these findings with Santurkar et al. (2018).

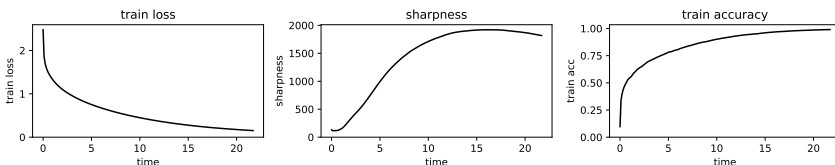

Figure 70: We train a **ELU CNN (+ BN)** using **gradient flow**.

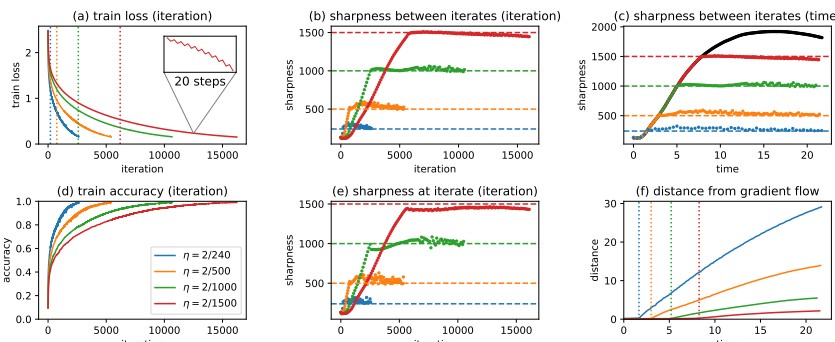

Figure 71: We train an **ELU CNN (+ BN)** using **gradient decent** at a range of step sizes. **(a)** we plot the train loss, with a vertical dotted line marking the iteration where the sharpness on the path first crosses $2/\eta$. The inset shows that the red curve is indeed behaving non-monotonically. **(b)** we track the sharpness "between iterates." This means that instead of computing the sharpness right at the iterates themselves (as we do elsewhere in the paper, and in pane (e) here), we compute the maximum sharpness on the line between between successive iterates. Observe that this quantity rises to $2/\eta$ (marked by the horizontal dashed line) and then hovers right at, or just above that value. **(c)** we plot the same quantity by "time" (= iteration × step size) rather than iteration. **(e)** we plot the sharpness at the iterates themselves. Note that for the red step size, this quantity plateaus at a value that is beneath $2/\eta$. **(f)** distance from the gradient flow trajectory.

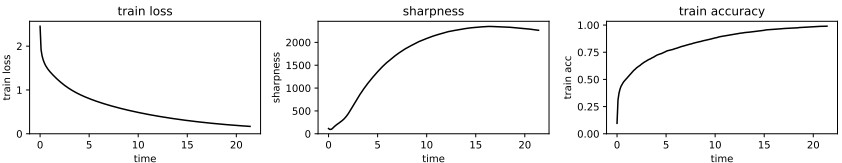

Figure 72: We train a **tanh CNN (+ BN)** using **gradient flow**.

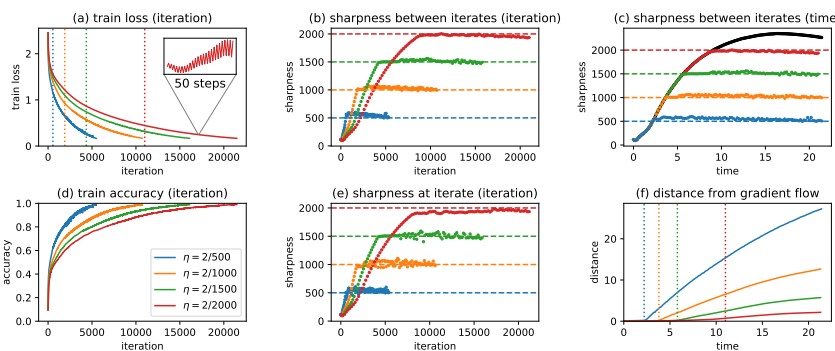

Figure 73: We train a **tanh CNN (+ BN)** using **gradient decent**.

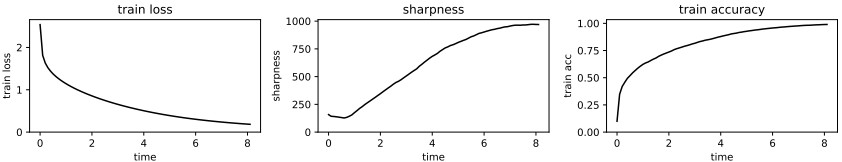

Figure 74: We train a **ReLU CNN (+ BN)** using **gradient flow**.

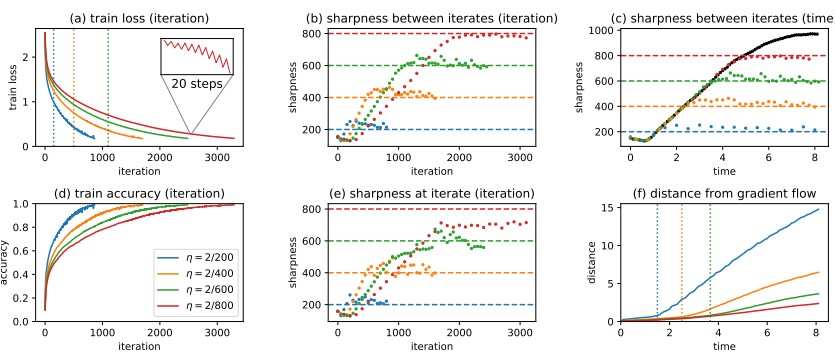

Figure 75: We train a **ReLU CNN (+ BN)** using **gradient descent**.

## K.1 RELATION TO SANTURKAR ET AL. (2018)

We have demonstrated that the sharpness hovers right at (or just above) the value $2/\eta$ when both BN and non-BN networks are trained using gradient descent at reasonable step sizes. Therefore, at least in the case of full-batch gradient descent, it cannot be said that batch normalization decreases the sharpness (i.e. improves the local $L$-smoothness) along the optimization trajectory.

Santurkar et al. (2018) argued that batch normalization improves the *effective smoothness* along the optimization trajectory, where effective smoothness is defined as the Lipschitz constant of the gradient in the update direction (i.e. the negative gradient direction, for full-batch GD). That is, given an objective function $f$, an iterate $\theta$, and a distance $\alpha$, the effective smoothness of $f$ at parameter $\theta$ and distance $\alpha$ is defined in Santurkar et al. (2018) as

$$\sup_{\gamma \in [0,\alpha]} \frac{\|\nabla f(\theta) - \nabla f(\theta - \gamma \nabla f(\theta))\|_2}{\|\gamma \nabla f(\theta)\|_2}$$

where the sup can be numerically approximated by evaluating the given ratio at several values $\gamma$ spaced uniformly between 0 and $\alpha$.

In Figure 76, we train two ReLU CNNs — one with BN, one without — at the same set of step sizes, and we monitor both the sharpness (i.e. the $L$-smoothness) and the effective smoothness. When computing the effective smoothness, we use a distance $\alpha = \eta$. Observe that for both the BN and the non-BN network, the effective smoothness initially hovers around zero, but once gradient descent enters the Edge of Stability, the effective smoothness jumps to the value $2/\eta$ and then remains there. Thus, at least for full-batch gradient descent on this particular architecture, batch normalization does *not* improve the effective smoothness along the optimization trajectory. (Despite this, note that for each step size, the BN network trains faster than the non-BN network, confirming that BN does accelerate training.)

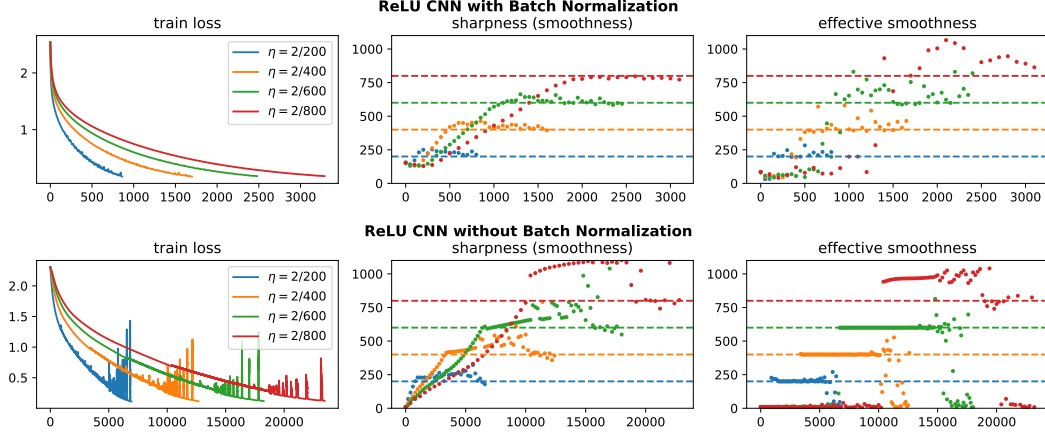

Figure 76: On a 5,000-size subset of CIFAR-10, we train a ReLU CNN both with BN (**top row**) and without BN (**bottom row**) at the same grid of step sizes. We plot the sharpness/smoothness (**center column**) as well as the effective smoothness (Santurkar et al., 2018) (**right column**). Observe that for both networks, the effective smoothness hovers around zero initially, and jumps up to $2/\eta$ once gradient descent enters the Edge of Stability.

Note that this finding is actually consistent with Figure 4(c) in Santurkar et al. (2018), which is meant to show that BN improves effective smoothness when training a VGG network using SGD. Their Figure 4(c) shows that during SGD with step size $\eta = 0.1$, the effective smoothness hovers around the value 20 for both the BN and the non-BN network. Since $20 = 2/(0.1)$, this is fully consistent with our findings (though they use SGD rather than full-batch GD). Figure 4(c) does show that the effective smoothness behaves more *regularly* for the BN network than for the non-BN network. But we disagree with their interpretation of this figure as demonstrating that BN improves the effective smoothness during training.

The other piece of evidence in Santurkar et al. (2018) in support of the argument that batch normalization improves the effective smoothness during training is their Figure 9(c). This figure shows that a deep linear network (DLN) trained without BN has a much larger (i.e. worse) effective smoothness during training than a DLN trained with BN. However, for this figure, the distance $\alpha$ used to compute effective smoothness was larger than the training step size $\eta$ by a factor of 30. The effective smoothness at distances larger than the step size does not affect training. We have verified that

when effective smoothness is computed at a distance equal to the training step size (i.e. $\alpha = \eta$), the effective smoothness for the DLN with BN and for the DLN without BN both hover right at $2/\eta$.

Specifically, in Figure 77 and Figure 78, we train a DLN both with and without BN (respectively), and we measure the effective smoothness at a distance $\alpha = 30\eta$, as done in Figure 9(c) of Santurkar et al. (2018). We use the same experimental setup and the same step size of $\eta = 1e\text{-}6$ as they do, and we repeat the experiment across four random seeds. Observe that when training the BN network, the effective smoothness hovers right at $2/\eta$ (marked by the horizontal black line), whereas when training the non-BN network, the effective smoothness is much larger. This is consistent with Figure 9(c) in Santurkar et al. (2018). However, in Figure 79 and 80, we measure the effective smoothness at the actual step size $\alpha = \eta$. When effective smoothness is computed in this way, we observe that for both the network with BN and the network without BN, the effective smoothness hovers right at $2/\eta$. Therefore, we conclude that there is no evidence that the use of batch normalization improves either the smoothness or the effective smoothness along the optimization trajectory. (That said, this experiment possibly explains why the batch-normalized network permits training with larger step sizes.)

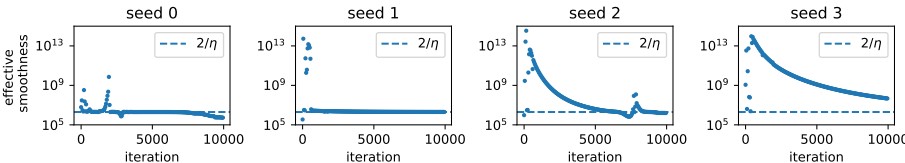

Figure 77: When training a deep linear network **without BN**, we measure effective smoothness at a distance $\alpha = 30\eta$ that is **30x larger than the step size**.

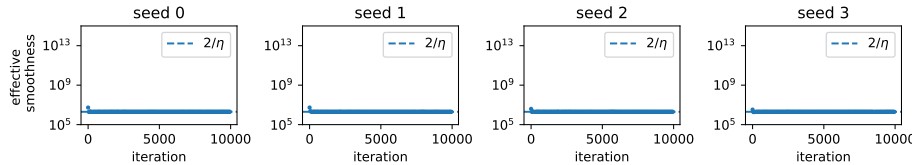

Figure 78: When training a deep linear network **with BN**, we measure effective smoothness at a distance $\alpha = 30\eta$ that is **30x larger than the step size**.

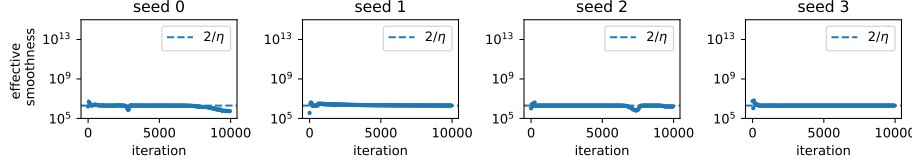

Figure 79: When training a deep linear network **without BN**, we measure effective smoothness at a distance $\alpha = \eta$ that is **equal to the step size**.

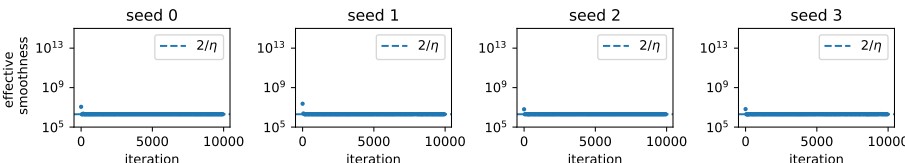

Figure 80: When training a deep linear network **with BN**, we measure effective smoothness at a distance $\alpha = \eta$ that is **equal to the step size**.

## L  ADDITIONAL TASKS

So far, we have verified our findings on image classification and language modeling. In this appendix, we verify our findings on three additional tasks: training a Transformer on the WikiText-2 language modeling dataset (L.1), training a one-hidden-layer network on a one-dimensional toy regression task (L.2), and training a deep linear network (L.3).

### L.1  TRANSFORMER ON WIKITEXT-2

We consider the problem of training a Transformer on the WikText-2 word-level language modeling dataset (Merity et al., 2016). See §I.4 for full experimental details. In Figure 81, we train using gradient flow (only partially, not to completion). Observe that the sharpness continually rises. In Figure 82, we train using gradient descent at a range of step sizes. Consistent with our general findings, for each step size $\eta$, we observe that the sharpness rises to $2/\eta$ and then hovers right at, or just above, that value. However, for this Transformer, we do *not* observe that gradient descent closely tracks the gradient flow trajectory at the beginning of training.

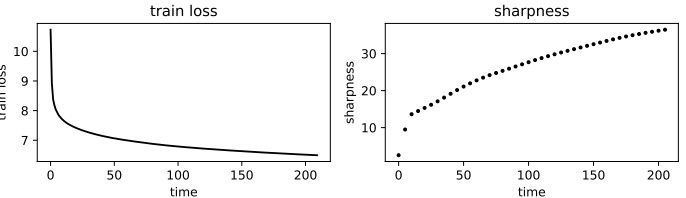

Figure 81: **Training a Transformer using gradient flow**.

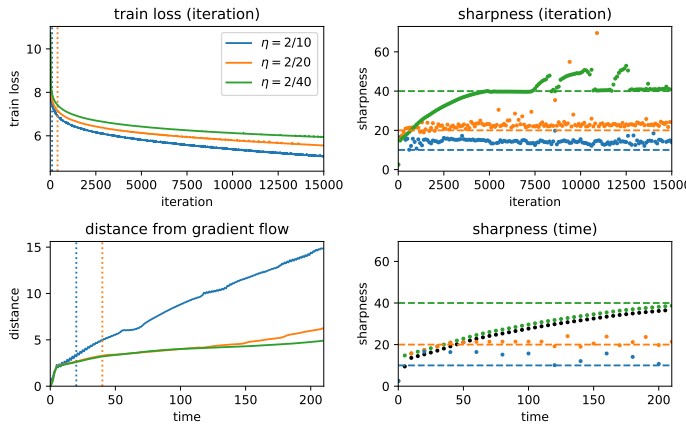

Figure 82: **Training a Transformer using gradient descent**. We train a Transformer for 15,000 iterations on the WikiText-2 language modeling dataset. (For this problem, training to completion would not be computationally practical.) **Top left:** we plot the train loss curves, with a vertical dotted line marking the iteration where the sharpness first crosses $2/\eta$. The train loss decreases monotonically before this line, but behaves non-monotonically afterwards. **Top right:** we plot the evolution of the sharpness, with a horizontal dashed line marking the value $2/\eta$. Observe that for each step size, the sharpness rises to $2/\eta$ and then hovers right at, or just above, that value. **Bottom left**: for the initial phase of training, we plot the distance between the gradient descent trajectory and the gradient flow trajectory, with a vertical dotted line marking the gradient descent iteration where the sharpness first crosses $2/\eta$. Observe that the distance begins to rise from the start of training, indicating that for this architecture, gradient descent does *not* track the gradient flow trajectory initially. **Bottom right**: for the initial phase of training, we plot the evolution of the sharpness by "time" = iteration $\times \eta$ rather than iteration. The black dots are the sharpness during gradient flow.

### L.2 ONE-DIMENSIONAL TOY REGRESSION TASK

**Task**   Our toy regression problem is to approximate a Chebyshev polynomial using a neural network. To generate a toy dataset, we take 20 points spaced uniformly on the interval $[-1, 1]$, and we label them noiselessly using the Chebyshev polynomial of some degree $k$. Note that the Chebyshev polynomial of degree $k$ is a polynomial with $k$ zeros that maps the domain $[-1, 1]$ to the range $[-1, 1]$. Figure 83 shows the Chebyshev datasets for degree 3, 4, and 5.

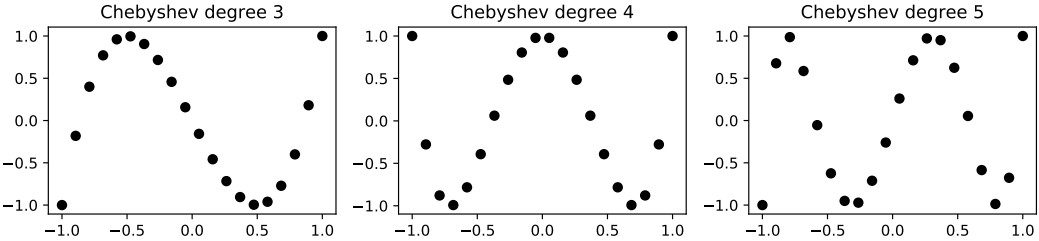

Figure 83: The datasets for the toy one-dimensional regression task.

**Network**   For the network, we use a tanh architecture with one hidden layer of $h = 100$ units, initialized using Xavier initialization. We train using the MSE loss until the loss reaches 0.05.

**Results**   In Figure 84, we fit the Chebyshev degree 3, 4, and 5 datasets using gradient flow. Empirically, the higher the degree, the more the sharpness rises during the course of training: on the degree 3 polynomial, the sharpness rises by a factor of 1.2; on the degree 4 polynomial, by a factor of 3.2; and on the degree 5 polynomial, by a factor of 63.5.

In Figure 85 and Figure 86, we fit the degree 4 and 5 datasets using gradient descent at a range of step sizes. We observe mostly the same Edge of Stability behavior as elsewhere in the paper. The only difference is that for the degree 5 dataset, after the sharpness hits $2/\eta$, the training loss *first* undergoes a temporary period of non-monotonicity in which no progress is made, and *then* it decreases monotonically until training is finished (in contrast to our other experiments where we observe that training loss behaves non-monotonically at the same time as it is consistently decreasing).

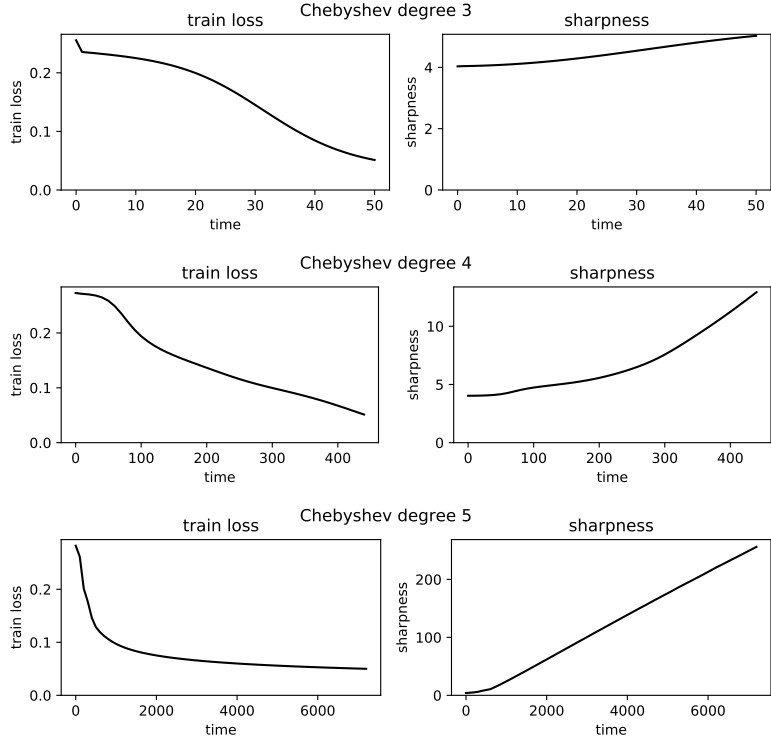

Figure 84: **Fitting Chebyshev polynomials using gradient flow**. We use gradient flow to fit a one-hidden-layer tanh network to the Chebyshev polynomials of degrees 3, 4, and 5. We observe that the sharpness rises more when fitting a higher-degree polynomial, likely because the dataset is more complex.

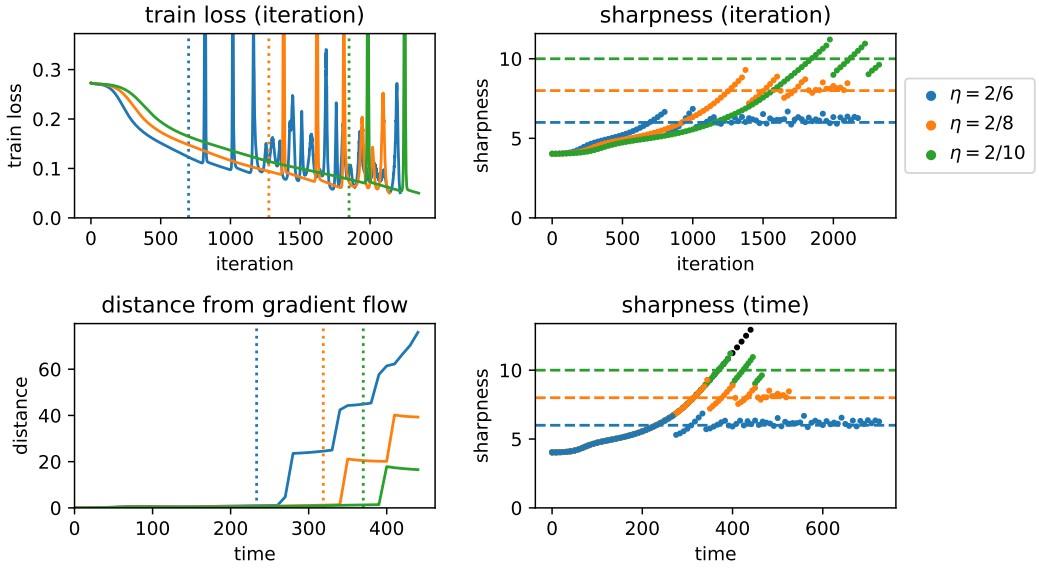

Figure 85: **Chebyshev degree 4 (gradient descent)**. We fit the Chebyshev polynomial of degree 4 using gradient descent at a range of step sizes (see legend).

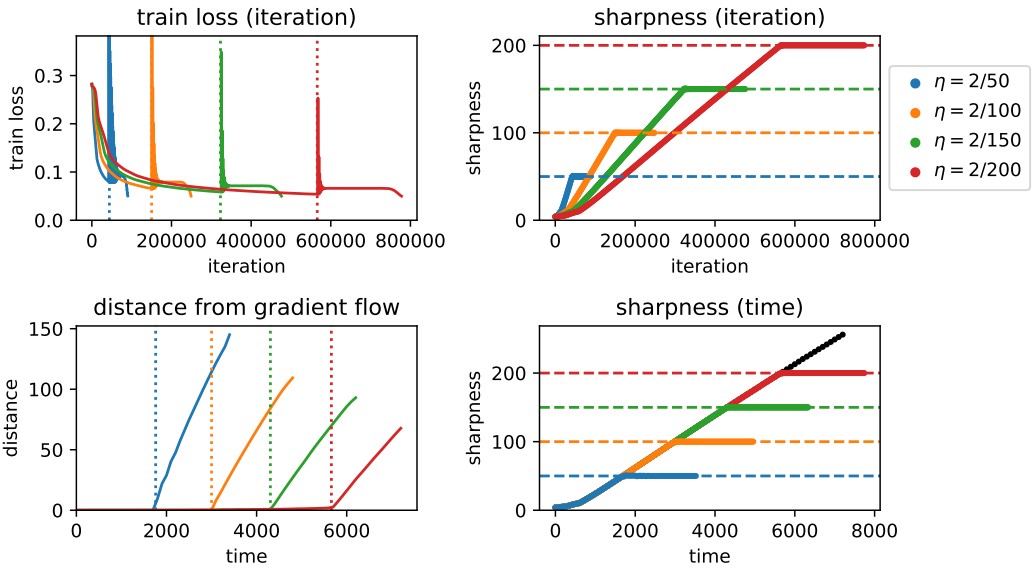

Figure 86: **Chebyshev degree 5 (gradient descent)**. We fit the Chebyshev polynomial of degree 5 using gradient descent at a range of step sizes (see legend).

### L.3 DEEP LINEAR NETWORK

**Task** The task is to map $n$ inputs $\mathbf{x}_1, \ldots, \mathbf{x}_n \subseteq \mathbb{R}^d$ to $n$ targets $\mathbf{y}_1, \ldots, \mathbf{y}_n \subseteq \mathbb{R}^d$ using a function $f : \mathbb{R}^d \to \mathbb{R}^d$. Error is measured using the square loss, i.e. the objective is $\frac{1}{n} \sum_{i=1}^{n} \|f(\mathbf{x}_i) - \mathbf{y}_i\|_2^2$. Let $\mathbf{X} \in \mathbb{R}^{n \times d}$ be the vertical stack of the inputs, and let $\mathbf{Y} \in \mathbb{R}^{n \times d}$ be the vertical stack of the targets. We first generate $\mathbf{X}$ as a random whitened matrix (i.e. $\frac{1}{n}\mathbf{X}^T\mathbf{X} = \mathbf{I}$). To generate $\mathbf{X}$ as a random whitened matrix, we sample a $n \times d$ matrix of standard Gaussians, and then set $\mathbf{X}$ to be $\sqrt{n}$ times the Q factor in the QR factorization of that matrix. We then generate $\mathbf{Y}$ via $\mathbf{Y} = \mathbf{X}\mathbf{A}^T$, where $\mathbf{A} \in \mathbb{R}^{d \times d}$ is a random matrix whose entries are sampled i.i.d from the standard normal distribution.

We use $n = 50$ datapoints with a dimension of $d = 50$.

**Network** The function $f : \mathbb{R}^d \to \mathbb{R}^d$ is implemented as a $L$-layer deep linear network: $f(\mathbf{x}) = \mathbf{W}_L \ldots \mathbf{W}_2\mathbf{W}_1\mathbf{x}$, with $\mathbf{W}_\ell \in \mathbb{R}^{d \times d}$. We initialize all layers of the deep linear network using Xavier initialization: all entries of each $\mathbf{W}_\ell$ are drawn i.i.d from $\mathcal{N}(0, \frac{1}{d})$.

We use a network with $L = 20$ layers.

**Results** In Figure 87, we train the network using gradient flow. (Since it is unclear whether the network can be trained to zero loss, and how long this would take, we arbitrarily chose to stop training at time 100.) In Figure 88, we train the network using gradient descent at a range of step sizes. We observe mostly the same Edge of Stability behavior as elsewhere in the paper. The only difference is that in Figure 88, the train loss does not really behave non-monotonically — for each step size $\eta$, there is a brief blip at some point, but otherwise, the train loss decreases monotonically.

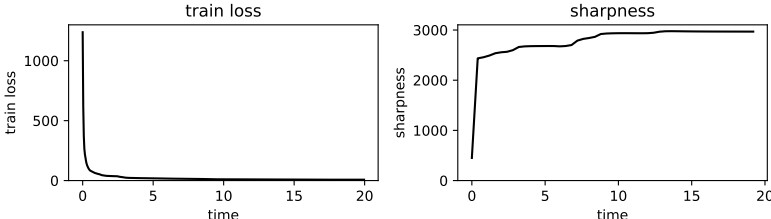

Figure 87: **Training a deep linear network using gradient flow**. We use gradient flow to train a deep linear network. Since it is unclear whether this network can be trained to zero loss (or how long that would take), we arbitrarily chose to stop training at time 100.

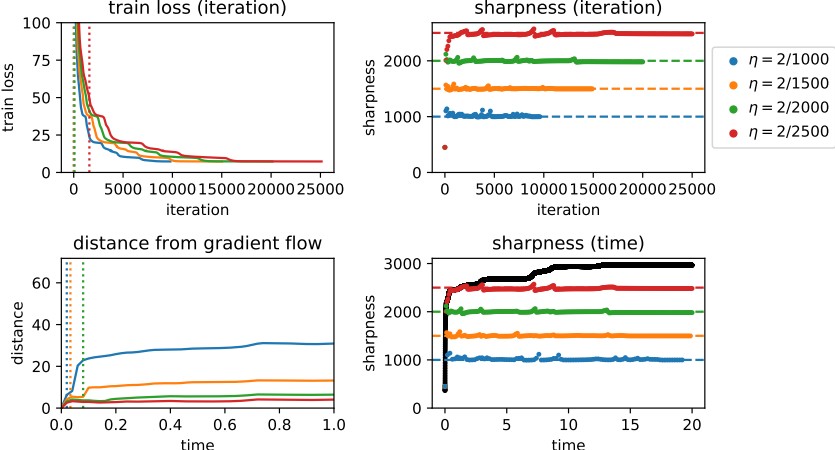

Figure 88: **Training a deep linear network using gradient descent**. We train a deep linear network using gradient descent at a range of step sizes (see legend). (Note that in the top left pane, the blue, orange and green dotted lines are directly on top of one another.)

## M    EXPERIMENTS: STANDARD ARCHITECTURES ON CIFAR-10

In this appendix, we demonstrate that our findings hold for three standard architectures on the standard dataset CIFAR-10. The three architectures are: a VGG with batch normalization (Figures 89-90), a VGG without batch normalization (Figures 91-92), and a ResNet with batch normalization (Figures 93 -94). See §I.2 for full experimental details.

For each of these three architectures, we confirm our main points: (1) so long as the sharpness is less than $2/\eta$, the sharpness tends to increase; and (2) if the sharpness reaches $2/\eta$, gradient descent enters a regime (the Edge of Stability) in which (a) the training loss behaves non-monotonically, yet consistently decreases over long timescales, and (b) the sharpness hovers right at, or just above, the value $2/\eta$. Moreover, we observe that even though these architectures use the ReLU activation function (which is not continuously differentiable), gradient descent closely tracks the Runge-Kutta trajectory until reaching the point on that trajectory where the sharpness hits $2/\eta$.

Furthermore, we observe that for these three standard architectures, the following additional points hold: (1) progressive sharpening occurs to a dramatic degree, and (2) stable step sizes are so small as to be completely unreasonable.

**Progressive sharpening occurs to a dramatic degree**    To assess the degree of progressive sharpening, we train these networks using Runge-Kutta / gradient flow, which can be viewed as gradient descent with an infinitesimally small step size. (In practice, the Runge-Kutta algorithm does have a step size parameter, and throughout training, we periodically adjust this step size in order to ensure that the algorithm remains stable.) Intuitively, training with gradient flow tell us how far the sharpness would rise "if it didn't have to worry about" instability caused by nonzero step sizes. In Figure 89, we train the VGG with BN to completion (99% training accuracy) using Runge-Kutta / gradient flow, and find that the sharpness rises from its initial value of 6.38 to a peak value of 2227.6. For the other two architectures, progressive sharpening occurs to such a degree that it is not computationally feasible for to train using Runge-Kutta / gradient flow all the way to completion. (The reason is that in regions where the sharpness is high, the Runge-Kutta step size must be made small, so Runge-Kutta requires very many iterations.) Therefore, we instead train these two networks only partially. In Figure 91, for the VGG without BN, we find that the sharpness rises from its initial value of 0.64 to the value 2461.78 at 37.1% accuracy, when we stop training. In Figure 93, for the ResNet, we find that the sharpness rises from its initial value of 1.07 to the value 760.6 at 43.2% accuracy, when we stop training. Thus, even though we observed in Appendix D that progressive sharpening attenuates as the width of fully-connected networks is made larger, it appears that either: (1) this does not happen for modern families of architectures such as ResNet and VGG, or (2) this does happen for modern families of architectures, but practical network widths lie on the narrow end of the scale.

**Stable step sizes are so small as to be unreasonable**    Recall from 3.3 that if $\lambda_{\max}$ is the maximum sharpness along the gradient flow trajectory, then any stable step size must be less than $2/\lambda_{\max}$. Therefore, for these three architectures, because progressive sharpening occurs to a dramatic degree (i.e. $\lambda_{\max}$ is extraordinarily large), any stable step size must be extraordinarily small, which means that training will require many iterations. Yet, at the same time, we find that these three networks can be successfully trained in far fewer iterations by using larger step step size. This means that training at a stable step size is extremely suboptimal. We now elaborate on this point:

**VGG with BN.** For this network, gradient flow terminates at time 15.66, and the maximum sharpness along the gradient flow trajectory is 2227.6. Therefore, the largest stable step size is $2/2227.59 = 0.000897$, and training to completion at this step size would take $14.91/0.000897 = 16622$ iterations. Meanwhile, we empirically observe that the network can also be trained to completion at the much larger step size of $\eta = 0.16$ in just 329 iterations. Therefore, using a stable step size is suboptimal by a factor of at least $16622/329 = 50.5$.

For the other two architectures, since we are unable to train *to completion* using gradient flow, we are unable to obtain a tight lower bound for the number of iterations required to run gradient descent to completion at a stable step size. Therefore, by extension, we are unable to compute a tight lower bound for the suboptimality factor of stable step sizes. As a substitute, we will instead compute

both: (1) a tight lower bound on the suboptimality of training *partially* at a stable step size, and (2) a very loose lower bound on the suboptimality of training to completion at a stable step size.

**VGG without BN.** For this network, gradient flow reaches 37.1% accuracy at time 8, and the maximum sharpness up through this point on the gradient flow trajectory is 2461.8. Therefore, the largest stable step size is $2/2461.8 = 0.00081$, and training to 37.1% accuracy at this step size would require $8/0.00081 = 9,876$ iterations. Meanwhile, we empirically observe that the network can also be trained to 37.1% accuracy at the larger step size of $\eta = 0.16$ in just 355 iterations. Therefore, when training this network to 37% accuracy, stable step sizes are suboptimal by a factor of at least $9876/355 = 27.8$. This is a tight lower bound on the suboptimality of training to 37.1% accuracy at a stable step size.

To obtain a loose lower bound on the suboptimality of training the VGG without BN *to completion* at a stable step size, we note that (a) since the maximum sharpness up through time 8 is 2461.8, the maximum sharpness along the *entire* gradient flow trajectory must be at least 2461.8; and (b) since by time 8 gradient flow has only attained 37.1% training accuracy, the time to reach 99% accuracy (i.e. completion) must be at least 8. (Note that both of these lower bounds are extremely loose.) Therefore, training this network to completion at a stable step size would require at least $8/(2/2461.8) = 9,876$ iterations (which is the same number of iterations as training to 37.1% accuracy at a stable step size). Meanwhile, we find that the network can be trained to completion at the larger step size of $\eta = 0.16$ in just 1782 iterations. Therefore, training to completion at a stable step size is suboptimal by a factor of at least $9,876/1782 = 5.54$.

**ResNet.** For this network, gradient flow reaches 43.2% accuracy at time 70, and the maximum sharpness up through this point on the gradient flow trajectory is 760.6. Therefore, the largest stable step size is $2/760.6 = 0.0026$, and training to 43.2% accuracy at this step size would require $70/0.0026 = 26,923$ iterations. Meanwhile, we empirically observe that the network can also be trained to 43.2% accuracy at the larger step size of $\eta = 2.0$ in just 99 iterations. Therefore, when training this network to 43.2% accuracy, stable step sizes are suboptimal by a factor of at least $26,923/99 = 271.9$. This is a tight lower bound on the suboptimality of training to 43.2% accuracy at a stable step size. For the loose lower bound on the suboptimality of training to completion at a stable step size, note that (by similar reasoning as the VGG-without-BN above), training to completion at a stable step size must require at least $26,923$ iterations. Meanwhile, we find that the network can be trained to completion at the larger step size of $\eta = 2.0$ in just 807 iterations. Therefore, training to completion at a stable step size is suboptimal by a factor of at least $26,923/807 = 33.3$.

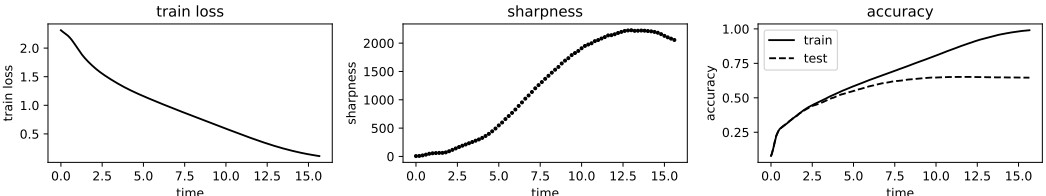

Figure 89: We train a **VGG with BN** to completion using **gradient flow** (Runge-Kutta). Observe that the sharpness rises dramatically from 6.38 at initialization to a peak of 2227.59.

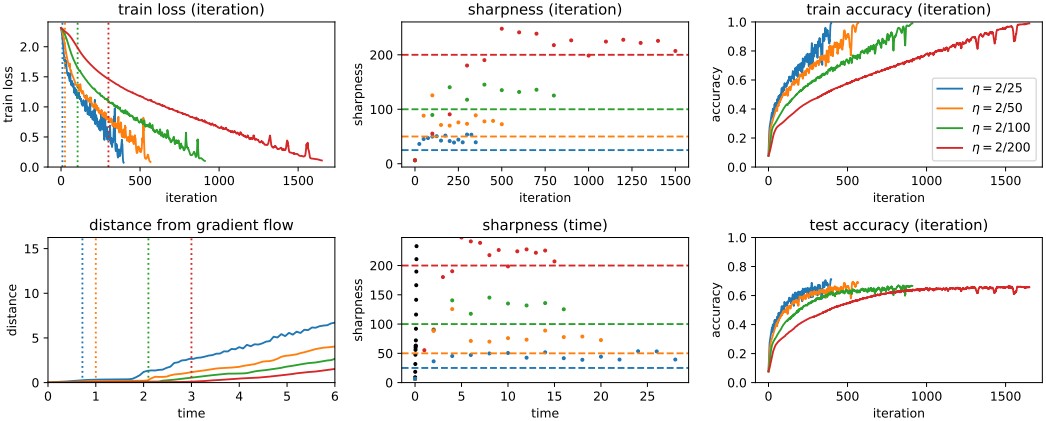

Figure 90: We train a **VGG with BN** to completion using **gradient descent** at different step sizes (see legend in the top right pane). **Top left**: we plot the train loss, with a vertical dotted line marking the iteration where the sharpness first crosses $2/\eta$. **Top center**: we plot the evolution of the sharpness, with a horizontal dashed line (of the appropriate color) marking the value $2/\eta$. **Top right**: we plot the train accuracy. **Bottom left**: for the initial phase of training, we monitor the distance between the gradient descent iterate at iteration $t/\eta$, and the gradient flow solution at time $t$, with a vertical dotted line (of the appropriate color) marking the time when the sharpness crosses $2/\eta$. Observe that the distance between gradient descent and gradient flow is almost zero before this instant, but starts to rise shortly afterwards. **Bottom center**: we plot the sharpness by time (= iteration $\times\eta$) rather than iteration. The black dots are the sharpness of the gradient flow trajectory, which shoots up immediately. **Bottom right**: we plot the test accuracy.

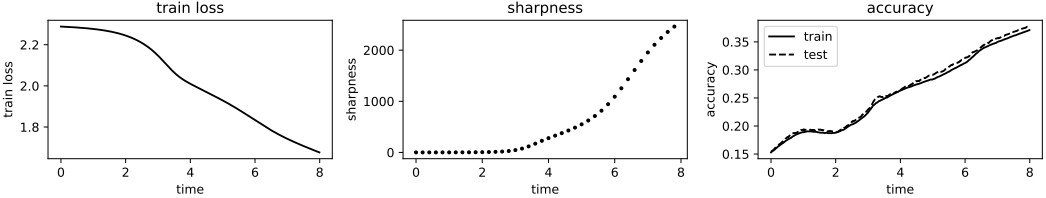

Figure 91: We train a **VGG without BN** to 37.1% accuracy using **gradient flow** (Runge-Kutta). Observe that the sharpness rises dramatically from 0.64 at initialization to 2461.78 at 37.1% accuracy. We train this network only partway because training this network to completion would be too computationally expensive: Runge-Kutta runs very slowly when the sharpness is high (because it is forced to take small steps) and for this network the sharpness is extremely high when the train accuracy is only 37% (which means that there is a long way to go).

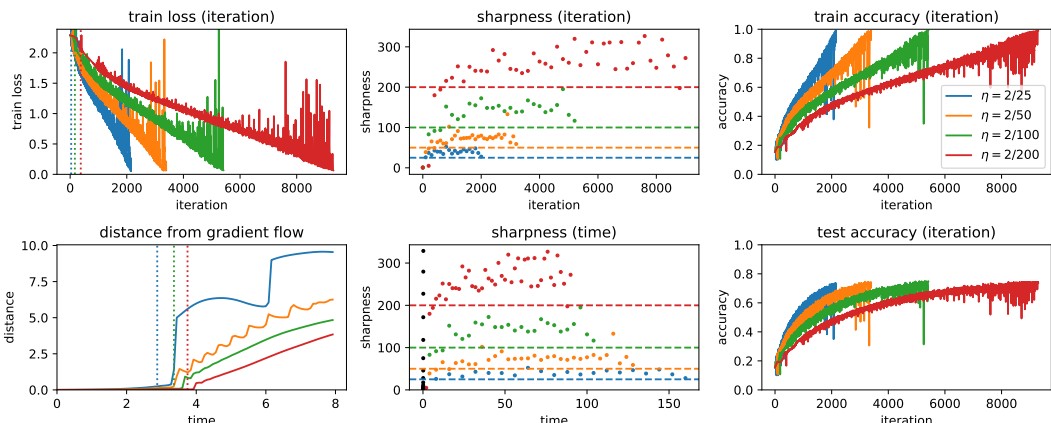

Figure 92: We train a **VGG without BN** to completion using **gradient descent** at a range of step sizes (see legend in the top right pane). Refer to the Figure 90 caption for more information.

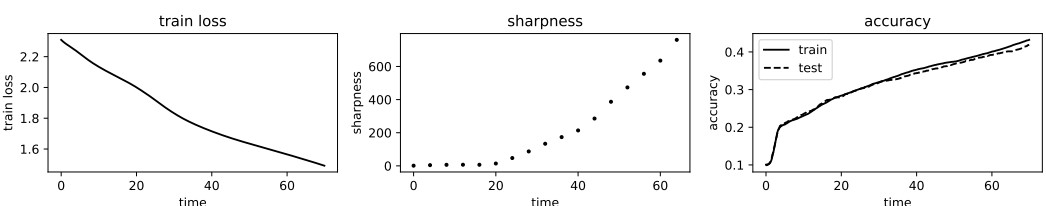

Figure 93: We train a **ResNet** to 43.2% accuracy using **gradient flow** (Runge-Kutta). Observe that the sharpness rises dramatically from 1.07 at initialization to 760.63 at 43.2% accuracy. We train this network only partway because training this network to completion would be too computationally expensive: Runge-Kutta runs very slowly when the sharpness is high (because it is forced to take small steps) and for this network the sharpness is extremely high when the train accuracy is only 42% (which means that there is a long way to go).

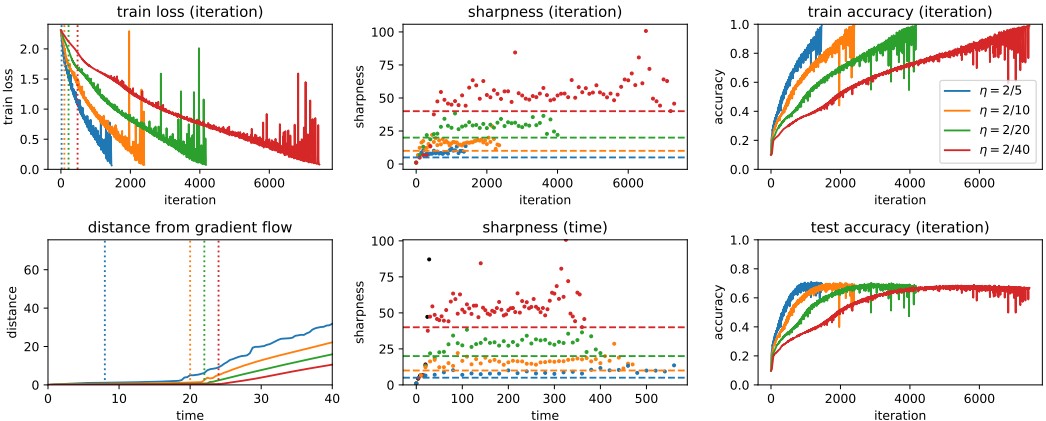

Figure 94: We train a **ResNet** to completion using **gradient descent** at a range of step sizes (see legend in the top right pane). Refer to the Figure 90 caption for more information.

# N EXPERIMENTS: MOMENTUM

This appendix contains systematic experiments for gradient descent with Polyak momentum and Nesterov momentum. Our aim is to demonstrate that the sharpness rises until reaching the maximum stable sharpness (MSS) given by equation 1, and then either plateaus just above that value, or oscillates around that value.

We experiment on a 5k-sized subset of CIFAR-10, using four architectures: a tanh fully-connected network (section N.1), a ReLU fully-connected network (section N.2), a tanh convolutional network (section N.3), and a ReLU convolutional network (section N.4). For each of these four architectures, we experiment with both the square loss (for classification) and cross-entropy loss. For each architecture and loss function, we experiment with both Polyak momentum at $\beta = 0.9$, and gradient descent with Nesterov momentum at $\beta = 0.9$. We run gradient descent at a range of several step sizes which were chosen by hand so that the MSS's are approximately spaced evenly. Note that for Polyak momentum with step size $\eta$ and momentum parameter $\beta = 0.9$, the MSS is $\frac{2+2\beta}{\eta} = \frac{3.8}{\eta}$. For Nesterov momentum with step size $\eta$ and momentum parameter $\beta = 0.9$, the MSS is $\frac{2+2\beta}{\eta(1+2\beta)} \approx \frac{1.35714}{\eta}$. We run gradient descent until reaching 99% accuracy.

We find that the sharpness rises until reaching the maximum stable sharpness (MSS) given by equation 1, and then either plateaus just above that value, or oscillates around that value. Sometimes these oscillations are rapid (e.g. Figure 96), sometimes they are a bit slower (e.g. Figure 103), and sometimes they are slow (e.g. Figure 109).

## N.1    FULLY-CONNECTED TANH NETWORK

In the leftmost plot, the vertical dotted line marks the iteration where the sharpness first crosses the MSS. (Note that unlike vanilla gradient descent, momentum gradient descent can sometimes cause the train loss to increase even when the algorithm is stable (Goh, 2017).)  In the middle plot, the horizontal dashed line marks the MSS.

### N.1.1    SQUARE LOSS

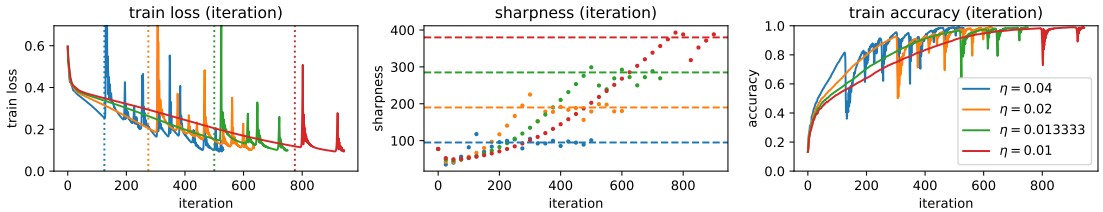

Figure 95: Gradient descent with **Polyak** momentum, $\beta = 0.9$.

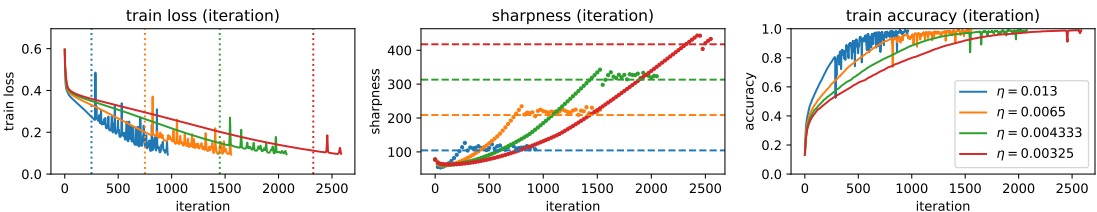

Figure 96: Gradient descent with **Nesterov** momentum, $\beta = 0.9$.

### N.1.2    CROSS-ENTROPY LOSS

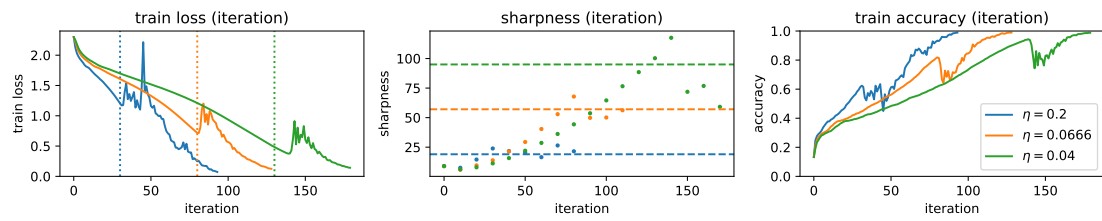

Figure 97: Gradient descent with **Polyak** momentum, $\beta = 0.9$.

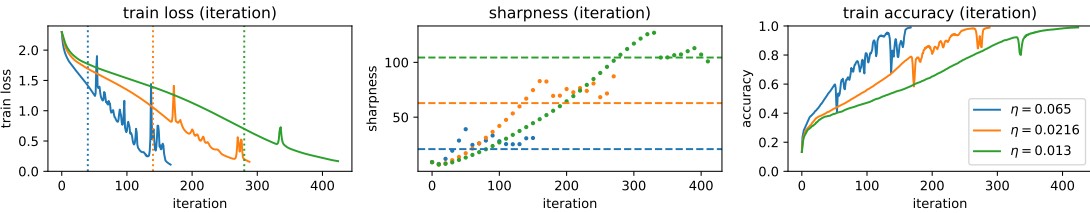

Figure 98: Gradient descent with **Nesterov** momentum, $\beta = 0.9$.

## N.2   FULLY-CONNECTED ReLU NETWORK

In the leftmost plot, the vertical dotted line marks the iteration where the sharpness first crosses the MSS. (Note that unlike vanilla gradient descent, momentum gradient descent can sometimes cause the train loss to increase even when the algorithm is stable (Goh, 2017).)

In the middle plot, the horizontal dashed line marks the MSS.

### N.2.1   SQUARE LOSS

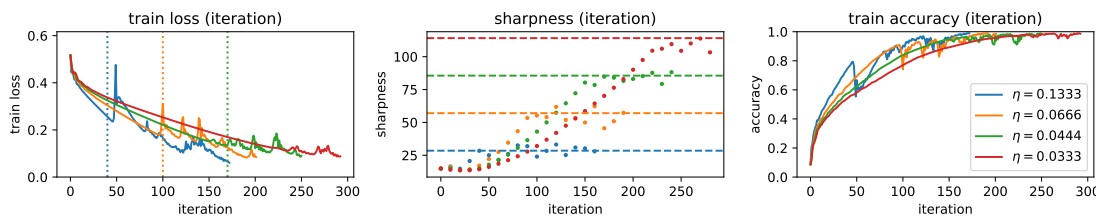

Figure 99: Gradient descent with **Polyak** momentum, $\beta = 0.9$.

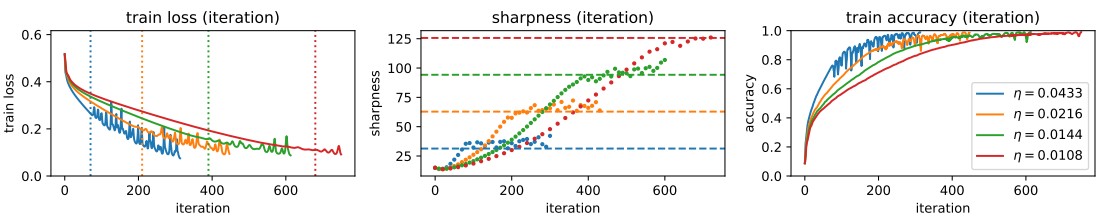

Figure 100: Gradient descent with **Nesterov** momentum, $\beta = 0.9$.

### N.2.2   CROSS-ENTROPY LOSS

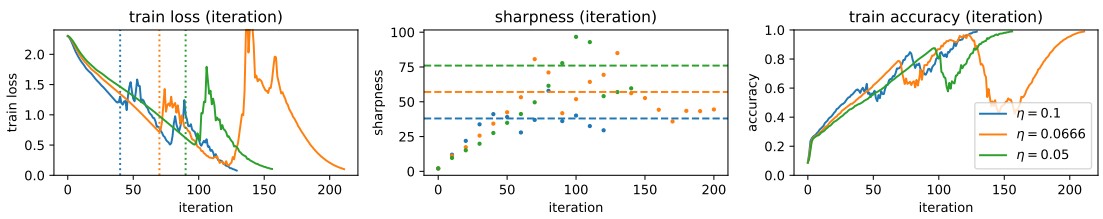

Figure 101: Gradient descent with **Polyak** momentum, $\beta = 0.9$.

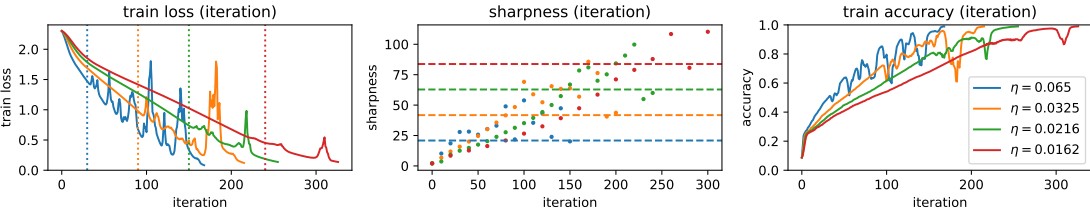

Figure 102: Gradient descent with **Nesterov** momentum, $\beta = 0.9$.

### N.3 CONVOLUTIONAL TANH NETWORK

In the leftmost plot, the vertical dotted line marks the iteration where the sharpness first crosses the MSS. (Note that unlike vanilla gradient descent, momentum gradient descent can sometimes cause the train loss to increase even when the algorithm is stable (Goh, 2017).)

In the middle plot, the horizontal dashed line marks the MSS.

#### N.3.1 SQUARE LOSS

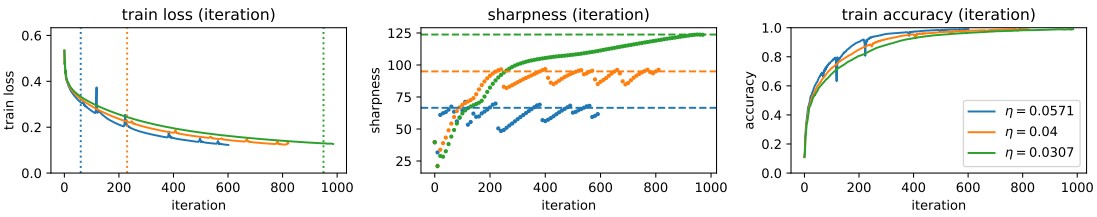

Figure 103: Gradient descent with **Polyak** momentum, $\beta = 0.9$.

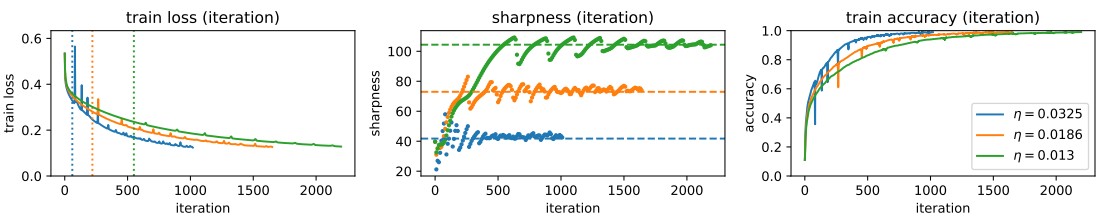

Figure 104: Gradient descent with **Nesterov** momentum, $\beta = 0.9$.

#### N.3.2 CROSS-ENTROPY LOSS

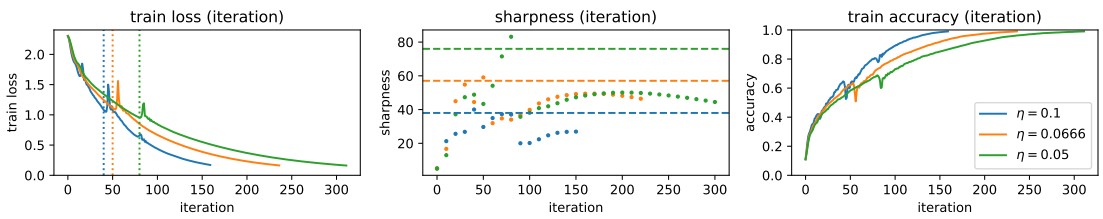

Figure 105: Gradient descent with **Polyak** momentum, $\beta = 0.9$.

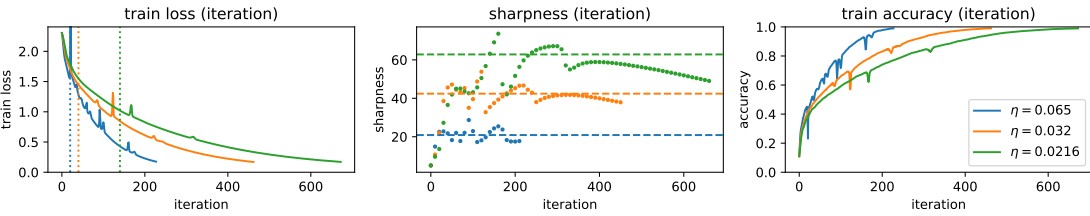

Figure 106: Gradient descent with **Nesterov** momentum, $\beta = 0.9$.

### N.4    CONVOLUTIONAL ReLU NETWORK

In the leftmost plot, the vertical dotted line marks the iteration where the sharpness first crosses the MSS. (Note that unlike vanilla gradient descent, momentum gradient descent can sometimes cause the train loss to increase even when the algorithm is stable (Goh, 2017).) In the middle plot, the horizontal dashed line marks the MSS.

#### N.4.1    SQUARE LOSS

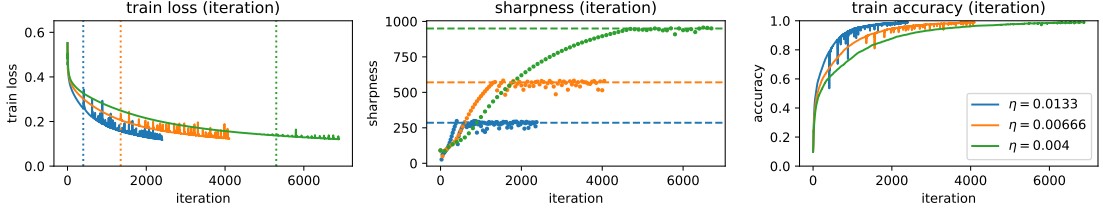

Figure 107: Gradient descent with **Polyak** momentum, $\beta = 0.9$.

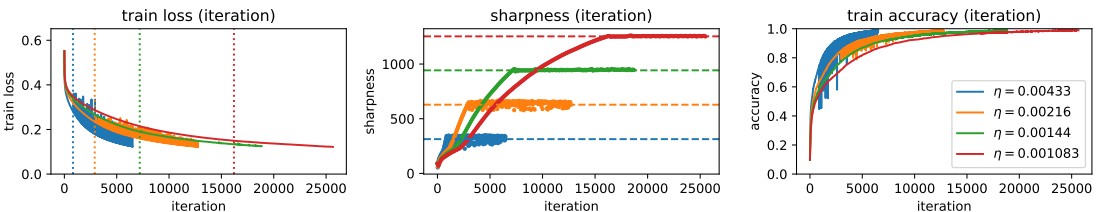

Figure 108: Gradient descent with **Nesterov** momentum, $\beta = 0.9$.

#### N.4.2    CROSS-ENTROPY LOSS

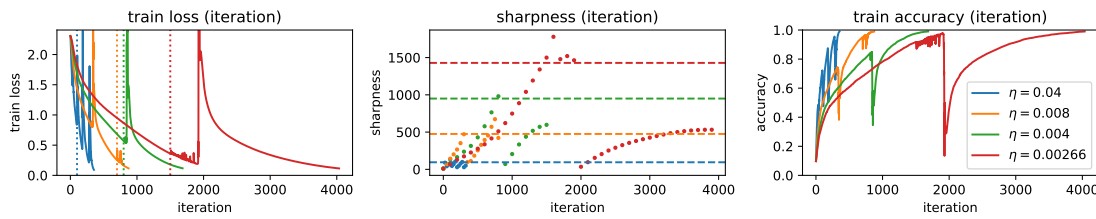

Figure 109: Gradient descent with **Polyak** momentum, $\beta = 0.9$.

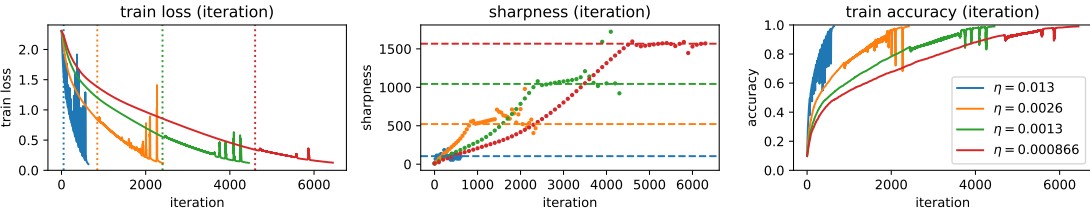

Figure 110: Gradient descent with **Nesterov** momentum, $\beta = 0.9$.

## O    EXPERIMENTS: LEARNING RATE DROP

In this appendix, we run gradient descent until reaching the Edge of Stability, and then we cut the step size. We will see that the sharpness starts increasing as soon as the step size is cut, and only stops increasing once gradient descent is back at the Edge of Stability (or training is finished). As a consequence of this experiment, one can interpret the Edge of Stability as a regime in which gradient descent is constantly "trying" to increase the sharpness beyond $2/\eta$, but is constantly being blocked from doing so. Our experiments focus on image classification on a 5k-sized subset of CIFAR-10. We study two architectures (a fully-connected tanh network and a convolutional ReLU network) and two loss functions (squared loss and cross-entropy loss).

## O.1 Fully-connected tanh network: square loss

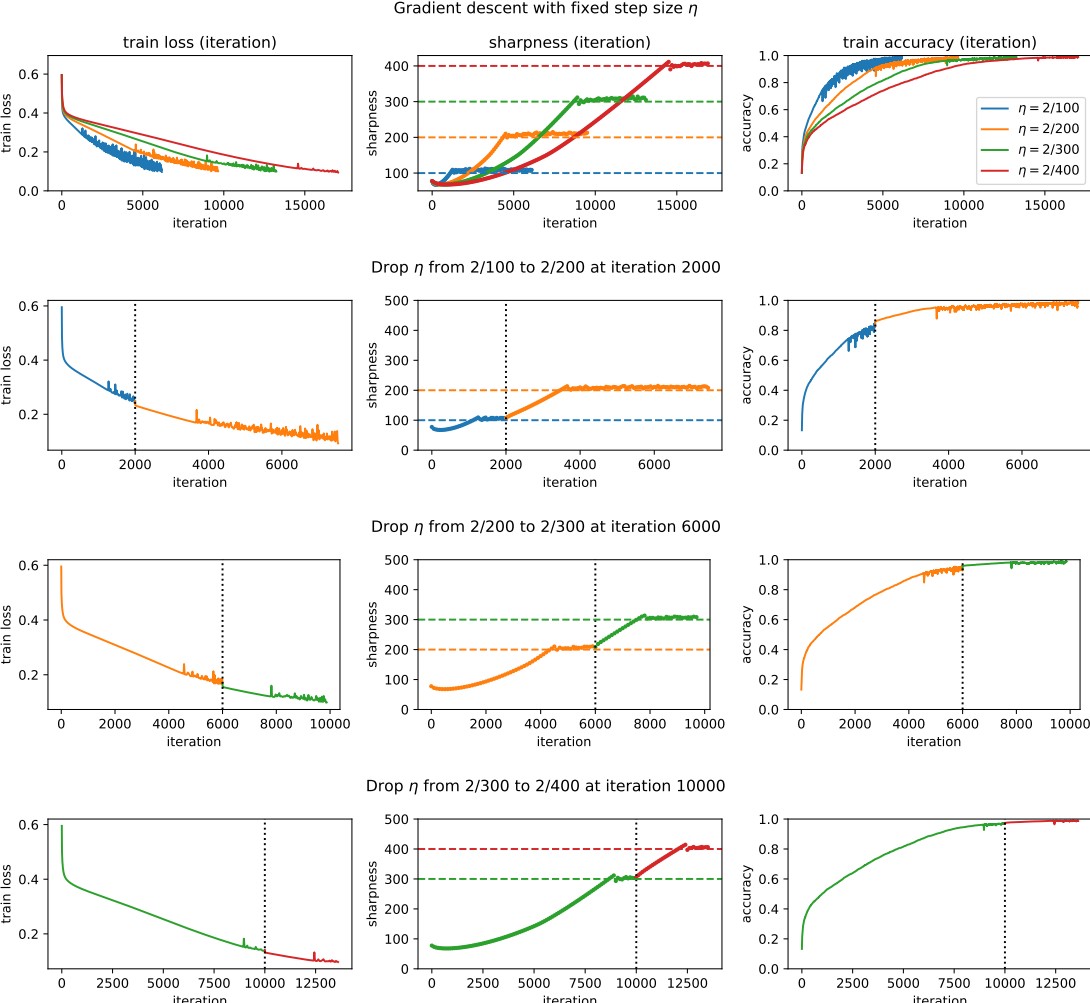

Figure 111: **Top row**: we train a network using gradient descent at a range of step sizes $\eta$ (see the legend in the right pane). **Bottom three rows**: Once gradient descent has reached the Edge of Stability, we cut the step size. The black vertical dotted line marks the iteration where the step size is cut. This iteration was chosen by hand, but was not cherry-picked. We use different colors (consistent with the legend) to plot the train loss, sharpness, and train accuracy before and after the learning rate drop. In the middle sharpness plot, the two horizontal lines mark the maximum stable sharpness $2/\eta$ for both the old and new step size. **Takeaway**: once we decrease the step size, the sharpness immediately starts to increase until it reaches the maximum stable sharpness $2/\eta$ for the new step size $\eta$ (or training finishes).

## O.2  FULLY-CONNECTED TANH NETWORK: CROSS-ENTROPY LOSS

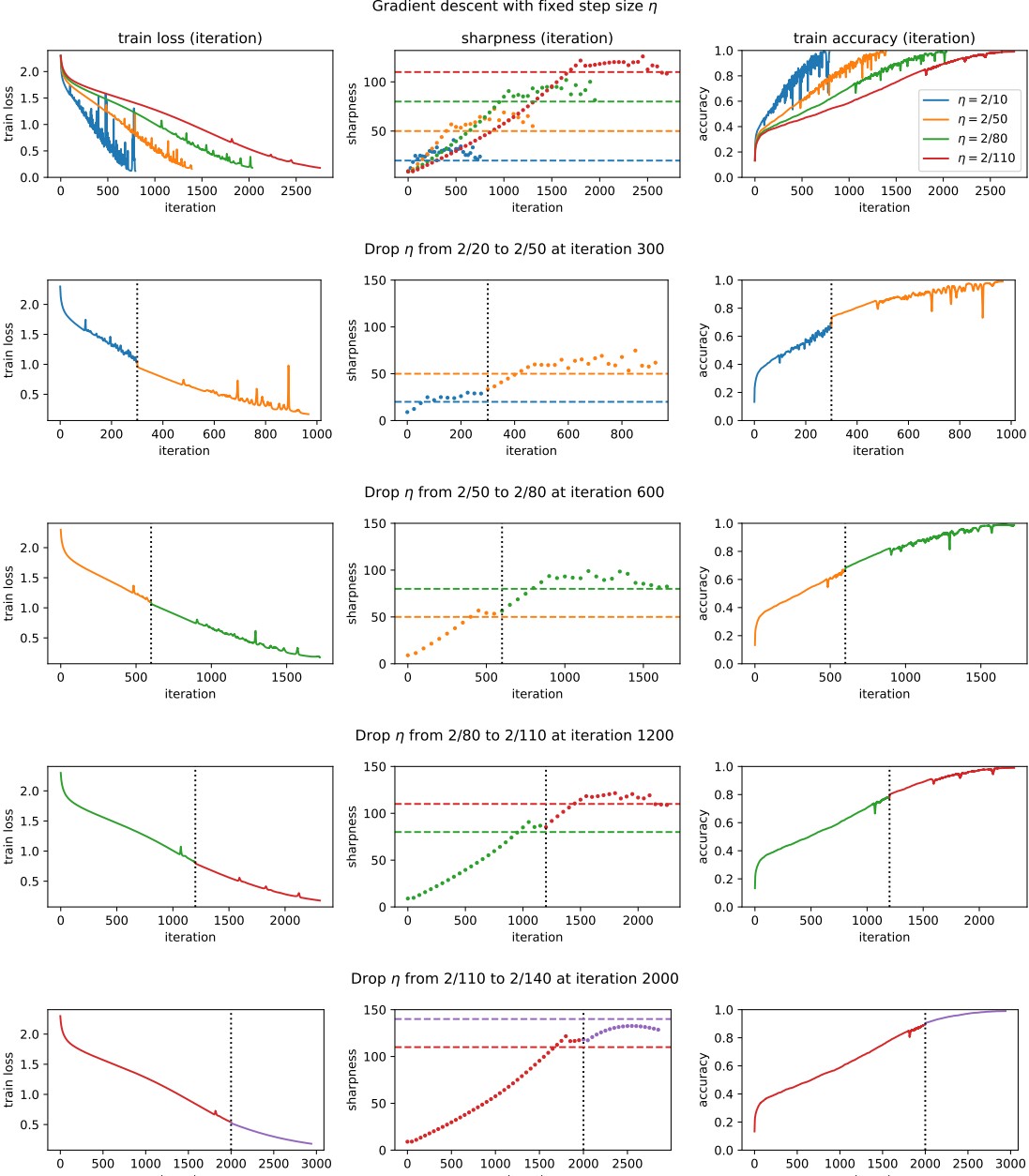

Figure 112: Refer to the caption for Figure 111.

## O.3 CONVOLUTIONAL RELU NETWORK: SQUARE LOSS

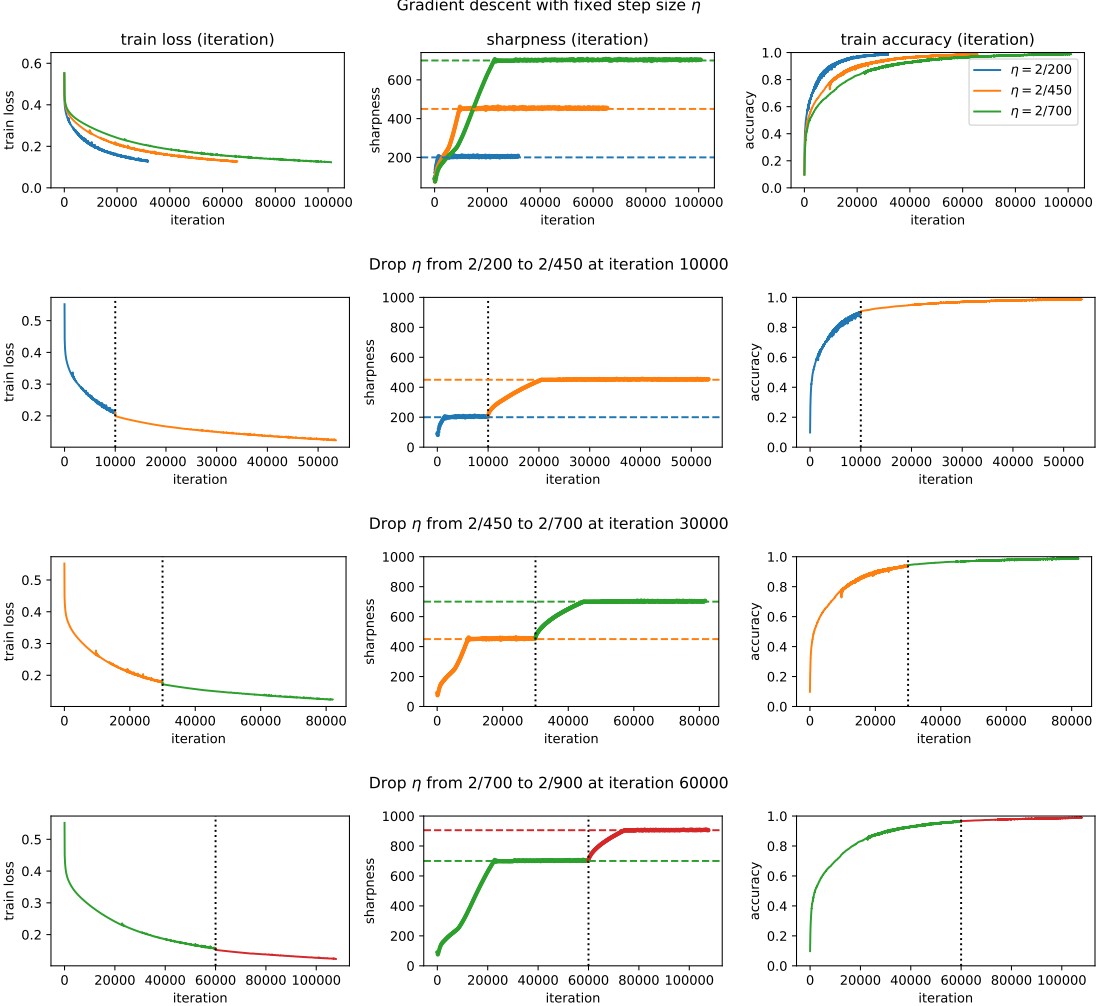

Figure 113: Refer to the caption for Figure 111.

## O.4 CONVOLUTIONAL RELU NETWORK: CROSS-ENTROPY LOSS

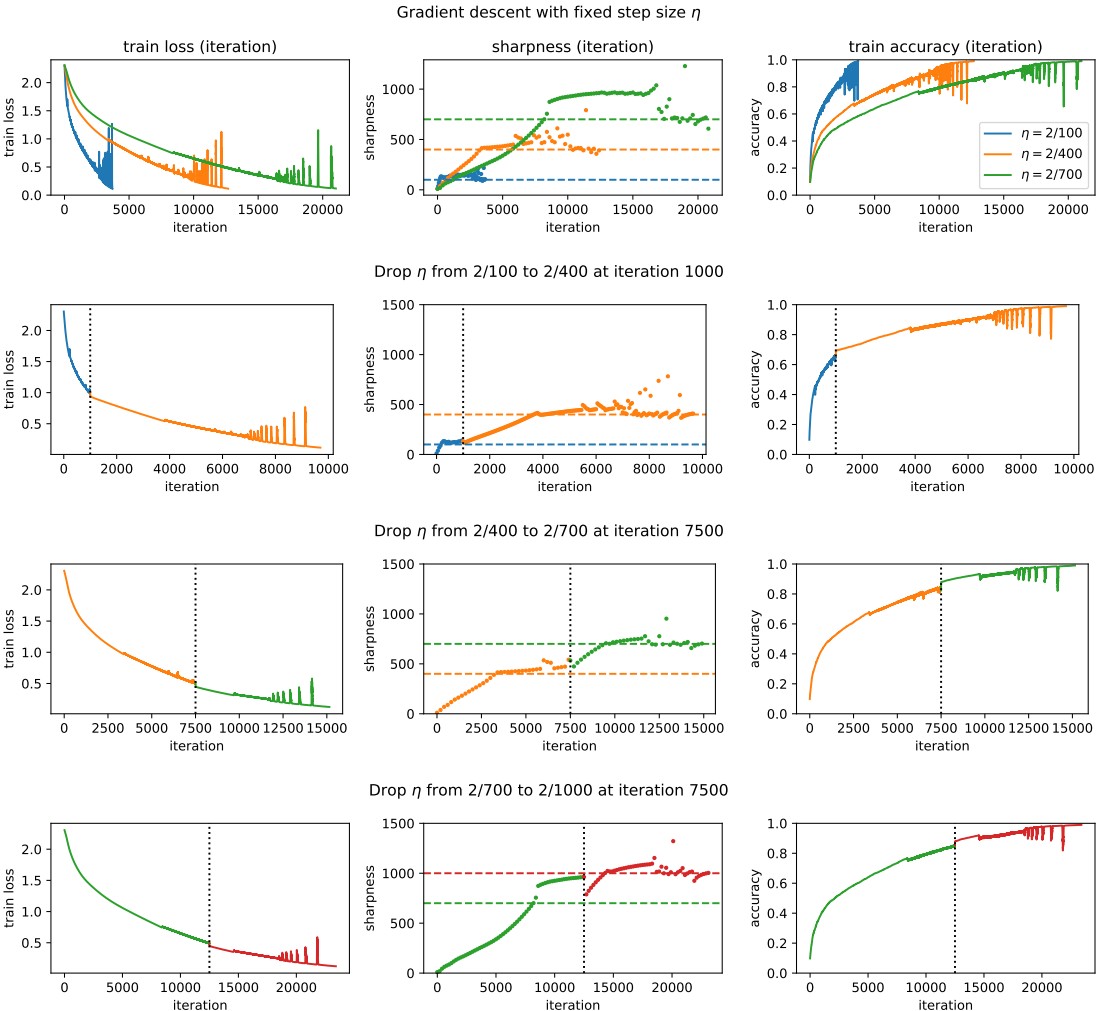

Figure 114: Refer to the caption for Figure 111.

## P    OTHER EIGENVALUES

Throughout most of this paper, we have studied the evolution of the maximum Hessian eigenvalue during gradient descent. In this appendix, we examine the evolution of the *top six* eigenvalues. While training the network from §3, we monitor the evolution of the top six Hessian eigenvalues. For each of cross-entropy loss and MSE loss, we train at four different step sizes. The results are shown in Figure 115. Observe that each of the top six eigenvalues rises and then plateaus. The precise details differ between MSE loss and cross-entropy loss. For MSE loss, each eigenvalue rises past $2/\eta$, and then plateaus just above that value. In contrast, for cross-entropy loss, some of the lesser eigenvalues plateau *below* $2/\eta$.

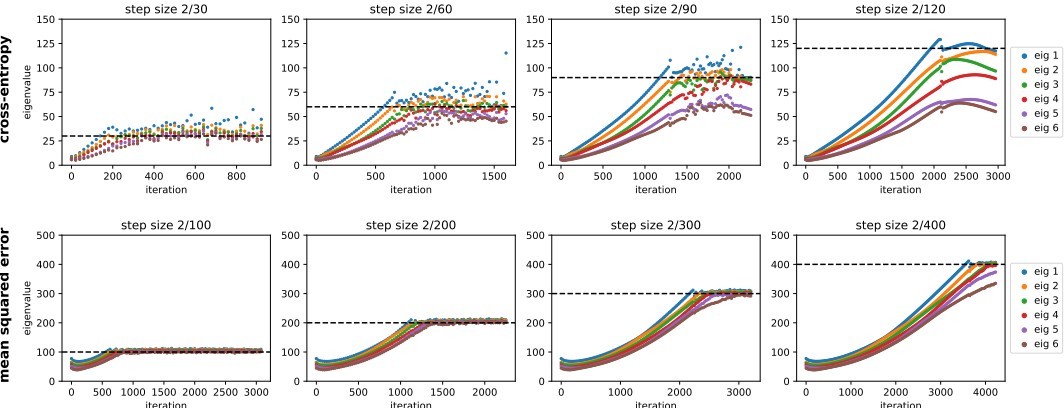

Figure 115: **Evolution of top six eigenvalues**. For both cross-entropy loss (**top**) and MSE loss (**bottom**), we run gradient descent at four different step sizes (columns), and plot the evolution of the top six eigenvalues (different colors; see legend).

