# OpenReview forum: "Gradient Descent on Neural Networks Typically Occurs at the Edge of Stability"
_ICLR.cc/2021/Conference — ICLR 2021 Poster_

### Official Review · AnonReviewer4 · 2020-10-27
**A very interesting observation about the complicated dynamical behavior of gradient descent in training neural networks.**

**Rating:** 8
**Confidence:** 5

**Review:**

Summary:

This submission numerically shows that during exploring the neural network landscape,  GD flow keeps increasing the sharpness.  As a result, GD with a fixed learning rate will exhibit two phases during the dynamics.  Denote by $\eta$ the fixed learning rate.  In the first phase, GD follows closely to the GD flow, and it finally converges to a region where the sharpness is roughly $2/\eta$.  Then, it transits into the second phase during which the sharpness hovers right at or above $2/\eta$. In the second phase, GD cannot increase the sharpness anymore due to the dynamical stability constraint. Thus, the authors name it the Edge of Stability phase.  What is interesting is that in the edge of stability phase, the loss is still decreasing steadily although not monotonically.

Pros:

I enjoy reading this submission. It is clearly written and the numerical evaluation is also sufficient.  To my best of knowledge, the observation that the edge of stability happens during the whole late phase of GD dynamics is new.  It reveals a very complicated dynamical behavior of GD for training neural networks, which has not been systematically investigated before.   Thus, I think this submission made a very important and original contribution to the understanding of GD dynamics  in deep learning.

Cons:

The relationship with the previous study on the dynamical stability of (S)GD is not sufficient discussed. In my opinion, just saying ''previous works have argued that the stability properties of optimization algorithms could potentially serve as a form of implicit bias in deep learning'' is obviously not precise and enough. A large number of numerical results in [1,2] already showed that the edge of stability happens for the convergent solutions, which implies that the edge of stability must happen at least in the very late phase of GD dynamics.   The new finds of this submission are that the edge of stability actually holds for a large portion of GD dynamics, which is very unexpected.   The authors should explicitly mention that the edge of stability was already observed in these previous works. Giving the right credit to the right references does not harm the contribution of this submission.  Especially, the jargon "Edge of stability" was first used in [1], and the authors even did not mention it.


[1] Giladi, Niv, et al. "At Stability's Edge: How to Adjust Hyperparameters to Preserve Minima Selection in Asynchronous Training of Neural Networks?." arXiv preprint arXiv:1909.12340 (2019).

[2] Wu, Lei, Chao Ma, and E. Weinan. "How sgd selects the global minima in over-parameterized learning: A dynamical stability perspective." Advances in Neural Information Processing Systems. 2018.

---

> ### Author Response · Authors · 2020-11-13
> **author response**
>
> Thanks very much for your review!  We agree that our Related Work section does not currently provide sufficient discussion on prior works pertaining to dynamical stability.  In the revision, when we have an extra page, we will be sure to discuss prior work on this subject in a much greater level of detail.  In response to your specific comments:
>
> > A large number of numerical results in [Wu et al (2018),  Giladi et al (2019)] already showed that the edge of stability happens for the convergent solutions
>
> **Wu et al (2018)**: Yes, as you point out, Wu et al (2018) previously observed that the sharpness of the convergent solution learned by gradient descent was not just *bounded by* $2 /\eta$ (as expected), but was, mysteriously, *approximately equal* to $2 / \eta$.  In the early stages of our research, this finding was very helpful to us: we'd observed "Edge of Stability" ourselves on a few architectures, but we weren't sure how general the phenomenon was.  The fact that it had also appeared in the experiments in Wu et al (2018) gave us confidence that the phenomenon might be quite general, which motivated us to investigate further.  We'd noted in earlier drafts of our paper that Wu et al (2018) made this observation, but lack of space forced us to condense the section on dynamical stability.  In the revision, we will make sure to note that Wu et al (2018) previously made this observation.
>
> **Giladi et al (2019)**:  Giladi et al (2019) did not observe the effect we call "Edge of Stability", neither during training nor at convergence.  That paper did argue that the convergent solution reached by gradient descent would have sharpness _less than or equal to_ $2 / \eta$, but did not  suggest or demonstrate that the sharpness would be at the very top of that allowable range.
>
> Perhaps you were referring to their Figure 4 (though if you had something else in mind, we’d greatly appreciate any pointers). In their Figure 4, they empirically validated that the maximum stable learning rate derived using a quadratic Taylor approximation also can be used to predict local divergence on the real neural network training objective.  Namely, they first trained a network to a low-loss iterate (they call it a "minimum") $\theta_0$ using some original learning rate $\eta_0$.  Then they computed the sharpness at $\theta_0$ — call this sharpness $a$.  However, they did *not* empirically examine the relationship between $a$ and $\eta_0$, as we do in our paper.  Instead, they demonstrated that they could use $a$ to accurately predict the learning rates at which gradient descent initialized at $\theta_0$ would escape that immediate neighborhood.  That is, they analytically computed the maximum stable learning rate for (asynchronous) gradient descent on a quadratic function with sharpness $a$, and they plotted this on the left side of Figure 4.  Then they empirically found the maximum learning rate at which gradient descent initialized at $\theta_0$ would remain in the neighborhood of $\theta_0$ (rather than immediately escaping), and they plotted this on the right side of Figure 4.  The point of this experiment was to demonstrate, both theoretically (left) and empirically (right), the dependence of gradient descent's stability properties on both the momentum parameter $m$ and the time lag $\tau$.
>
> > Especially, the jargon "Edge of stability" was first used in [1], and the authors even did not mention it.
>
> As we will clarify in the revision, our jargon "Edge of Stability" is indeed inspired by Giladi et al (2019).  That paper used this term to refer to the well-known fact that for gradient descent, there is a step size threshold for which step sizes over this threshold will trigger (local) divergence.  However, just to reiterate, their paper did not suggest or demonstrate our main contribution: that, no matter the step size, gradient descent always eventually navigates to a region where _that step size_ is on the edge of stability.

---

> > ### Comment · AnonReviewer4 · 2020-11-24
> > **Thanks  and extra suggestions**
> >
> > Dear Authors,
> >
> > Thanks for your reply. Here are some extra suggestions.
> >
> > In Section 4,  you said, "our findings hold for the more commonly-used cross-entropy loss, the only difference being that the sharpness starts to decrease towards the end of the training.".
> >
> > First, in general, the cross-entropy loss is not more commonly-used.  For regression problems, the quadratic loss is more commonly-used.
> >
> > You should add some explanations about why the sharpness decreases towards zero for the cross-entropy loss. My understanding is that the local quadratic approximations degenerate for the cross-entropy loss, whose Hessian vanishes at the global minima.  See Figure 4 in [Wu et al, 2018].

---

> > > ### Author Response · Authors · 2020-11-24
> > > **response**
> > >
> > > Thanks for your comments!
> > >
> > > We disagree with the reasoning expressed in Figure 4 in Wu et al (2018).  Though it's possible that we are mistaken, we don't see how the quadratic approximation for the logistic loss function breaks down when its input is large (which is what they presumably mean when they refer to minima of classification tasks).   What Wu et al (2018) have in mind here is scenarios like the function $f(x) = x^4$ around the origin, where the quadratic taylor approximation $f(x) \approx  f(0) + f'(0) x+ \frac{1}{2}f''(0) x^2 = 0$ is extremely inaccurate because the second derivative is small yet the other derivatives are large (so you lose a lot by truncating the Taylor expansion after two terms).  In contrast, while it's true that the second derivative of the logistic loss goes to zero as its input increases, we do not believe that the higher derivatives are sufficiently large that the quadratic Taylor approximation is inaccurate.
> > >
> > > In the revision, we will add an appendix with our explanation for why the sharpness goes to zero at the end of training for the cross-entropy loss.   The answer seems to be the following.  For simplicity, suppose that the task is binary classification rather than multiclass.  Denote the dataset as $\{(x_i, y_i)\} \subset \mathbb{R}^d \times \\\{-1, 1\\\}$, the network as $f_w: \mathbb{R}^d \to \mathbb{R}$ (with weights $w \in \mathbb{R}^p$) and the logistic loss as $\ell: \mathbb{R} \to \mathbb{R}$.   The training objective is $L(w) = \frac{1}{n} \sum_{i=1}^n \ell(y_i \\, f_w(x_i))$.
> > > By the Gauss-Newton approximation, the Hessian of the training objective is approximately $\nabla^2 L(w) \approx \frac{1}{n}  J^T H J$, where $J \in \mathbb{R}^{n \times p}$ are the network gradients $\frac{\partial f(x_i)}{\partial w} \in \mathbb{R}^p$ for all $n$ training examples stacked vertically,  and  $H \in \mathbb{R}^{n \times n}$ is a diagonal matrix whose element $(i, i)$ is $\ell''(y_i \\, f(x_i)))$, the second derivative of the logistic loss evaluated at the network's prediction for example $i$.  Now, when training with logistic/cross-entropy loss, once the network has correctly classified an example $i$, the network often tries to "game the loss function" by increasing the margin $y_i f(x_i)$.   The crucial fact is that $\ell''$ decreases as its argument increases.  Thus, as the network is increasing the margin of correctly classified examples,  the diagonal entries of $H$ decrease, so $\nabla^2 L(w)$ decreases.

---

> > > > ### Comment · AnonReviewer4 · 2020-11-24
> > > > **????**
> > > >
> > > > What you state is exactly what I imply. I didn't see any point why you disagree with me.  When the Hessian vanishes, the "quadratic approximation" is not quadratic anymore.

---

> > > > > ### Author Response · Authors · 2020-11-24
> > > > > **clarification**
> > > > >
> > > > > Thank you very much for engaging with us!  We are sorry for giving a confusing response.
> > > > >
> > > > > What we mean is: for the logistic loss, even though the Hessian vanishes as the input increases, the function itself is also vanishing, so the Taylor approximation isn't necessarily becoming inaccurate.  Here's a quick calculation where we try to assess the accuracy of the quadratic Taylor approximation to the logistic loss.
> > > > >
> > > > > The logistic loss is $\ell(x) = \log(1 + \exp(-x))$.
> > > > >
> > > > > The quadratic Taylor approximation of $\ell$ around $z$ is:
> > > > > $$\tilde{\ell}(x; z) = \log(1+ \exp(-z)) - \frac{1}{1+\exp(z)} (x-z) + \frac{1}{2} \frac{\exp(z)}{(1 + \exp(z))^2} (x-z)^2$$
> > > > >
> > > > > We need some way to quantify the local accuracy of the quadratic Taylor approximation around any point $z$.
> > > > > One idea is to look at the discrepancy between the real function and the quadratic Taylor approximation at positions $x=z-1$ and $x=z+1$.  This is arbitrary, and we are open to other definitions of accuracy.
> > > > >
> > > > > When we plot $z \mapsto | \tilde{\ell}(z-1; z) - \ell(z-1) |$, we see that this function drop to zero as $z \to \infty$, which indicates that the Taylor approximation becomes _more_ accurate as $z \to \infty$ (at least, by our arbitrary definition of accuracy).
> > > > >
> > > > > Likewise, when we plot  $z \mapsto | \tilde{\ell}(z+1; z) - \ell(z+1) |$, we see that this function also drops to zero as $z \to \infty$.
> > > > >
> > > > > Please let us know if you think our reasoning here is faulty.
> > > > >
> > > > > ----
> > > > >
> > > > > **Edit**: ah wait, we may have misunderstood your point.  When you wrote:
> > > > > > My understanding is that the local quadratic approximations degenerate for the cross-entropy loss, whose Hessian vanishes at the global minima.
> > > > >
> > > > > By "degenerate", do you mean just (1) that the Hessian vanishes, or did you also mean (2) the quadratic Taylor approximation becomes inaccurate?  We agree with (1), but not necessarily (2).  We interpreted Figure 4 in Wu et al (2018) to mean (2).

---

> > > > > > ### Author Response · Authors · 2020-11-24
> > > > > > **more intuition**
> > > > > >
> > > > > > Here's some more intuition.  For moderate to large $x$, the logistic loss $\ell(x) = \log(1+\exp(-x))$ is extremely well approximated by the function $f(x) = \exp(-x)$.  The derivatives of this function are $f'(x) = - \exp(-x)$, $f''(x) = \exp(-x)$, $f'''(x) = -\exp(-x)$, and so on.  Therefore, as $x$ gets large, it's true that $f''(x)$ vanishes, but so do all the higher-order derivatives (as well as the function itself).  In other words, as $x$ increases, every term in the Taylor series is decreasing, not just the quadratic term.  So, truncating the Taylor series after two terms doesn't necessarily become a worse approximation for large $x$.
> > > > > >
> > > > > > Note that it would be a different story if the third, fourth, fifth derivatives etc. stayed the same in magnitude for large $x$, but the second derivative decreased for large $x$.  In that case, the quadratic Taylor approximation would become increasingly inaccurate for large $x$.
> > > > > >
> > > > > > ----
> > > > > >
> > > > > > **Update**: here is another perspective on this same point.  For any $z$ (the value around which we Taylor-expand) , we can write the logistic loss $\ell(x)$ as a sum of $\tilde{\ell}(x; z)$ (i.e. the sum of the first three terms in the Taylor series) plus $\ell(x) - \tilde{\ell}(x ;z)$ (i.e. the sum of the higher-order terms in the Taylor series).  If you plot both $\tilde{\ell}(x; z)$ and $\ell(x) - \tilde{\ell}(x ;z)$ on the same axes, you will see that even for large $z$, the first quantity dominates the second, i.e. that the first three terms in the Taylor series dominate the higher-order terms.  This means that even for large $z$, the quadratic Taylor approximation is valid.  Intuitively, even though the quadratic term drops in scale as $z \to \infty$, so do each of the higher-degree terms.

---

> > > > > > > ### Comment · AnonReviewer4 · 2020-11-24
> > > > > > > **Thanks for your clarification**
> > > > > > >
> > > > > > > Dear Authors,
> > > > > > >
> > > > > > > I really appreciate your quick responses.  I generally only imply that the Hessian vanishes at the global minima. So it is observed that the sharpness decreases. This is consistent with your explanation.
> > > > > > >
> > > > > > > Second, I do think the quadratic approximation is a bad model to characterize the dynamical stability of GD  as the parameter approaching the global minima.  This does not imply that adding more higher-order terms can save us.  For example, the Tayler series of $f(t)=e^{-1/t}$ cannot capture the local behavior of $f$ around $t=0$ no matter how many terms are used.  Basically, we need other types of expansions to analyze the local behavior of it.
> > > > > > >
> > > > > > > Using the language of dynamical system,  the condition: $\eta<2/\lambda_1(H)$ is actually called the condition of linear stability. This condition becomes sharp only when the quadratic approximation can capture the "local" behavior of the objective function.  Apparently, there does not exist any quadratic function that can capture the limiting behavior of $ f(x)=e^{-x}$ as $x\to \infty$. In other words,  we need another notion of stability.

---

> > > > > > > > ### Author Response · Authors · 2020-11-24
> > > > > > > > **we agree!**
> > > > > > > >
> > > > > > > > Ah, got it -- thank you for clarifying.  We completely agree!

---

> ### Author Response · Authors · 2020-11-22
> **stability of SGD**
>
> By the way, regarding the dynamical stability of SGD, you may be interested in the point we made here to R2:  https://openreview.net/forum?id=jh-rTtvkGeM&noteId=KHKfRvNx0A.  It's very unclear how to model the stability of SGD.  Several papers use stability in expectation, but there are very simple examples of objective functions where SGD is unstable in expectation, yet stable with probability almost 1.

---

> > ### Comment · AnonReviewer4 · 2020-11-24
> > **My thoughts on the stability of SGD**
> >
> > Dear Authors,
> >
> > Thanks for sharing your thoughts about the stability of SGD. I agree with the difference between the stability in expectation and the stability in high prob. Here are some of my thoughts.
> >
> > 1. The stability in expectation can lead to stability in high prob, but not vice versa. The simple example you provide is a counterexample.
> >
> > 2. Even for stability in expectation, the condition derived in [Wu et al. 2018] is still only a sufficient condition in high dimension. This is due to that, in high dimensions, it is possible that the eigenvector corresponding to the largest eigenvalue of H_t can be in the null space of H_{t+1}.  However,  the experiments in [Wu et al. 2018] actually show that the condition they derived is quite "empirically necessary" for neural networks.

---

> > > ### Author Response · Authors · 2020-11-24
> > > **thanks**
> > >
> > > Yes, we completely agree with those points!

---

### Official Review · AnonReviewer2 · 2020-10-28

**Rating:** 8
**Confidence:** 4

**Review:**

This work identifies a new empirical phenomenon in the training dynamics of deep nets: when trained with full-batch GD, the curvature of the train loss increases up to a critical value of 2/(step size), at which point it plateaus for the remainder of training.
This phenomenon is demonstrated robustly for networks trained with MSE loss, across various architectures and datasets, and a slightly weaker version of this holds for cross-entropy loss as well.
This work contributes to our understanding of deep network dynamics -- it is a precise and apparently robust phenomenon that was surprisingly not noticed before (perhaps because of the requirement of GD vs SGD).
In terms of impact: This work will be instructive for DL optimization theory, since it points out that certain assumptions which are usually made in theoretical works (e.g. step size << curvature) are far from true in practice -- moreover, it guides theory towards more realistic assumptions.
It may also have later impact in practice, by leading to a better understanding of the interaction between optimization algorithm, step size, and architecture. Thus I recommend acceptance.

Weaknesses and desired clarifications:
- It should be mentioned more prominently that these results are primarily for networks trained with MSE loss -- and that a similar but weaker phenomena holds for cross-entropy loss.
- Why is the main example in Section 3 given for a non-standard network for CIFAR-10? A 2-layer MLP with ELU activation. Why not a standard network with standard activation? (VGG-11 or ResNet-18, etc).
- The distinction between SGD and GD seems crucial for this phenomenon, so more discussion would be good. In particular, as noted in the related works, some papers using SGD claim an opposite effect. This is especially important to clarify since SGD is most often used in practice. If time allows, experiments with increasing batch size could shed light on the importance of GD vs SGD.
- The Related Works is currently written as an account of what previous works do *not* do, as opposed to what they do. It would help contextualize this work to relate it to prior works which are consistent (or inconsistent) with this phenomena -- especially works studying the Hessian of deep nets. Some of the mechanisms proposed in prior works (eg Lewkowycz et al 2020 and works on deep linear networks) may also be helpful to understand the phenomena in this work.



Comments which do not affect the score:

- I wonder if you have measured the 2nd eigenvalue during training as well? In particular, after the 1st eigenvalue has saturated at 2/eta, does the 2nd eigenvalue also "progressively sharpen" up to 2/eta ? (And so on for later eigenvals).
- I am glad to see the experiments on deep linear networks, it suggests that it may be possible to theoretically understand this phenomenon in such simple settings. This would be a nice topic for future work.

---

> ### Author Response · Authors · 2020-11-13
> **response to "it should be mentioned more prominently that these results are primarily for networks trained with MSE loss"**
>
> Thank you very much for all of your insightful comments!
>
> Here we respond to:
>
> > It should be mentioned more prominently that these results are primarily for networks trained with MSE loss -- and that a similar but weaker phenomena holds for cross-entropy loss.
>
> We’d first like to clarify that your comment applies to just one of the results in our paper.  Our paper has essentially three main results:  (1) negative results for optimization theory, (2) a new qualitative characterization of the behavior of gradient descent (that GD is constantly "trying to" increase the sharpness, but is constantly being blocked from doing so), and, finally, (3) "a simple rule that can numerically predict, to respectable accuracy, the value of" the sharpness during gradient descent training.  This simple rule is that "the sharpness hovers right at, or just above, the value $2 / \eta$."  Your comment applies to Result (3), but not (1) or (2).
>
> Regarding Result (3), you correctly observe that for some networks/losses, the gap between the sharpness and the value $2 / \eta$ is minuscule (e.g. Figure 3, or the square loss networks in Figure 6), while for other networks/losses, the gap between the sharpness and the value 2 / eta is small yet non-miniscule (e.g. Figure 1, or the cross-entropy networks in Figure 6).  We tried to choose wording -- ”just above” -- that applies to both cases, the only difference being one of degree, i.e. of how big is “just above.”  If you think that different wording would be better than “the sharpness hovers right at, or just above, the value $2/\eta$”, then we would be happy to make a change.  Additionally, in section 3 we will insert a sentence noting that the gap between the sharpness and the value $2/\eta$ is not always as miniscule as in Figure 3.
>
> As you point out, for the networks in Figure 6, the loss function makes a difference: for MSE loss, the gap is minuscule, while for cross-entropy loss, the gap is non-minuscule.  In our experience, this cross entropy-vs.-MSE pattern is true for smallish networks like the ones in Figure 6, but not always for big complicated architectures like VGG.  When training big complicated architectures like VGG, we sometimes observe that the gap is non-minuscule even when training with the MSE loss.  (In other words, the MSE version of Figure 1 looks much like Figure 1.) In the case of big vs. small, we think this is because for big, complicated networks like VGG, the training objective is just generally less well-behaved, so the quadratic Taylor approximation is not as accurate over long distances.
>
> Ultimately,  while it would definitely be preferable if the gap were always minuscule (as in Figure 3, or the top row of Figure 6), we think that even the level of agreement exhibited in Figure 1 is noteworthy, considering that little is known quantitatively about the neural network training process.

---

> > ### Comment · AnonReviewer2 · 2020-11-22
> > **response**
> >
> > Thank you for the detailed response.
> > Can you elaborate on the important of GD vs SGD for your results?
> >
> > In particular, does this difference explain why [Jastrzebski et al.] did not identify the 2/eta threshold as universal in their experiments?
> > It is my understanding that [Jastrzebski et al.] performs similar experiments as the current work, except using SGD instead of GD (and using cross-entropy instead of MSE). So I would like to understand why this 2/eta was not observed in previous works.

---

> > > ### Author Response · Authors · 2020-11-22
> > > **SGD**
> > >
> > > Thanks for your astute questions!  Your understanding of Jastrzebski et al. (2020), and its relation to our paper, is correct.  We believe the reason the $2/\eta$ rule is not already known to the literature is because no one has studied the evolution of the sharpness during _full-batch_ gradient descent.
> > >
> > > **What happens to the sharpness during SGD?**  During SGD with _very large_ batch sizes, the sharpness behaves just like it does during full-batch gradient descent, i.e. it equilibrates just above $2/\eta$.   For SGD more generally, the sharpness along the optimization trajectory tends to be: (1) lower when the learning rate is larger (note that the $2/\eta$ rule for full-batch gradient descent is a special case of this), and (2) lower when the batch size is smaller.  These facts have been previously reported in the literature, and we have observed them ourselves too.  **However**, during SGD, the sharpness sometimes does not equilibrate at any value, much less one that can be numerically "predicted" based on the hyperparameters.  For example, in Figure 5(a) of Jastrzebski et al. (2020), you can see that the sharpness gradually drifts downwards during training.  (This figure is in log scale, but the drift would be more dramatic if plotted on linear scale.)  While that figure was made with cross-entropy loss, we have ourselves seen that this downward drift also occurs with square loss too.  So, it empirically does not seem like the sharpness during SGD will obey any simple rule like it does during GD.
> > >
> > > **Why does the sharpness equilibrate to a fixed (and predictable) value for GD, but not SGD?**  We believe the answer is: for GD, the sharpness is wholly determinative of stability, whereas for SGD, the sharpness is "correlated" (we use that work in a non-rigorous sense, not a formal statistical sense) with stability, but is not wholly determinative of stability.  To quadratic order, gradient descent is completely stable if the sharpness is less than $2/\eta$, and completely unstable if the sharpness is greater than $2/\eta$.  So, given that progressive sharpening occurs, we believe that it makes intuitive sense that the sharpness would end up hovering approximately at largest stable sharpness of $2/\eta$.  On the other hand, while it's not completely clear how best to model "stability" for SGD (see below), even the most popular contender --- stability in expectation -- does not depend directly on the sharpness, unless you make an additional strong assumption (see below).   Therefore, there is no reason why the sharpness would equilibrate at some  "maximum stable sharpness" for SGD--- the sharpness just isn't (by itself) an important stability metric for SGD.
> > >
> > > **If the sharpness doesn't directly determine the stability of SGD, why does it vary in a consistent way as a function of the learning rate and batch size?**  In other words: during SGD, why does the sharpness tend to be lower when the learning rate is larger or the batch size is smaller?   We think that answering this question was one of the main motivations for Jastrzebski et al. (2020).  Their answer was: if you (1) model the stability of SGD in a certain way ("stability in expectation"), and (2)  make the strong assumption that the Hessian shares a leading eigenvector with the covariance matrix of per-example gradients, then the stability of SGD, in expectation, depends directly on the sharpness.  Their paper did not empirically verify either of these two assumptions.  On the one hand, we believe that the reason  why the sharpness does indeed depend on the learning rate and batch size is because these two assumptions are _partially_ true; on the other hand we believe the reason why the sharpness during SGD does not seem to satify a simple formula (like it does during GD) is because these two assumptions are _not fully_ true.  (We discuss "stability in expectation" more below.)
> > >
> > > **Might the "sharpness $\approx 2/\eta$" rule generalize somehow to SGD?**   We hope and hypothesize that SGD may indeed satisfy a similar rule, involving some statistic _that is different from the sharpness_.   For example, suppose that SGD is unstable if and only if STATISTIC (STATE) $\le$ THRESHOLD.  (For full-batch GD, STATE is the iterate itself, STATISTIC is the sharpness, and THRESHOLD is $2/\eta$ for vanilla GD.) We'd then find it believable that, in some generalization of progressive sharpening, STATISTIC of STATE would tend to rise during training, and that as a consequence, STATISTIC of STATE would rise until hovering right at, or just above, THRESHOLD.  Now, there are a lot of unanswered questions here: (a) how to model "stable/unstable" in the case of a random algorithm like SGD, (b) what is STATE, (c) what is STATISTIC?
> > >
> > > _continued below_

---

> > > > ### Author Response · Authors · 2020-11-22
> > > > **SGD (continued from above)**
> > > >
> > > > **How to model stability for SGD?**  The stability properties of full-batch gradient descent are very simple: if the sharpness is less than $2/\eta$, then GD is deterministically stable, whereas if the sharpness is greater than $2/\eta$, then GD is deterministically unstable.   On the other hand, SGD is a random algorithm, and so the notion of stability is much, much more complicated.  One natural idea, which has been used several times in the recent literature [1, 2, 3], is to consider stability _in expectation_.  So, SGD is considered unstable if _in expectation_ it diverges.  However, nobody has verified whether the actual behavior of SGD on neural networks is well-modeled by its stability in expectation, and we are in fact deeply skeptical.  The reason why we are skeptical is that even on what is arguably the simplest toy objective function, the stability properties of SGD in expectation are a very poor model for the actual stability of the algorithm.  Namely, consider optimizing the objective function $f(w) = \frac{1}{2} \mathbb{E}_{x \sim \mathcal{N}(0, 1)}[(wx)^2]$ using SGD with minibatch size 1.  You can think of this as a 1-dimensional linear regression problem where the optimal weight vector $w^*$ is 0; it is the simplest objective function we could think of.
> > > >
> > > > On this objective function, the SGD update is $w_{t+1} = (1 - \eta x^2_t) w_t$ with $x_t \sim \mathcal{N}(0, 1)$, so we can write a closed-form expression for the SGD iterate at time $t$:
> > > > $$w_{t} = w_0 \prod_{i=1}^t (1 - \eta x^2_i)$$ where each $x_i$ is i.i.d from $\mathcal{N}(0, 1)$.  Thus, $w_{t}^2 =  w_0^2 \prod_{i=1}^t (1 - \eta x^2_i)^2$ and
> > > > $$\mathbb{E}[w_t^2] = w_0^2 \prod_{i=1}^t \mathbb{E}[(1- \eta x^2_i)^2]  = (1 - 2 \eta + 3 \eta^2)^t w_0^2$$ where in the last step we used the distribution of $x$ to simplify.
> > > >
> > > > Therefore, on the one hand, if $|1 - 2 \eta + 3 \eta^2| < 1 \iff \eta < 2/3$, then $\mathbb{E}[w^2_t] \to 0$  as $t \to \infty$, meaning that $\{w_t\}$ will converge to the optimum of 0.  But on the other hand, if $\eta > 2/3$, then $|1 - 2 \eta + 3 \eta^2| > 1$, so $\mathbb{E}[w^2_t] \to \infty$  as $t \to \infty$, i.e. SGD will diverge in expectation.
> > > >
> > > > However, the interesting thing is that if you use a computer to simulate SGD on this objective function, you will see that for a range of learning rates larger than $2/3$, SGD does not diverge at all, but rather converges to the optimum.  The "trick" is that for these learning rates $\eta$, even though $\mathbb{E}[w^2_t]$ blows up _in expectation_, the random variable $w^2_t$ does not concentrate around its expectation.  Instead, the distribution of the random variable $w^2_t$ is something like:
> > > > $$ w^2_t = \begin{cases}
> > > > \text{basically zero} &\mbox{ with probability } \text{ almost 1} \\\\
> > > > \text{extremely large number} &\mbox{ with probability } \text{ almost 0 } \end{cases} $$
> > > >
> > > > So, even though $ w^2_t $ for large $t$ is astronomically large number _in expectation_, it is basically zero with probably almost 1.  In other words, SGD on this toy objective function is basically a real-life version of those counterexamples that are taught in math classes that highlight the subtle differences between various kinds of convergence --- SGD on this toy objective converges with high probability, but diverges in expectation.
> > > >
> > > > In sum, the question of how to even measure "stability" for random algorithms like SGD is unsettled.
> > > >
> > > >
> > > > [1] Lei Wu, Chao Ma, and E Weinan. How sgd selects the global minima in over-parameterized learning:
> > > > A dynamical stability perspective. In Advances in Neural Information Processing Systems, pp.
> > > > 8279–8288, 2018.
> > > >
> > > > [2] Stanislaw Jastrzebski, Maciej Szymczak, Stanislav Fort, Devansh Arpit, Jacek Tabor, Kyunghyun
> > > > Cho*, and Krzysztof Geras*. The break-even point on optimization trajectories of deep neural
> > > > networks. In International Conference on Learning Representations, 2020.
> > > >
> > > > [3] Niv Giladi, Mor Shpigel Nacson, Elad Hoffer, and Daniel Soudry. At stability’s edge: How to adjust
> > > > hyperparameters to preserve minima selection in asynchronous training of neural networks? In
> > > > International Conference on Learning Representations, 2020.
> > > >
> > > > **How to define "STATE" for SGD?** _This is a follow-up to the last point from the parent post._ It's not even clear how to define "STATE" for SGD.  For full-batch gradient descent, STATE is an iterate, but for SGD, the appropriate notion might be a distribution.  For example, we can imagine a scenario where SGD is "stably" ensconced in some stationary distribution, yet is "unstable" around each point in parameter space that it visits.  By this, we mean that that if dropped at any of those points, it would immediately leave that point and go far away (which is how "stability" is defined for deterministic GD); yet the overall distribution is "stable", in that SGD doesn't leave the stationary distribution.

---

> > > > > ### Comment · AnonReviewer2 · 2020-11-22
> > > > > **response, and citations**
> > > > >
> > > > > Thank you for the response.
> > > > >
> > > > > Note that Wu et al (https://papers.nips.cc/paper/2018/file/6651526b6fb8f29a00507de6a49ce30f-Paper.pdf)
> > > > > has related experiments showing that the curvature at the end of training reaches approx 2/eta in some settings.
> > > > > See Table 2.
> > > > > This should be properly discussed/cited in your work.

---

> > > > > > ### Author Response · Authors · 2020-11-22
> > > > > > **re: citation**
> > > > > >
> > > > > > Yes, we completely agree with you that in the revision, we need to credit Wu et al (2018) with noticing that the sharpness was $2/\eta$.  R4 made this point in their original review, and we responded to them here: https://openreview.net/forum?id=jh-rTtvkGeM&noteId=K7-PewUyI_
> > > > > >
> > > > > > update: the OpenReview link may not be working, so we will copy-and-paste what we said to R4:
> > > > > > > **Wu et al (2018)**: Yes, as you point out, Wu et al (2018) previously observed that the sharpness of the convergent solution learned by gradient descent was not just *bounded by* $2 /\eta$ (as expected), but was, mysteriously, *approximately equal* to $2 / \eta$.  In the early stages of our research, this finding was very helpful to us: we'd observed "Edge of Stability" ourselves on a few architectures, but we weren't sure how general the phenomenon was.  The fact that it had also appeared in the experiments in Wu et al (2018) gave us confidence that the phenomenon might be quite general, which motivated us to investigate further.  We'd noted in earlier drafts of our paper that Wu et al (2018) made this observation, but lack of space forced us to condense the section on dynamical stability.  In the revision, we will make sure to note that Wu et al (2018) previously made this observation.

---

> ### Author Response · Authors · 2020-11-13
> **response to "why is the main example given for a non-standard activation function (ELU)"**
>
> Here, we address your question:
>
> > Why is the main example in Section 3 given for a non-standard network for CIFAR-10? A 2-layer MLP **with ELU activation**. Why not a standard network with **standard activation?** (VGG-11 or ResNet-18, etc).
>
> Specifically, in this comment, we explain why we used ELU rather than a more standard activation like ReLU or tanh. In a separate comment, we explain why we use a fully-connected network rather than VGG.
>
> **Why didn't we use ReLU?**  While our findings also hold for ReLU (as can be seen from our experiments section), ReLU is not differentiable, so the Taylor approximation of the training objective is not well-defined everywhere.  While the $2/\eta$ rule seems to hold regardless, we didn't want to highlight a ReLU network for our running example.
>
> **Why didn't we use tanh?**  The short answer is that in the submission, we chose to make a certain point using argument-by-visualization, and the visualization happened to be much clearer with ELU rather than tanh activations.  For the revision, we now have a more rigorous way of making that same point (which does not rely on visualization), and therefore for the revision, we will use tanh activations in this experiment rather than ELU, as we agree with you that ELU is pretty non-standard.
>
>   Here is the long answer.  One finding of our paper is that until reaching the Edge of Stability, gradient descent at different step sizes follows the same path (the gradient flow trajectory), just at different speeds.  In section 3, we demonstrated this point in a less-than-ideal way: by plotting the train loss by time, and noting that the train loss curves coincide exactly until the time when the sharpness hits 2/eta, yet split from each other after this time.  The reason why we used tanh rather than ELU is that for the tanh version of Figure 3(b), the train loss curves weren't *that* different from each other after reaching the Edge of Stability, so it was hard to tell from the tanh version of Figure 3(b) that the gradient descent trajectories split from one another after reaching the Edge of Stability.  For the revision, we have a more rigorous way of making our point that gradient descent follows the gradient flow trajectory until the sharpness hits 2/eta, which does not involve the Figure 3(b) visualization.  In particular, we are using Runge-Kutta to numerically integrate the gradient flow ODE, and we are plotting the Euclidean distance between the gradient descent trajectory and the Runge-Kutta trajectory.  This way, even for the tanh network, we can clearly demonstrate our point that gradient descent tracks the gradient flow trajectory until reaching the point where the sharpness hits $2/\eta$.

---

> ### Author Response · Authors · 2020-11-13
> **response to "why is the main example given for a non-standard network"**
>
> Here, we address your question:
>
> > Why is the main example in Section 3 given for a **non-standard network for CIFAR-10? A 2-layer MLP** with ELU activation. Why not a **standard network** with standard activation? **(VGG-11 or ResNet-18, etc)**.
>
> Specifically, in this comment, we explain why we use a fully-connected network rather than a standard architecture like VGG.  In a separate comment, we explain why we used ELU over a more standard activation like ReLU or tanh.
>
> There are three reasons:
>
>   1.  We felt that it was important to make clear that the effects we study (especially progressive sharpening) occur for both simple networks (like the one we used in this Figure) and complicated modern networks like VGG/ResNet.  On the one hand, if in the main text we had only demonstrated progressive sharpening on simple networks, then readers might think "maybe modern SOTA architectures don't exhibit progressive sharpening, and moreover, maybe that is why they are SOTA."  On the other hand, if in the main text we had only demonstrated progressive sharpening on complicated modern networks, then readers might think "maybe progressive sharpening only occurs due to architectural feature XYZ that is present in VGG."  Thus, we wanted to prominently demonstrate progressive sharpening for both simple networks and modern networks.  In Figure 1 we used a complicated modern network, and thus for Figure 3 we wanted to use a simple network.
>
>   2.  (This is a follow up to our other comment that is in response to your question on CE vs. MSE loss.)  In Figure 1, we already presented a numerical example for which there was a non-miniscule gap between the sharpness and $2 /\eta$, so for Figure 3 we wanted to present a numerical example in which the gap was miniscule.  This is why we used a smallish network, and also why we used MSE loss rather than CE.  In the revision, we will add a line to section 3 explicitly noting that the gap is sometimes non-miniscule, like in Figure 1.
>
>   3.  We wanted to prominently acknowledge that it is *possible* to train networks to completion at "stable step sizes", i.e. step sizes so small that gradient descent never enters the Edge of Stability -- our argument is just that these stable step sizes are so small/suboptimal that you would never use them in practice.  For standard architectures like VGG/ResNet on the full CIFAR-10 dataset, the only architecture where it was even feasible for us to train to completion at a stable step size was a VGG network with BatchNorm -- and even there, that experiment still took about a week to run (since the step size needed to be so small).  However, we didn't want to discuss BatchNorm until later in the paper.  Unfortunately, for the VGG network *without* BatchNorm (featured in Figure 1), training to completion at a stable step size was simply not a feasible experiment to run.  For this reason, we decided to use a 5k subset of CIFAR-10 rather than the full CIFAR-10 dataset.  The smallish network was adequately sized for this small dataset (i.e. we had no problem training to 100% accuracy).  In retrospect, we maybe could have used a standard architecture (e.g VGG) on the CIFAR-5k dataset, but that still would have been a "non-standard" training setup due to the use of CIFAR10-5k rather than CIFAR10.

---

> ### Author Response · Authors · 2020-11-13
> **response to misc. comments**
>
> Here, we respond to the remaining comments.  (Additionally, we are planning to soon upload more commentary on SGD, since multiple reviewers asked for more information about SGD.)
>
> > In particular, as noted in the related works, some papers using SGD claim an opposite effect.
>
> The only paper we are aware of which showed an opposite effect is Ghorbani et al. (2019), who found that a learning rate drop caused the sharpness to decrease.  A high priority of ours for the revision is to try to confirm our suspicion that this contradictory observation is due to their use of SGD rather than GD.  Ideally, we would reproduce their observation in SGD and then show that it goes away as the batch size is increased.
>
> > Some of the mechanisms proposed in prior works (eg Lewkowycz et al 2020 and works on deep linear networks) may also be helpful to understand the phenomena in this work.
>
> We completely agree.  There is potentially a deep connection between the "Edge of Stability" regime, and the "catapult" effect studied in Lewkowycz, et al. (2020).  It seems plausible that whatever property of neural loss landscapes allows for the "catapult" effect may also be what allows for successful optimization at the Edge of Stability.  In fact, one way to think about the  "Edge of Stability"  regime is arguably as a never-ending series of "micro-catapults."  We will mention these points in the revision.
>
> >  I wonder if you have measured the 2nd eigenvalue during training as well? In particular, after the 1st eigenvalue has saturated at 2/eta, does the 2nd eigenvalue also "progressively sharpen" up to 2/eta ? (And so on for later eigenvals).
>
> Yes, this is exactly what happens. We will add this to the revision.
>
> > I am glad to see the experiments on deep linear networks, it suggests that it may be possible to theoretically understand this phenomenon in such simple settings. This would be a nice topic for future work.
>
> We agree!  Part of our motivation for the deep linear network experiment, and the 1-hidden-layer tanh experiment,  was to provide a starting point for authors who want to try to theoretically explain the empirical phenomena we discuss.
>
> We should mention that progressive sharpening only seems to kick in for _deep_ linear networks (ours was 30 layers).  It does not occur for e.g. two-layer linear networks, which were studied in Lewkowycz et al 2020.

---

### Official Review · AnonReviewer3 · 2020-10-29
**An interesting observation of GD and insightful discussions**

**Rating:** 5
**Confidence:** 4

**Review:**


This paper presents an interesting observation for GD. That is, the sharpness of the learnt model in the final phase of the training (measured by the largest eigenvalue of the training loss Hessian) hovers right at the value 2/\eta while the training loss. At the same time, the loss goes to unstable and non-monotonically decreasing. This pattern is consistent across architecture, activation functions, tasks, loss functions and BN. Comprehensive experiments are conducted to show this common observation. The paper is easy to follow.

Besides the empirical results in the main body, authors give insightful discussions in Intro and related work section. Specifically, authors propose a novel guess, that GD eventually transitions to “Edge of stability”, where GD can finally succeed with non-small enough step size. Although I am not sure how GD can do this, the concept of “Edge of stability” is still attractive.

I have two concerns for this work.
1) Authors did not investigate why sharpness finally hover over 2/\eta. Is it a trivial consequence followed by some relationship between the update rule of GD and the definition of sharpness, without any condition? Even if yes, we may further think about how to leverage it along the existing discussions in this paper. Hope to have authors' feedbacks on this later.
2) Given people use SGD to train neural networks, discussions about the insight from the observation of GD to SGD will enhance the impact of this paper.

---

> ### Author Response · Authors · 2020-11-13
> **response to "why does the sharpness hover over $2/\eta$"? (part 1)**
>
> Thank you very much for your helpful feedback!
>
> We don’t have a rigorous mathematical proof for why the sharpness hovers just above $2/\eta$, but we do have a partial explanation which we believe makes intuitive sense:
>
> 1. On the one hand, we empirically observe that in neural network training, the sharpness always “wants to” increase (progressive sharpening).
>
> 2. On the other hand, gradient descent with step size $\eta$ on a quadratic function will *provably* diverge if the sharpness is greater than $2 / \eta$.  In part #2 of this response (a separate comment), we show why this is true for _one-dimensional_ quadratic functions; Appendix B of our paper covers the more general case of a multidimensional quadratic function.  Now, neural network training objectives are not globally quadratic, but around any iterate one can make a second-order Taylor approximation to get a *local* quadratic approximation.  If the sharpness at some iterate is greater than $2/\eta$, then gradient descent would provably diverge if run on the quadratic Taylor approximation around that iterate.  Therefore, to the extent that gradient descent on the real training objective resembles gradient descent on the quadratic Taylor approximation, gradient descent will be unstable in neighborhoods where the sharpness exceeds $2/\eta$.
>
> In summary, on the one hand, gradient descent always “wants to” increase the sharpness; but on the other hand, gradient descent cannot linger for long in any neighborhood where the sharpness exceeds $2/\eta$.  Thus, it makes sense that gradient descent spends its time in regions where the sharpness is approximately equal to the maximum stable sharpness, $2/\eta$.  Indeed, one can non-rigorously argue that by process of elimination, this is the only possible outcome: the sharpness couldn’t be consistently much greater than $2/\eta$, because gradient descent would leave the neighborhood, yet the sharpness couldn’t be consistently less than $2/\eta$, because the sharpness would increase.
>
> A question you might have is: if gradient descent on quadratic functions diverges whenever the sharpness is greater than $2/\eta$, how can it be that in Figure 1 (and many other figures) the sharpness is consistently a little bit bigger than $2/\eta$?   The answer likely lies in the fact that on quadratic functions, the *speed* of divergence depends on the margin between the sharpness and the value $2/\eta$.  If the sharpness exceeds $2/\eta$ by a lot, then this divergence will be fast, but if the sharpness exceeds $2/\eta$ by a little, then this divergence will be slow.  So, if you run gradient descent in some region of the loss landscape in where the sharpness is just a little bit greater than $2/\eta$, then yes it’s true that gradient descent would *eventually* diverge if run on the quadratic Taylor approximation around that point, but this divergence would occur slowly — slower than the speed at which the quadratic Taylor approximation is itself changing.
>
> You can view the momentum experiments in Figure 5 as further confirmation of our hypothesis that the reason why the sharpness hovers at $2/\eta$ during vanilla gradient descent is because $2/\eta$ is the maximum stable sharpness of gradient descent on quadratic functions.  For momentum gradient descent (with either Polyak or Nesterov momentum), the maximum stable sharpness is given not by $2/\eta$ but by a more complicated expression that is printed analytically in Equation (1) and plotted as the dashed horizontal lines in Figure 5.  Sure enough, during momentum gradient descent, the sharpness hovers just above these values.
>
> It is true that we do not provide a rigorous mathematical proof for why the sharpness hovers just above $2/\eta$, nor do we have an explanation for why progressive sharpening occurs.  Rather, one major goal of our paper is to bring these phenomena to the attention of the ML theory / optimization theory communities so that future work can make our observations mathematically ironclad.  As we write in the conclusion:  “We think that understanding _precisely_ [emphasis new] how gradient descent is able to succeed at the Edge of Stability should be viewed as an important open problem in optimization theory.”
>
> We are eager to hear any more questions or comments that you have.

---

> ### Author Response · Authors · 2020-11-13
> **respone to "why does the sharpness hover over $2/\eta$ (part 2)**
>
> Here is a simple example which should illustrate where the threshold $2/\eta$ comes from.  Consider running gradient descent on the one-dimensional quadratic objective $f(x) = \frac{1}{2} a x^2$ (with $a > 0$) which has optimum at $x=0$ and global sharpness $a$. The gradient is $f'(x) = ax$, so the gradient descent update is:  $x_t = x_{t-1} - \eta a x_{t-1} = (1 - \eta  a)  x_{t-1}$.  Thus, the closed-form expression for the iterate at time t is: $x_t = (1 - \eta  a)^t x_0$.  Now, if $|1-\eta a| < 1$, then {$x_t$} will converge to the optimum zero, while if $|1 - \eta a| > 1$, then {$x_t$} will diverge.  Since the step size $\eta$ is positive, you can easily verify that $|1 - \eta a| > 1$ if and only if $a > 2/\eta$.  Thus, gradient descent on this problem diverges if and only if $a > 2 / \eta$.
>
> The story is similar for the more general case of a $d$-dimensional quadratic objective.  On a $d$-dimensional quadratic objective, if you express the iterates in the basis of Hessian eigenvectors, then gradient descent decomposes into $d$ separate one-dimensional problems, one for each Hessian eigenvector/eigenvalue.  See this blog post for the details: https://distill.pub/2017/momentum/.  For each of these problems, the role of $a$ is played by the corresponding eigenvalue.  Thus, if the sharpness (the leading eigenvalue) is greater than $2 / \eta$, then the gradient descent iterates will oscillate unstably along the leading Hessian eigenvector, until they eventually diverge.

---

> ### Author Response · Authors · 2020-11-22
> **regarding SGD**
>
> > Given people use SGD to train neural networks, discussions about the insight from the observation of GD to SGD will enhance the impact of this paper.
>
> This is a great question --- in fact, SGD is what we plan on working on next.  In our response [here](https://openreview.net/forum?id=jh-rTtvkGeM&noteId=IiUTgoOMqce) to R3, we summarize both (a) what empirically occurs to the sharpness during SGD, and (b) how we hypothesize that "the Edge of Stability" might generalize from GD to SGD.  See also Appendix A of our paper for an experiment which supports the hypothesis that some version of "Edge of Stability" occurs during SGD.
>
> It's also worth noting that since gradient descent is a special case of SGD, our "negative findings" for optimization theory apply not just to gradient descent, but also to SGD.  In particular, if $L$-smoothness-based convergence analyses do not apply to gradient descent (as we have shown), then they also probably don't apply to SGD, since GD is a special case of SGD.

---

### Official Review · AnonReviewer1 · 2020-11-06
**Interesting empirical observation but may be missing some key points**

**Rating:** 6
**Confidence:** 4

**Review:**

Summary of results: This empirical paper finds that deep neural network "sharpness" (as measured by the top eigenvalue of the Hessian) tends to saturate at or hover just above the value 2/\eta, where \eta is the step size in gradient descent (GD), during the course of optimization. This is accompanied by non-monotonicity in the loss. (Results also hold for gradient descent with momentum.) This phenomena occurs for full-batch GD, a variety of tasks, and two different loss functions (square loss and cross-entropy, although in the latter case late-time dynamics of the sharpness is different, with the sharpness decreasing). The result also holds across a variety of architectures (VGG-11, with and without batch norm, convolutional networks and fully-connected networks with different nonlinearities, a deep linear network, a Transformer model -- although I comment on the architecture dependence below.) The authors refer to this phenomenon of sharpness hovering at or above the 2/\eta bound as optimization on the "edge of stability." The authors posit that this observation goes against our current understanding of optimization in deep learning: (i) theoretical work may use assumptions such as monotonic descent or L-smoothness, which is in violation of the "edge of stability" regime where the loss behavior is not monotonic; (ii) relatedly, it corresponds to a regime of instability in a quadratic Taylor approximation, so this is not a good assumption either. The empirical observations do not appear to straightway carry over to SGD (although there is some discussion of similarities in the appendix) -- the authors leave this is as an open problem.

Quality and clarity: The work is of good quality. The experiments that are presented clearly demonstrate the described phenomenon, attempt to sample from a variety of problems (ranging from a simple 1D regression task to training VGG-11 on CIFAR-10 in Fig.1), and are well-explained.

Originality: The observed phenomena (of sharpness progressively increasing and then hovering at or near 2/\eta through much of GD dynamics) appears new to the best of my knowledge. (However, prior papers have pointed out that GD optimization can stably proceed for learning rates above 2/\eta and are accompanied with nonmonotonic loss: one example is Lewkowycz, et al. (2020).)

Significance: I think the observations are somewhat important, although perhaps not as much as the authors seem to stress in writing (e.g. in the abstract, "Our results ... shed significant light on the dynamics of gradient descent with a fixed step size ..." -- this paper focuses only on empirical observations rather than explanation of the mechanism behind the stability). The results are for full-batch GD which somewhat limits the applicability of the results to practice where SGD is more common. Nonetheless, I do think understanding how optimization is stabilized in this regime is an interesting problem.

Other comments: I think that a key point not fully understood (at least, as expressed in the writing) or investigated in the experiments is the role of network width (i.e. I'm skeptical that the observations will be "consistent across architectures" as expressed in the opening paragraph in the Introduction if wider networks are also investigated empirically). This is a main reason for not giving the paper a higher score. In Appendix D, the authors try to reconcile the results with existing theory on infinite-width limits. Fig. 11(c) shows that across networks of varying width but trained at the same learning rate, wider networks end up with smaller values of sharpness at the end of training (here, say that all experiments are stopped at the same value of the training loss). Hence if a narrower network saturates the 2/\eta bound at late times in gradient descent, the wider networks will fall below this value. (This is, of course, consistent with no evolution in the curvature in the infinite width limit.) That is to say, whether or not the "edge of stability" regime is reached depends quite strongly on how wide the networks are.

While some realistic networks are explored (e.g. VGG-11, *narrow* Transformers), which is a positive point, all of the remaining networks used in the experiments (as far as I can gather) are on the narrow side (e.g. layer widths of order 100 or 200). I think a shortcoming of the paper is that the strong width (i.e. architecture) dependence of the phenomenon is not fully appreciated or discussed by the authors (e.g. for instance, by discussing in the main text that it only sets in for narrow networks) or investigated empirically. (Alternatively, noting for readers that all the networks chosen are rather narrow, out of transparency.)  I note that the authors do mention in Appendix D: "...one might hypothesize that progressive sharpening might attenuate as networks (with NTK parameterization) become increasingly wide." However, I don't think that NTK parameterization is necessary for this to be true. In short, I think the width dependence of the phenomenon is an important factor that affects the significance and applicability of the observations and could have been treated with greater transparency (with additional experiments and additions to the main text and abstract).

A comment on relation to prior work: the authors write that Lewkowycz, et al. (2020) imply that "actual progress would occur in regions where the sharpness remains strictly less than 2/\eta. Our experiments demonstrate otherwise." I don't believe this is a conclusion of that paper (progress happens when sharpness is above 2/\eta). (Note also that the paper tends to study wide networks.) Could the authors elaborate on what they mean here?

---

> ### Author Response · Authors · 2020-11-13
> **regarding Lewkowycz, et al. (2020)**
>
> Here, we respond to your points regarding Lewkowycz, et al. (2020).
>
> > A comment on relation to prior work: the authors write that Lewkowycz, et al. (2020) imply that "actual progress would occur in regions where the sharpness remains strictly less than 2/\eta. Our experiments demonstrate otherwise." I don't believe this is a conclusion of that paper (progress happens when sharpness is above 2/\eta). (Note also that the paper tends to study wide networks.) Could the authors elaborate on what they mean here?
>
> It is true that Lewkowycz, et al. (2020) exhibits a stage of training in which the sharpness is greater than $2 / \eta$, and where the training loss behaves non-monotonically.  However, in Lewkowycz, et al. (2020) this “catapult” stage of training is understood to be temporary and fleeting, and actual optimization (in the sense of the loss consistently going down) is only supposed to commence after this “catapult” stage is definitively over.  By contrast, our paper details how the sharpness is $\approx 2 / \eta$ (and the loss is non-monotonic) for the bulk of training --- notably, at the same time as the loss is consistently decreasing.
>
> In more detail, Lewkowycz, et al. (2020) examined what happens when the step size $\eta$ is set greater than $2 /\lambda_0$, where $\lambda_0$ is the initial sharpness.  In this scenario, at the beginning of training, gradient descent oscillates unstably along the leading Hessian eigenvector until it is flung from its initial position.  What  Lewkowycz, et al. (2020) showed is that, rather than diverge entirely, on some architectures gradient descent will “land on its feet” (our words) in a new region that is flat enough to accommodate the step size, i.e. a region where the sharpness is $< 2 /\eta$.  To quote from that paper: if $\eta > 2/\lambda_0$, then
>
> > optimization begins with a period of exponential growth in the loss, coupled with a rapid decrease in the curvature [sharpness], until curvature **stabilizes at a value $\lambda_{\text{final}} < 2 / \eta$.  Once the curvature drops below $2 / \eta$, training converges**…
>
> Note that the new sharpness is called $\lambda_{\text{final}}$, indicating that once gradient descent finds a region where the sharpness is $< 2 /\eta$, the sharpness is supposed to stay there for the remainder of training.  Indeed, this is stated explicitly in their section 4.1:
> > one striking prediction of the model is that after a period of excursion, the logic differences settle back to O(1) values, the **NTK stops changing** [emphasis ours], and evolution is again well approximated by a linear model with constant kernel at large with.
>
> Notice also that they write: “once the curvature drops below $2 / \eta$, training converges.”  This is what we meant when we wrote that “actual progress would occur in regions where the sharpness remains strictly less than $2 / \eta$.”
>
> Their overall story is illustrated by their Figure 2(a) and 2(b).  In Figure 2(b), which plots the sharpness, we can see that after an initial drop (the catapult), the sharpness ceases to change for the remainder of training.  In Figure 2(a), which plots the train loss, we can see that after an initial spike (the catapult), the train loss monotonically decreases for the remainder of training.
>
> To summarize, the overall story in Lewkowycz, et al. (2020) is fully consistent with the conventional local-quadratic view of optimization.  In their paper, the main function of the large learning rate is to trigger a switch from a sharp initialization to a flat initialization.  After this initialization switch, the conventional optimization wisdom concerning step sizes is supposed to kick in again.
>
> -----
>
>
> > However, prior papers have pointed out that GD optimization can stably proceed for learning rates above 2/\eta and are accompanied with nonmonotonic loss: one example is Lewkowycz, et al. (2020).
>
> We are not aware of any such papers apart from Lewkowycz, et al. (2020) (which we discussed above), as well Xing et al. (2018) [1], which we discussed in our Related Work section.  Xing et al. (2018) empirically studied full-batch gradient descent, and noted that the train loss eventually starts behaving non-monotonically.  However, that paper didn’t relate this phenomenon to dynamics of the sharpness, or discuss implications for optimization theory.
>
> If there are other papers which observe either of these two effects, we are happy to add the appropriate citations.
>
> [1] Chen Xing, Devansh Arpit, Christos Tsirigotis, and Yoshua Bengio. A walk with sgd.

---

> ### Author Response · Authors · 2020-11-13
> **regarding network width (part 1/2)**
>
> Thank you for your astute review!
>
> We completely agree with you that the current draft inadequately treats the issue of network width.  Indeed, we independently came to this conclusion ourselves soon after submitting the draft.
>
> The clearest way to think about the effect of network width is via the notion of the gradient flow trajectory: for narrow networks, the sharpness rises a lot over the gradient flow trajectory, while for wide networks, the sharpness only rises a little bit (tending towards “not at all” as the width $\to \infty$).  We’ve uploaded a new section, Appendix G, in which we train fully-connected networks via gradient flow at a range of widths.  (To train using gradient flow, we use Runge-Kutta RK4 to numerically integrate the gradient flow ODE.)  We do this for both standard parameterization and NTK parameterization.  You can see that for both standard and NTK parameterization, progressive sharpening happens to a greater degree for narrow networks than for wide networks.  That is, if we define $\lambda_0$ to be the initial sharpness, and we define $\lambda_{\max}$ to be the maximum sharpness along the gradient flow trajectory, then we can see that both $\lambda_{\max}$ and $\lambda_{\max} / \lambda_0$ get smaller as the width gets larger.  Now, there does seem to be a slight difference between standard and NTK: for NTK parameterization, it looks like $\lambda_{\max}/\lambda_0$ will converge to 1 as width $\to \infty$ (consistent with NTK theory), whereas that may or may not be true for the standard parameterization; at the very least, the rate of convergence seems to be a lot slower for standard parameterization.
>
> Now let’s move from gradient flow to gradient descent.  You write: “whether or not the ‘edge of stability’ regime is reached depends quite strongly on how wide the networks are.”  In our view, the best way to frame this issue is not “does gradient descent enter the Edge of Stability or not”, but rather “for which step sizes does gradient descent enter the Edge of Stability,” and in particular “how unreasonably small are the step sizes for which gradient descent avoids entering the Edge of Stability.”  Recall that if $\eta$ is less than $2 / \lambda_{\max}$ (let’s call these step sizes “stable step sizes”), then gradient descent will track the gradient flow trajectory all the way to the end, without ever entering the Edge of Stability.  On the other hand, if $\eta$ is between $2/\lambda_0$ and $2/\lambda_{\max}$, then gradient descent will enter the Edge of Stability at some point.  We showed empirically in our paper that stable step sizes are suboptimal from the perspective of convergence speed. The difference between narrow and wide networks is the _degree_ of suboptimality of stable step sizes.  For wide networks, $\lambda_{\max}$ will be only a little bit larger than $\lambda_0$, and so stable step sizes will only be _mildly_ suboptimal (converging to “not at all” suboptimal as the width tends to $\infty$ under NTK parameterization and maybe under standard parameterization too).  On the other hand, for narrow networks (e.g. the width 32 network in Appendix G), $\lambda_{\max}$ will be a lot larger than $\lambda_0$, and stable step sizes will be _highly_ suboptimal.
>
> _continued below_

---

> > ### Author Response · Authors · 2020-11-13
> > **regarding network width (part 2/2)**
> >
> > _note: this is a continuation of the parent post_
> >
> > **Realistic architectures.**  Here is a crucial empirical point: in our experience, when training realistic architectures (say, VGG-11, either with or without BN) on realistic datasets (e.g. CIFAR-10), the sharpness always _dramatically_ increases under gradient flow / gradient descent.  In other words, realistic architectures on realistic datasets seem to behave more like the narrow (width 32) networks in Appendix G than the wide networks in Appendix G.  In fact, for realistic architectures on realistic datasets, progressive sharpening occurs to a much _greater_ degree than anything depicted in Appendix G.
> >
> > To give a concrete example, we’re currently in the middle of training a VGG-BN on CIFAR-10 using gradient flow, and the sharpness has already increased from 6 at initialization to 2139 at 89% accuracy, which is an increase by a factor of 356 (and we’re not even finished training yet!).  Back-of-the-envelope math shows that training to 89% accuracy at a stable step size of 2/2139 = 0.000935 would require 12,834 iterations.  (The 89% accuracy is reached at time 12, and 12/0.000935 = 12,834.)  Yet on the other hand, we’ve verified that the network can be trained to 99% accuracy in ~400 iterations using GD at step size 0.08.   In other words, training the network at a stable step size is suboptimal (in terms of iteration count) by a factor of _at least_ 32.  That is why our submission said that stable step sizes are “unreasonably” small.
> >
> > This being said, we do think that it would be a good idea for us to better separate this empirical point about realistic architectures from the rest of our paper.  The statement “stable step sizes are suboptimally small” seems to hold universally.  However, while the stronger statement “stable step sizes are so suboptimal as to be _unreasonable_” does seem to hold for realistic architectures on realistic datasets, it does not hold universally  --- for instance, it does not hold for the NTK width 1024 network in Appendix G.
> >
> > We do not know why progressive sharpening occurs to such a dramatic degree for realistic architectures on realistic datasets.  One possibility is that these networks are, in the grand scheme of things, narrow, and if they were made wider, progressive sharpening would no longer occur to such a dramatic degree.  However, it may be the case that the requisite widths would need to be so large as to not be implementable on standard GPUs.  To explore this question, in the revision, we will conduct an experiment similar to Appendix G, but with the WideResNet architecture (which has a tunable knob for width).  We will use as large a width as our GPU permits.  We strongly suspect that even for the widest WRNs implementable on a GPU,  progressive sharpening will still occur to a dramatic degree (i.e. the sharpness will rise during gradient flow by a factor of at least, say, 100), indicating that stable step sizes are so suboptimal as to be completely unreasonable.  However, if that hypothesis turns out to be false, we will explicitly acknowledge in the main text that there do exist realistic networks for which stable step sizes are reasonable.
> >
> > In summary, for the final revision, we promise the following:
> >
> > 1. When we first introduce progressive sharpening, we will note that progressive sharpening occurs to a smaller degree for wider networks, and that (at least under NTK parameterization, and maybe under standard parameterization too) it stops happening as the width goes to infinity.  Currently this information is relegated to the appendix, but we will fully work this point into the paper’s narrative.
> > 2. We will carefully note that while the statement “stable step sizes are suboptimally small” seems to hold universally (to reiterate, stable step sizes are those at which GD never enters the Edge of Stability), the stronger statement “stable step sizes are completely impractical/unreasonable” does not necessarily hold for sufficiently wide networks.  That said, it does seem to hold for realistic architectures, like VGG, on realistic datasets, like CIFAR-10.
> > 3. For WideResNets on CIFAR-10, we will systematically study progressive sharpening as the WRN width parameter is varied.
> >
> > Please let us know if there are other changes you think would enhance the paper.

---

> > > ### Comment · AnonReviewer1 · 2020-11-25
> > > **effect of network width**
> > >
> > > Thank you for your detailed reply! I also appreciate the addition of Appendix G in the revision.
> > >
> > > I do agree that "realistic networks" (which very often have a few bottleneck layers of narrow width) do show the effect much more strongly and behave like the ~ width 32 networks. On the other hand, the max sharpness drops quite rapidly with width (and e.g. 128 is not a particularly wide network, which as I understand would not exhibit the "Edge of Stability" phenomenon) -- so I think it is important to weave it in, in a central way, in the main text and abstract, as you suggest.
> > >
> > > (One comment: a result in Lewkowycz, et al. (2020) is that, when the learning rate is in the "catapult phase" they define, the infinite width limit gives rise to non-NTK dynamics -- for these learning rates, as networks are made wider, I would expect that the "progressive sharpening" happens to an even lesser degree than would be expected from NTK dynamics.)

---

> > > > ### Author Response · Authors · 2020-11-25
> > > > **follow-up**
> > > >
> > > > Thank you for your comments!
> > > >
> > > > We completely agree that we need to weave the dependence on width into the main text.
> > > >
> > > > A few follow-up points:
> > > >
> > > > > 128 is not a particularly wide network
> > > >
> > > > In our experience, it's tricky to judge networks as "wide" or "narrow" in absolute terms --- these judgements have to be made in the context of a specific dataset.   This experiment was conducted on a 5,000-size subset of CIFAR-10.   In general, progressive sharpening happens to a greater degree when the dataset is "hard" or when the dataset is large.    So, we had used the full CIFAR-10 dataset, more sharpening would have occurred.  On the other hand, if we had used a 5,000-size subset of fashion-MNIST (a simpler dataset, in our experience), less sharpening would have occurred.  We should add an experiment to the paper which demonstrates that progressive sharpening depends on dataset difficulty and dataset size.
> > > >
> > > > > which as I understand would not exhibit the "Edge of Stability" phenomenon
> > > >
> > > > From Figure 15 (right pane), we can see that for the width 128 NTK-parameterized network, the sharpness rose by a factor of 15x during training.  So, in order to avoid the Edge of Stability, the learning rate would have to be less than $\frac{2}{15 \lambda_0}$.  By contrast, networks can usually be successfully trained at learning rates of $\frac{2}{\lambda_0}$ (though that is an empirical point which needs to be verified anew for each network), which in this case would be larger by a factor of 15x.  Now, training with the 15x larger step size would not be 15x faster (you don't get linear speedup), but in our experience it would likely be faster by at least, say, 5x (though that is another empirical point which would need to be verified).  So, based on our experience, we'd estimate that for the width 128 NTK parameterized network, training with _non_-Edge-of-Stability step sizes would be 5x slower than training with Edge-of-Stability step size.  So we'd still say that for this network, if one were training with gradient descent, one would end up using an Edge-of-Stability step size.  But it's certainly less clear-cut than, say, the VGG networks, where it is be almost impossible to train at a non-Edge-of-Stability step size.

---

### Decision · Program_Chairs · 2021-01-07
**Final Decision**

**Decision:**

Accept (Poster)

**Comment:**

The paper demonstrates that Gradient Descents generally operates in a regime where the spectral norm of the Hessian is as large as possible given the learning rate.

The paper presents a very thorough empirical demonstration of the central claim, which was appreciated by the reviewers.

A central issue to me in accepting the work was its novelty. Prior work has shown very closely related effects for SGD. The reviewers appreciated in discussions the novelty of the precise claim about the spectral norm hovering at around $\frac{2}{\eta}$. R4 and R2 also raised the issue that the related work discussion is not sufficient. Please make sure that you discuss very carefully related work in the paper, including a more detailed discussion in the Introduction.

The two key issues raised by R3, who voted for rejection, were that (1) the work studies Gradient Descent (rather than SGD), and (2) lack of theory. I agree with these concerns. Perhaps the Authors should address (1) by citing more carefully prior work that shows that a similar phenomenon does seem to happen in training with SGD. As for (2), I agree here with R1,R2 and R4 that empirical evaluation is a key strength of the paper.

Based on the above, it is my pleasure to recommend the acceptance of the paper. Thank you for submitting your work to ICLR, and please make sure you address all remarks of the reviewers in the camera-ready version.